EMBO
Molecular Medicine

# PlexinD1 is a driver and a therapeutic target in advanced prostate cancer

Jing Wei [1,9], Jing Wang [1,2,9 ✉], Wen Guan [1], Jingjing Li[1,7], Tianjie Pu [1,8], Eva Corey [3], Tzu-Ping Lin[4,5], Allen C Gao[6] & Boyang Jason Wu [1 ✉]

## Abstract

**Aggressive prostate cancer (PCa) variants associated with androgen receptor signaling inhibitor (ARSI) resistance and metastasis remain poorly understood. Here, we identify the axon guidance semaphorin receptor PlexinD1 as a crucial driver of cancer aggressiveness in metastatic castration-resistant prostate cancer (CRPC). High PlexinD1 expression in human PCa is correlated with adverse clinical outcomes. PlexinD1 critically maintains CRPC aggressive behaviors in vitro and in vivo, and confers stemness and cellular plasticity to promote multilineage differentiation including a neuroendocrine-like phenotype for ARSI resistance. Mechanistically, PlexinD1 is upregulated upon relief of AR-mediated transcriptional repression of PlexinD1 under ARSI treatment, and subsdquently transactivates ErbB3 and cMet via direct interaction, which triggers the ERK/AKT pathways to induce noncanonical Gli1-dictated Hedgehog signaling, facilitating the growth and plasticity of PCa cells. Blockade of PlexinD1 by the protein inhibitor D1SP restricted CRPC growth in multiple preclinical models. Collectively, these findings characterize PlexinD1's contribution to PCa progression and offer a potential PlexinD1-targeted therapy for advanced PCa.**

**Keywords** Castration-resistant Prostate Cancer; Cellular Plasticity; Neuroendocrine Differentiation; Androgen Receptor Signaling inhibitors; PlexinD1
**Subject Categories** Cancer; Molecular Biology of Disease; Urogenital System

## Introduction

Prostate cancer (PCa) is the second most frequently occurring cancer and the fifth leading cause of cancer mortality among men worldwide in 2022 (Bray et al, 2024). Androgen receptor (AR) is considered the primary oncoprotein in PCa, making androgen deprivation therapy (ADT) the mainstay treatment for PCa. Although ADT produces a favorable response initially, the vast majority of tumors relapse to aggressive castration-resistant disease (CRPC), which is often associated with treatment resistance, metastasis, and death (Saad and Fizazi, 2015). The development of CRPC has been attributed to multiple mechanisms including AR amplification and mutations, AR splice variants, intratumoral and alternative androgen production, upregulation of AR-substituted transcription factors like glucocorticoid receptor (GR), and activation of growth factor signaling crosstalk with the AR pathway (Chandrasekar et al, 2015; Watson et al, 2015). Other than the AR-centric underpinnings, emerging evidence has suggested that cellular plasticity, especially lineage plasticity often triggered by androgen receptor signaling inhibitors (ARSIs) such as enzalutamide (ENZ), is a key mechanism underlying CRPC development and progression, enabling PCa cells to switch from a luminal lineage to another, such as a neuroendocrine (NE) lineage, after redifferentiation, transdifferentiation, or cellular reprogramming to evade the effect of therapies (Davies et al, 2018). Despite these mechanistic explorations, the molecular determinants that confer CRPC and associated malignant behaviors still remain largely undefined, which hinders the development of new, effective molecularly targeted therapies to prolong patients' survival.

To fill this knowledge gap, we performed a gene expression profiling analysis of isogenic androgen-dependent PCa and bona fide metastatic CRPC cells, and identified PlexinD1 as a putative CRPC driver. Plexins are a large family of transmembrane proteins that act as the receptors of semaphorins (Semas) to regulate axon guidance in the developing central nervous system. Recent studies have revealed that plexins and semaphorins have aberrant expression levels and play multiple roles in a spectrum of cancers, including PCa. For example, the B-type plexins, which are the best studied in PCa, are overexpressed in PCa and context-dependently promote PCa growth, metastasis, chemoresistance, and AR/GR nuclear translation in response to Sema3C or Sema4D (Garg et al, 2023; Li et al, 2020; Peacock et al, 2018; Shorning et al, 2023;

[1]Department of Pharmaceutical Sciences, College of Pharmacy and Pharmaceutical Sciences, Washington State University, Spokane, WA 99202, USA. [2]Department of Medical Oncology, Dana-Farber Cancer Institute, Boston, MA 02215, USA. [3]Department of Urology, University of Washington, Seattle, WA 98195, USA. [4]Department of Urology, Taipei Veterans General Hospital, Taipei, Taiwan 11217, Republic of China. [5]Department of Urology, School of Medicine and Shu-Tien Urological Research Center, National Yang Ming Chiao Tung University, Taipei, Taiwan 11221, Republic of China. [6]Department of Urologic Surgery, University of California, Davis, Sacramento, CA 95817, USA. [7]Present address: Engineering Research Center of Cell & Therapeutic Antibody, School of Pharmacy, Shanghai Jiao Tong University, Shanghai 200240, China. [8]Present address: Department of Surgery, Memorial Sloan Kettering Cancer Center, New York, NY 10065, USA. [9]These authors contributed equally: Jing Wei, Jing Wang. ✉E-mail: jing_wang1@dfci.harvard.edu; boyang.wu@wsu.edu

Williamson et al, 2019; Wong et al, 2007). We also recently reported that PlexinA2 in conjunction with Sema3C and neuropilin-1 mediates perineural invasion of PCa cells under the control of monoamine oxidase A (Yin et al, 2021). Nevertheless, the role and mechanism of PlexinD1 in CRPC have not been elucidated, nor has its targeting potential been evaluated. In this study, we investigated whether PlexinD1 is a crucial executor driving CRPC and set out to develop a PlexinD1-targeted inhibitor against aggressive PCa.

# Results

## PlexinD1 and associated Sema ligands are upregulated in CRPC cells

To identify potential molecular drivers in CRPC, we performed RNA sequencing (RNA-seq) analysis in LNCaP and ENZ-resistant C4-2B (C4-2B$^{ENZR}$) cells. The C4-2B$^{ENZR}$ cell line is a bone metastatic castration-resistant (mCRPC) derivative of the androgen-dependent LNCaP cell line with acquired resistance to ENZ, representing CRPC cell lines truly resistant to ARSIs (Liu et al, 2015). Gene set enrichment analysis (GSEA) revealed that the pathways upregulated following acquisition of resistance to castration and ENZ treatment in C4-2B$^{ENZR}$ cells are associated with stem/neuronal phenotype, hypoxia, epithelial-mesenchymal transition (EMT), development, and inflammation, which was paralleled by downregulated AR signaling, compared to LNCaP cells (Fig. 1A). In searching for potential pathways associated with the CRPC phenotype of C4-2B$^{ENZR}$ cells, we subjected the transcriptomic profiling data from C4-2B$^{ENZR}$ versus LNCaP cells to enrichment analysis using the gene sets from the Kyoto Encyclopedia of Genes and Genomes (KEGG) Pathway Database. We identified the axon guidance pathway as one of the top pathways significantly enriched in C4-2B$^{ENZR}$ cells. We also found significant enrichment of the mitogen-activated protein kinase (MAPK) and phosphoinositide 3-kinase (PI3K)-AKT signaling pathways, both reported to be activated in CRPC cells (He et al, 2022) (Fig. 1B). Examining all detectable genes belonging to the four prominent families of axon guidance cues (Slit/Robo, semaphorin/plexin, Netrin/Unc5/DCC, and ephrin/Eph) (Dickson, 2002), we found 29 genes differentially expressed ($p < 0.05$) in C4-2B$^{ENZR}$ versus LNCaP cells, including 20 overexpressed and 9 underexpressed genes, among which ROBO2 and PLXND1 (encoding PlexinD1) are the two most upregulated axon guidance pathway genes (Fig. 1C; Appendix Table S1). Given more semaphorin/plexin (10/29) than Slit/Robo (2/29) genes altered in C4-2B$^{ENZR}$ versus LNCaP cells, we hypothesized that the semaphorin/plexin family genes are influential in CRPC and therefore focused on PlexinD1 as a possible regulator of CRPC. GSEA also revealed the upregulation of two gene ontology (GO) gene sets related to the semaphorin/plexin signaling pathway in C4-2B$^{ENZR}$ relative to LNCaP cells (Fig. 1D).

Next, we confirmed upregulation of PlexinD1 protein along with Sema3E and Sema3C, which both are ligands directly binding to PlexinD1 without requiring the presence of neuropilin co-receptors (Gu et al, 2005; Smolkin et al, 2018), in C4-2B$^{ENZR}$ compared with LNCaP cells. We also found lower expression of AR and PSA (a classical AR target protein) and enhanced expression of NE

markers SYP and NSE in C4-2B$^{ENZR}$ cells, consistent with our previous findings that C4-2B$^{ENZR}$ cells exhibit treatment-induced NE traits (Bland et al, 2021) (Fig. 1E). By an in situ proximity ligation assay (PLA), we visualized endogenous PlexinD1-Sema3E and PlexinD1-Sema3C protein complexes in LNCaP and C4-2B$^{ENZR}$ cells. Quantitative analysis of fluorescence restricted to the cytoplasm revealed 10- and 4-fold increases in PlexinD1 interaction with Sema3E and Sema3C, respectively, indicative of potentially higher ligand-induced PlexinD1 receptor activity, in C4-2B$^{ENZR}$ compared with LNCaP cells. Parallel incubation with a PlexinD1 antibody only, as a negative control, yielded minimal fluorescence in the cells (Fig. 1F). Analyzing a panel of non-malignant and malignant human prostate cell lines, we further showed that PlexinD1 protein is expressed at higher levels in multiple CPPC cell lines regardless of AR status, including AR+ 22Rv1, AR-repressed C4-2B$^{ENZR}$, and AR- PC-3, DU145, LASCPC-01 and NCI-H660 cell lines, compared to normal prostate epithelial RWPE-1 and androgen-dependent AR+ LNCaP and LAPC4 cell lines (Fig. 1G). A similar trend was observed for PLXND1 mRNA across different PCa cell lines (Appendix Fig. S1A). Moreover, Sema3E and Sema3C have similar protein expression patterns to PlexinD1, with more present in most CRPC cell lines than in RWPE-1 and androgen-dependent PCa cells (Fig. 1G). Collectively, these results suggested upregulation of PlexinD1 in CRPC cells.

## Elevated PlexinD1 expression levels are associated with worse clinical outcomes in PCa

To seek initial evidence of PlexinD1's role in PCa, we assessed PlexinD1 protein levels in a tissue microarray (TMA), including primary human PCa ($n = 50$) and normal human prostate tissues ($n = 30$), by immunohistochemistry (IHC). We found higher epithelial expression of PlexinD1 in cancerous relative to normal tissue (Fig. 2A). Categorizing tumor samples on the basis of high Gleason scores (GS 7–10) versus low Gleason scores (GS 4–6), we demonstrated increased expression of PlexinD1 upon progression to aggressive, poorly differentiated high-grade PCa (Fig. 2B). Examining two independent PCa clinical datasets [GSE3325 (Varambally et al, 2005) (Data ref: Varambally et al, 2005) and GSE3933 (Lapointe et al, 2004) (Data ref: Lapointe et al, 2004)], we also showed that PLXND1 mRNA levels were uniformly upregulated in cancer compared with normal tissue (Fig. 2C). To evaluate a potential association of PlexinD1 with hormone therapy, we assessed PlexinD1 expression in a NYU PCa TMA ($n = 86$) by IHC and detected higher PlexinD1 levels in post- versus pre-hormone therapy tumors (Fig. 2D). Concordantly, analysis of two independent PCa datasets showed increased PLXND1 mRNA levels in tumors from patients who had biochemical recurrence [GSE21032 (Taylor et al, 2010) (Data ref: Taylor et al, 2010)] as well as in hormone-refractory tumors [GSE6099 (Tomlins et al, 2007) (Data ref: Tomlins et al, 2007)] compared with respective controls (Fig. 2E). We also investigated the association of PlexinD1 with metastasis, a common feature of CRPC, in another PCa TMA comprising primary PCa ($n = 11$) and bone metastasis ($n = 19$). IHC analysis revealed higher PlexinD1 expression in bone metastasis relative to primary tumor samples (Fig. 2F), which was congruent with PLXND1 mRNA increases in metastasis versus primary tumors from four independent PCa clinical datasets [Vanaja (Vanaja et al, 2003), Yu (Yu et al, 2004), GSE21032, and

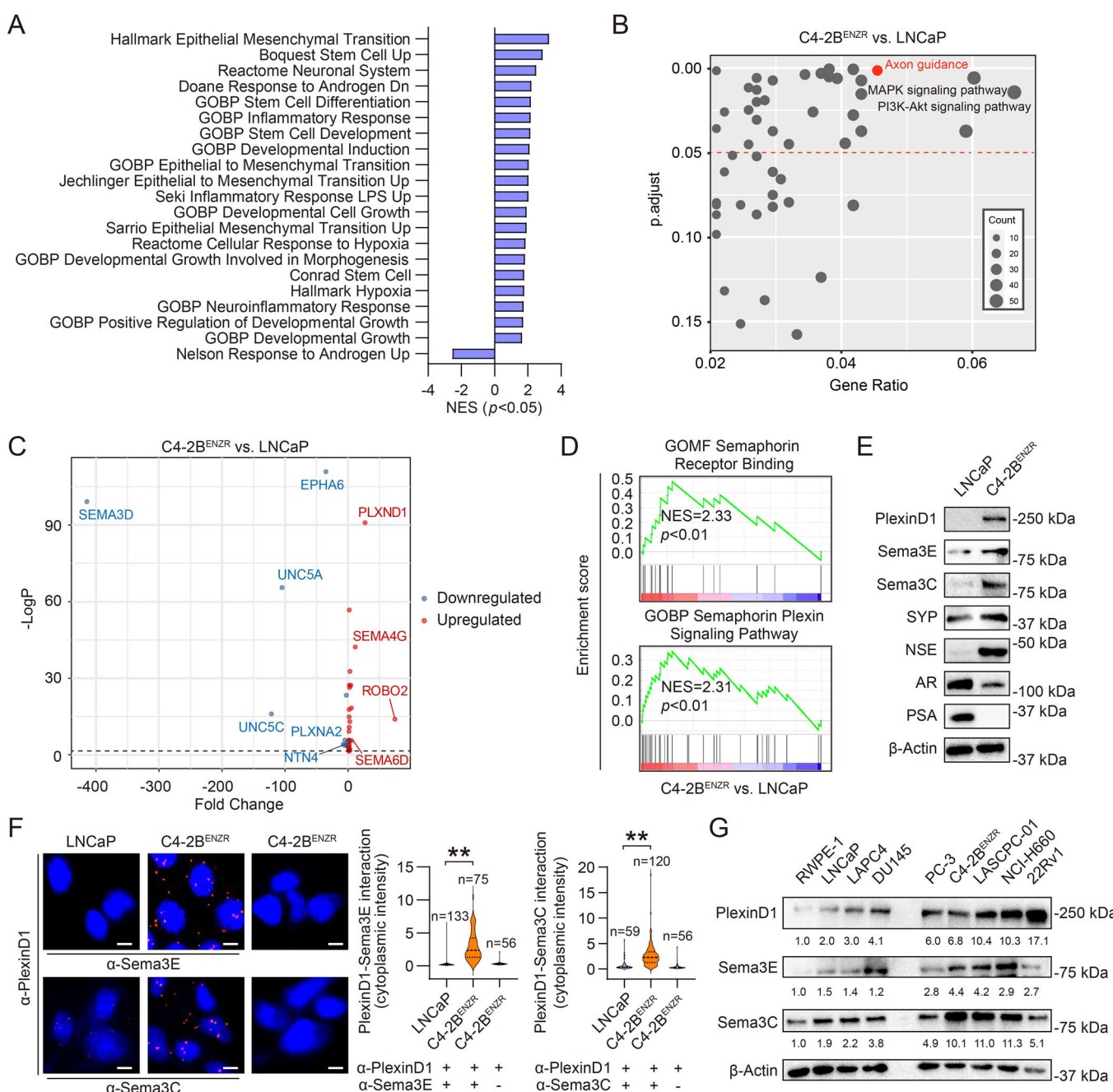

**Figure 1. PlexinD1 and associated Sema ligands are upregulated in CRPC cells.**

(A) GSEA of multiple cellular event and plasticity related gene sets enriched in C4-2B^ENZR vs. LNCaP cells. (B) Dot plot depicting the axon guidance pathway among KEGG pathways highly enriched in C4-2B^ENZR vs. LNCaP cells. (C) Volcano plot of axon guidance regulator genes detected in C4-2B^ENZR vs. LNCaP cells ($n = 2$ biological replicates). (D) GSEA plots of two semaphorin/plexin signaling-related gene sets enriched in C4-2B^ENZR vs. LNCaP cells. (E) Western blot of PlexinD1, Sema3E, Sema3C, and NE and AR signaling markers in LNCaP and C4-2B^ENZR cells. (F) Representative PLA images and quantification of PlexinD1-Sema3E/Sema3C interaction by per-cell cytoplasmic fluorescence intensity in LNCaP and C4-2B^ENZR cells. PlexinD1 antibody incubation alone served as negative controls. The $n$ of each group indicates the number of cells examined for quantification. Scale bars: 5 μm. (G) Representative Western blotting images and quantification of PlexinD1, Sema3E, and Sema3C protein expression in a panel of cell lines as indicated, with the averaged individual protein levels after normalization to β-actin in RWPE-1 cells from three independent experiments set as 1. Note that unequal amounts of total cell lysates from different cell lines were loaded intentionally due to the large discrepancy in target protein levels among different cell lines. Data information: In (F), data are presented as mean ± SEM. In (A, D), P values were determined by permutation test. In (C), P values were determined by unpaired two-tailed Student's t-test. In (F), P values were determined by one-way ANOVA with Dunnett's multiple comparisons test. **P < 0.01. Exact P values are listed in Appendix Table S4. Source data are available online for this figure.

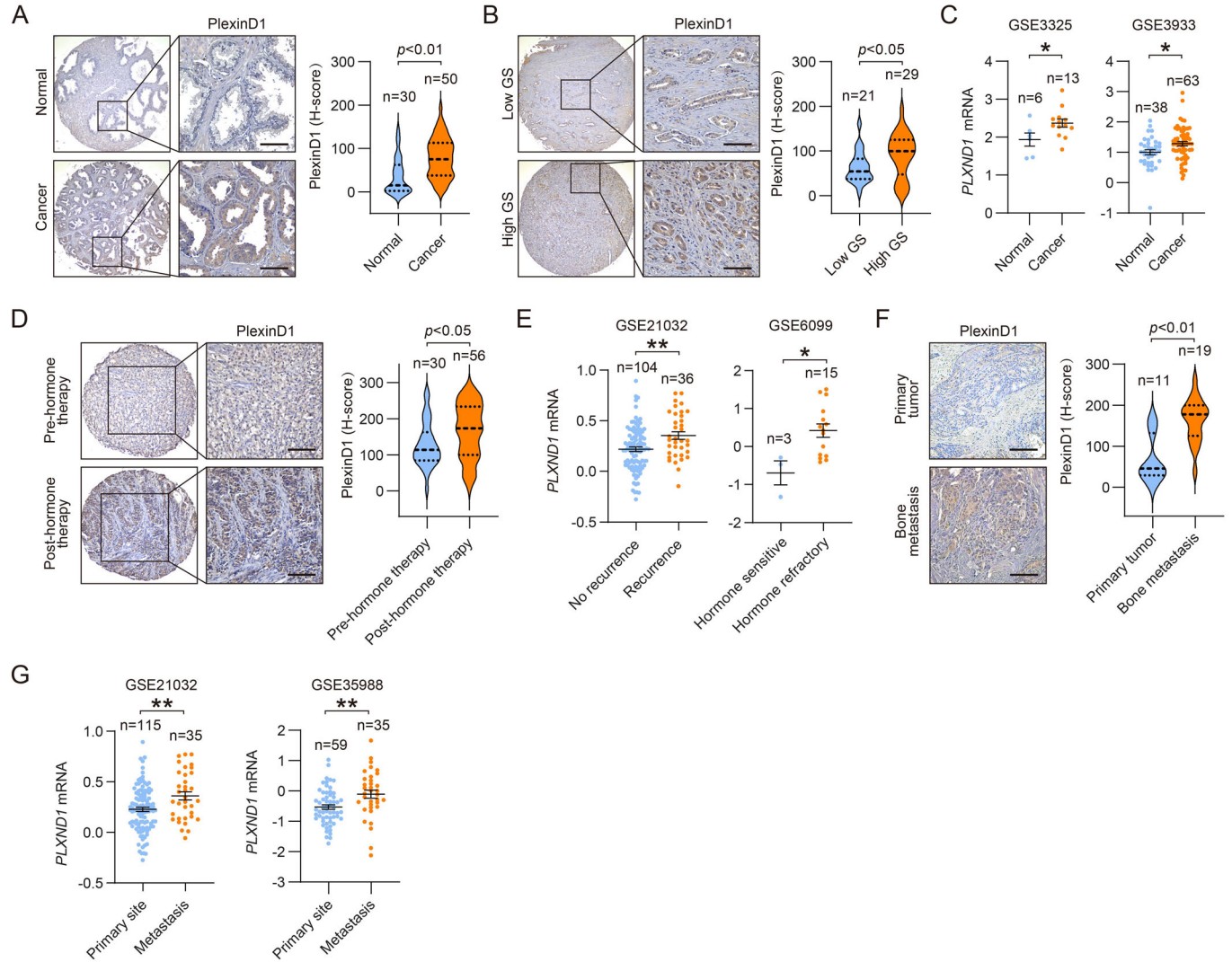

**Figure 2. Elevated PlexinD1 expression levels are correlated with worse clinical outcomes in PCa.**

(A) Representative images and quantification of PlexinD1 IHC staining in primary PCa vs. normal prostate tissue from a US Biomax TMA. Scale bars: 100 μm.
(B) Representative images and quantification of PlexinD1 IHC staining in primary PCa categorized by low Gleason score (GS) (GS 4–6) and high GS (GS 7–10) from the same TMA as in (A). Scale bars: 100 μm. (C) Comparisons of *PLXND1* mRNA levels in PCa vs. normal prostates from GSE3325 and GSE3933. (D) Representative images and quantification of PlexinD1 IHC staining in PCa tissue from patients post- vs. pre-hormone therapy from the NYU TMA. Scale bars: 100 μm. (E) Comparisons of *PLXND1* mRNA levels in recurrent vs. non-recurrent and hormone-refractory vs. -sensitive PCa from GSE21032 and GSE6099, respectively. (F) Representative images and quantification of PlexinD1 IHC staining in bone metastatic vs. primary PCa. Scale bars: 100 μm. (G) Comparisons of *PLXND1* mRNA levels in metastatic vs. primary PCa from GSE21032 and GSE35988. Data information: In (A–G), data are presented as mean ± SEM with the *n* of each group indicating the number of patient tumor samples included for comparisons. *P* values were determined by unpaired two-tailed Student's *t*-test. *P < 0.05, **P < 0.01. Exact *P* values are listed in Appendix Table S4. Source data are available online for this figure.

GSE35988 (Grasso et al, 2012) (Data ref: Grasso et al, 2012)] (Fig. 2G; Appendix Fig. S1B). Together, these results demonstrated that PlexinD1 upregulation is correlated with unfavorable clinical outcomes in PCa patients.

## PlexinD1 is suppressed by androgen signaling

Since an inverse relationship between PlexinD1 expression and AR signaling was observed in C4-2B^ENZR versus LNCaP cells, we examined the role of AR signaling in the regulation of PlexinD1 using three androgen-responsive AR+ PCa cell lines, LNCaP,

LAPC4 and VCaP. Growing cells in media supplemented with charcoal-stripped serum (CSS) for androgen depletion resulted in marked increases in *PLXND1*, *SEMA3C* and *SEMA3E* mRNA expression associated with downregulated *KLK3* (encoding PSA) and upregulated *NCAM1* (encoding CD56, a NE marker) mRNA levels, which was reversed by treatment with the synthetic androgen R1881 (Fig. 3A). Similarly, we demonstrated that treatment with R1881 reduced, while addition of ENZ increased, PlexinD1 protein levels, accompanied by opposite changes in AR and PSA protein expression, in a time-dependent manner in both LNCaP and LAPC4 cells (Fig. 3B,C).

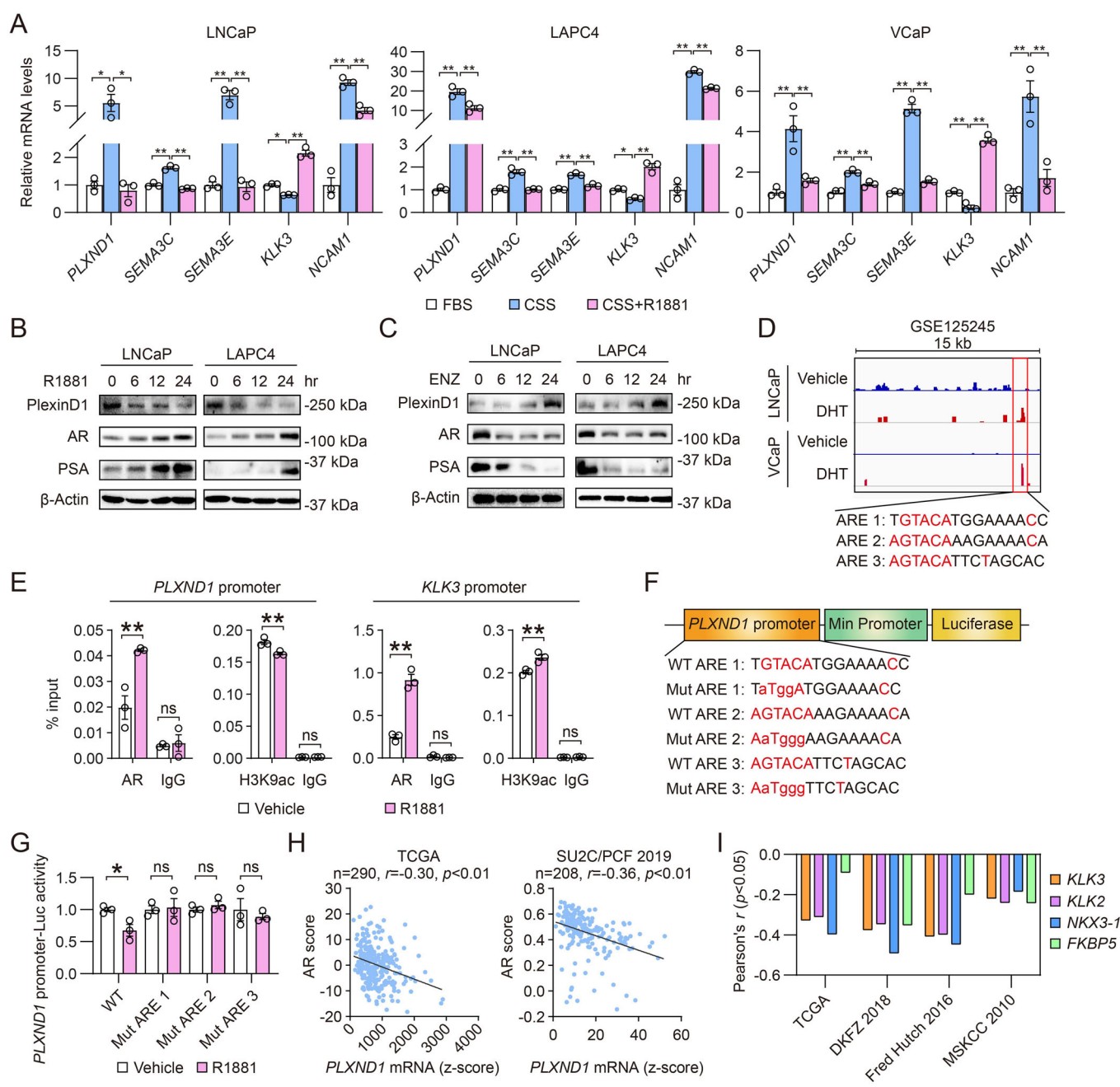

**Figure 3. Androgen signaling negatively regulates PlexinD1 expression in PCa.**

(**A**) qPCR of *PLXND1*, *SEMA3C*, *SEMA3E*, *KLK3*, and *NCAM1* in LNCaP, LAPC4, and VCaP cells upon culture under CSS condition for 24 h followed by R1881 stimulation (10 nM, 24 h) ($n = 3$ biological replicates). (**B, C**) Western blot of PlexinD1, AR, and PSA in LNCaP and LAPC4 cells upon treatment with 10 nM R1881 (**B**) or 10 µM ENZ (**C**) for the indicated times. (**D**) Genomic browser representation of AR binding at *PLXND1* promoter encompassing three AREs, with the nucleotides identical to the canonical ARE highlighted in red, by interrogating AR ChIP-seq dataset GSE125245. (**E**) ChIP-qPCR of AR and H3K9ac occupancy at an ARE-centric *PLXND1* promoter sequence as well as an AR-bound *KLK3* promoter region upon R1881 stimulation (10 nM, 6 h) in LNCaP cells. Data represent the percent of input ($n = 3$ technical replicates). (**F**) Schematic diagrams of WT and mutated forms of individual AREs in *PLXND1* ARE-Luc constructs. (**G**) Determination of WT and mutated *PLXND1* ARE-Luc activities upon R1881 stimulation (10 nM, 6 h) in LNCaP cells ($n = 3$ biological replicates). (**H**) Pearson correlation analysis of *PLXND1* mRNA expression with AR score in the indicated datasets from cBioPortal. (**I**) Pearson correlation analysis of mRNA co-expression between *PLXND1* vs. different AR target genes in the indicated datasets from cBioPortal. Data information: In (**A**, **E**, **G**), data are presented as mean ± SEM. In (**A**), *P* values were determined by one-way ANOVA with Tukey's multiple comparisons test. In (**E**), *P* values were determined by two-way ANOVA with Tukey's multiple comparisons test. In (**G**), *P* values were determined by unpaired two-tailed Student's *t*-test. In (**H**), *P* values were determined by Pearson's correlation coefficient *t*-test. \**P* < 0.05, \*\**P* < 0.01; ns, not significant. Exact *P* values are listed in Appendix Table S4. Source data are available online for this figure.

Given that *PLXND1* expression is repressed by AR signaling, we explored whether AR binds to the *PLXND1* gene locus for direct transcriptional regulation. Interrogating existing data of AR chromatin immunoprecipitation (ChIP) sequencing (ChIP-seq) in LNCaP and VCaP cells (GSE125245), we identified an AR-bound region in *PLXND1* promoter sequence. Further, we identified three distinct consensus androgen response element (ARE1-3, −3297 to −3311, −3321 to −3335, and −3368 to −3382, respectively, with *PLXND1* transcription start site set as +1) half-sites within the AR-bound region, which have a high homology with the canonical ARE sequence GGT/AACAnnnTGTTCT (Roche et al, 1992) (Fig. 3D). To confirm AR occupancy on *PLXND1* promoter, we conducted ChIP-quantitative PCR (ChIP-qPCR) assays and revealed significant enrichment for AR binding at the sequence encompassing all three AREs upon R1881 stimulation, which was substantially lower in the absence of R1881. A similar observation was made in a known ARE-bound *KLK3* promoter region as a positive control. To assess the transcriptional status of *PLXND1* upon R1881 stimulation, we demonstrated in parallel a reduction of H3K9ac enrichment, a mark of active promoter state, in the AR-bound region of *PLXND1* promoter, confirming its transcriptionally repressed state. Conversely, increased enrichment of H3K9ac marks on *KLK3* promoter was observed under R1881 treatment, indicating its transcriptionally active state (Fig. 3E). To further determine whether the identified AREs are functional, we inserted the AREs-centric sequence upstream of the minimal promoter-driven luciferase (Luc) gene to construct a *PLXND1* promoter-Luc reporter (Fig. 3F). Compared to wild-type (WT) *PLXND1* promoter-Luc repressed by R1881, mutations of select nucleotides in each individual ARE rendered the resulting reporters unresponsive to R1881 in LNCaP cells (Fig. 3G).

To seek a clinical relationship of PlexinD1 with AR, we found a negative correlation between *PLXND1* mRNA and AR score defined by assessment of 30 AR target genes (Abida et al, 2019) in both TCGA and SU2C/PCF 2019 (Abida et al, 2019) (Data set: Abida et al, 2019) datasets, representing primary PCa and CRPC, respectively, on the cBioPortal for Cancer Genomics (Fig. 3H), which was corroborated by negative mRNA expression correlations of *PLXND1* with several classical AR target genes (*KLK3*, *KLK2*, *NKX3-1*, and *FKBP5*) in four independent cohorts [TCGA, DKFZ 2018 (Gerhauser et al, 2018) (Data ref: Gerhauser et al, 2018), Fred Hutch 2016 (Kumar et al, 2016) (Data ref: Kumar et al, 2016), and MSKCC 2010 (GSE21032)] from cBioPortal (Fig. 3I). Collectively, these results suggested that *PLXND1* is transcriptionally suppressed by AR via direct AR binding to *PLXND1* promoter.

Despite the demonstrated AR transcriptional repression of *PLXND1*, it is intriguing to note that PlexinD1 protein is most highly expressed in AR+ 22Rv1 cells among all the PCa cell lines examined. The overwhelmingly high expression of PlexinD1 protein in 22Rv1 cells was also confirmed in a recent study (Chen et al, 2024). This could be explained in part by the loss of canonical AR signaling as evidenced by very low levels of PSA expression and enrichment of NE markers in 22Rv1 cells as reported recently (Thaper et al, 2022). On the other hand, we speculated that additional transcriptional regulators may outcompete AR repression of PlexinD1 to upregulate PlexinD1 expression particularly in 22Rv1 cells given the shared expression pattern for *PLXND1* mRNA and protein across different PCa cell lines. To test this idea, we interrogated the Cancer Cell Line Encyclopedia (CCLE), a database including gene expression data for human cancer cell lines, and identified *BRN2*, a neural transcription factor known as an AR-suppressed driver of NE differentiation (Bishop et al, 2017), expressed with a notably higher mRNA level in 22Rv1 cells than in other common PCa cell lines (e.g., LNCaP, PC-3, and DU145), which was corroborated by a recent study (Thaper et al, 2022). We also demonstrated substantially higher expression levels of BRN2 protein in 22Rv1 and C4-2B^ENZR than in other PCa cell lines (Appendix Fig. S1C), which is in accord with the PlexinD1 protein expression pattern in these cell lines. Further, we showed that BRN2 knockdown reduced *PLXND1* mRNA levels in 22Rv1 cells, while overexpression of BRN2 enhanced *PLXND1* mRNA levels in LNCaP cells, compared to respective controls (Appendix Fig. S1D). These data suggested BRN2 as a potential transcriptional regulator likely to contribute to the particular abundance of PlexinD1 in 22Rv1 cells.

## PlexinD1 is necessary to maintain CRPC aggressive behaviors in vitro and in vivo

To evaluate the function of PlexinD1 in regulating various cell behaviors of CRPC cells, we stably knocked down PlexinD1 expression in multiple CRPC cell lines. PlexinD1 silencing by lentiviral infection with two independent shRNAs caused considerable reductions in the proliferation of C4-2B^ENZR and 22Rv1 cells and impaired the ability of these cells to form colonies in soft agar compared with controls expressing a scrambled control shRNA (Fig. 4A–C). To determine the effects of PlexinD1 ligands on CRPC cell proliferation, we treated control and PlexinD1-knockdown cells with recombinant Sema3C or Sema3E proteins in serum-free medium. We found that both Sema3 ligands induced the proliferation of control C4-2B^ENZR and 22Rv1 cells, with a stronger effect by Sema3E than Sema3C, while these inductions were reversed upon PlexinD1 knockdown (Appendix Fig. S2). Of note, Sema3C but not Sema3E still caused modest but statistically significant increases in the proliferation of both cell lines in the absence of PlexinD1, which could be likely due to activation of other Plexins such as PlexinB1 known to mediate Sema3C's pro-proliferative effect in CRPC cells (Peacock et al, 2018). These data suggested that both Sema3C and Sema3E as PlexinD1 ligands could utilize PlexinD1 to promote CRPC proliferation, with more reliance on PlexinD1 for Sema3E. To determine the effect of PlexinD1 silencing in non-tumorigenic and non-CRPC cells, we demonstrated that siRNA-mediated knockdown of PlexinD1, with siRNA target sequences validated in C4-2B^ENZR and human normal endothelial HUVEC cells (Appendix Fig. S3A), did not interfere with the proliferation of representative non-tumorigenic and non-CRPC cells, including HUVEC, RWPE-1, and LNCaP, which was paralleled with decreased proliferation of C4-2B^ENZR cells (Appendix Fig. S3B). The anti-proliferative effect of PlexinD1 silencing was further seen in other CRPC cell lines, PC-3, DU145, and LASCPC-01 (Appendix Fig. S4A–C). We also analyzed PlexinD1 target genes in PlexinD1-knockdown versus control C4-2B^ENZR cells using RNA-seq. GSEA revealed that "MYC Targets", "Oxidative Phosphorylation", "Fatty Acid Metabolism", and several cell-cycle-related signatures from the Hallmark gene set were downregulated in C4-2B^ENZR cells upon PlexinD1 depletion compared with controls (Fig. 4D). Conversely, we showed that stable overexpression of PlexinD1 in PlexinD1-low non-CRPC LNCaP cells promoted the proliferation and anchorage-independent colony formation of LNCaP cells compared with controls expressing an empty vector (Fig. 4A–C).

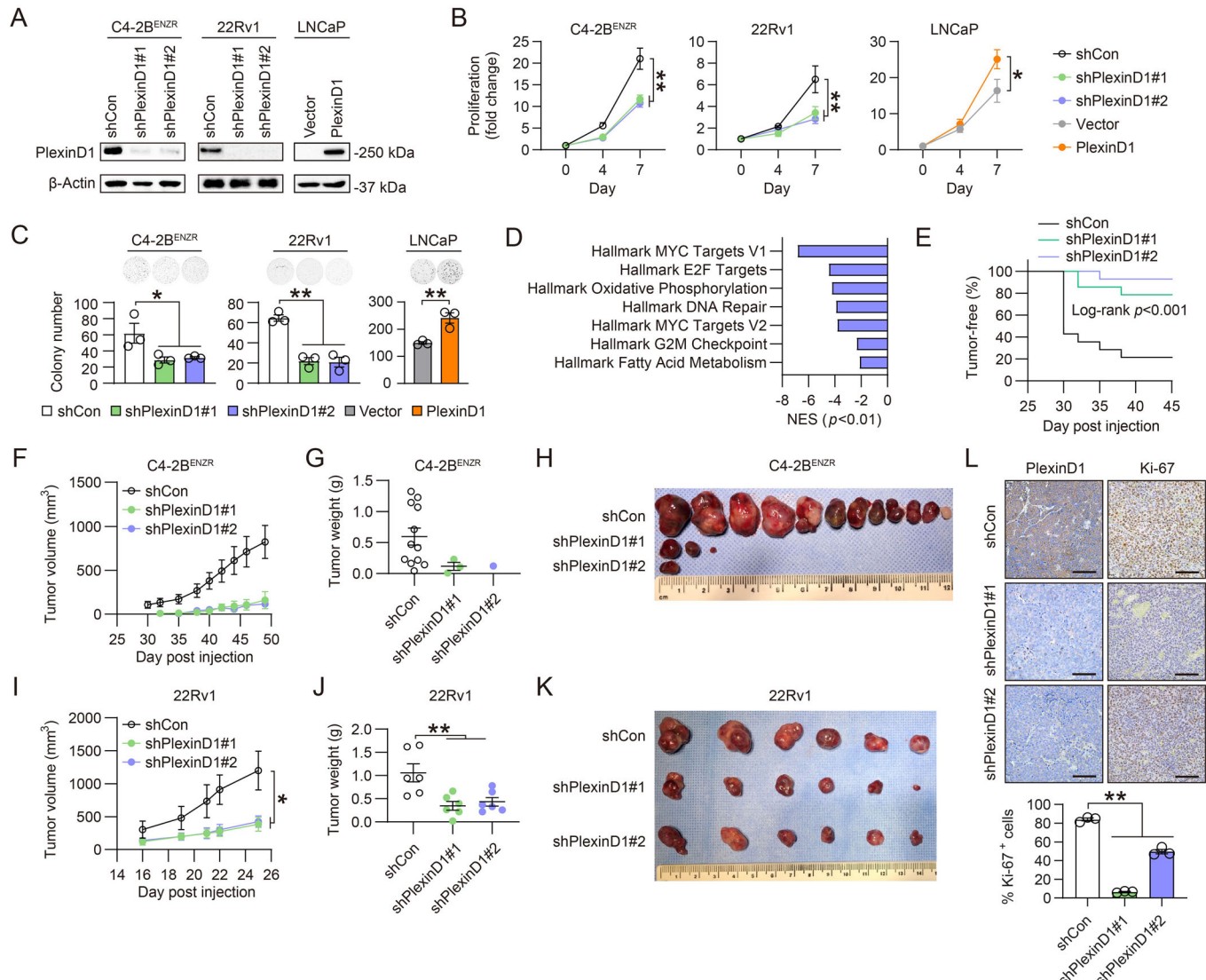

**Figure 4. PlexinD1 promotes PCa growth in vitro and in vivo.**

(A) Western blot of PlexinD1 in control and PlexinD1-manipulated PCa cells. (B) Cell proliferation assays of control and PlexinD1-manipulated PCa cells. Data represent the fold changes of cell proliferation during a 7-day observation period ($n = 4$ biological replicates). Fold change on the day of cell seeding (day 0) in each group was set as 1. (C) Representative images and quantification of colonies formed by control and PlexinD1-manipulated PCa cells ($n = 3$ biological replicates). (D) GSEA of Hallmark gene sets most negatively enriched in PlexinD1-knockdown vs. control C4-2B^ENZR cells. (E) Kaplan-Meier tumor-free curves of mice inoculated with control and PlexinD1-knockdown C4-2B^ENZR cells ($n = 14$ tumor inoculations). (F–H) Tumor growth curves of mice bearing C4-2B^ENZR tumors (F), endpoint tumor weights (G), and anatomic tumor images (H) for the experiment described in (E) ($n = 12$, 3 and 1 tumor for shCon, shPlexinD1#1 and shPlexinD1#2 groups, respectively). (I–K) Tumor growth curves (I), endpoint tumor weights (J), and anatomic tumor images (K) of mice inoculated with control and PlexinD1-knockdown 22Rv1 tumors ($n = 6$ tumors). (L) Representative IHC staining of PlexinD1 and Ki-67 and quantification of % of Ki-67+ cells in 22Rv1 tumor samples ($n = 3$ tumors). Scale bars: 100 μm. Data information: In (B, C, F, G, I, J, L), data are presented as mean ± SEM. In (B, C), $P$ values were determined by one-way ANOVA with Dunnett's multiple comparisons test (for C4-2B^ENZR and 22Rv1) and unpaired two-tailed Student's $t$-test (for LNCaP). In (D), $P$ values were determined by permutation test. In (E), $P$ value was determined by log-rank test. In (F, G, I, J, L), $P$ values were determined by one-way ANOVA with Dunnett's multiple comparisons test. *$P < 0.05$, **$P < 0.01$. Exact $P$ values are listed in Appendix Table S4. Source data are available online for this figure.

To analyze the consequence of PlexinD1 silencing in vivo, we subcutaneously engrafted C4-2B^ENZR cells expressing either a control or a *PLXND1* shRNA into immunodeficient male mice and monitored tumor growth. Mice growing PlexinD1-knockdown cells formed significantly fewer tumors (shPlexinD1#1: 3 tumors/14 injection sites; shPlexinD1#2: 1 tumor/14 injection sites) in contrast to control mice (shCon: 12 tumors/14 injection sites) (Fig. 4E). The

tumors developed from PlexinD1-knockdown cells also propagated more slowly, with lower tumor weight at the endpoint, compared to control tumors (Fig. 4F–H). Using 22Rv1 cells that have more robust tumorigenicity than C4-2B^ENZR cells, we demonstrated that PlexinD1 silencing also resulted in a significant reduction in tumor growth rate and tumor weight compared with controls (Fig. 4I–K). Characterizing 22Rv1 tumor samples by IHC, we found lower

PlexinD1 protein expression in PlexinD1-knockdown relative to control tumors, indicating effective and sustainable PlexinD1 knockdown in vivo, as well as a 67% drop of Ki-67+ cells for decreased mitotic index in PlexinD1-knockdown tumors compared with controls (Fig. 4L). We also observed similar growth suppression of CRPC tumors grown from PC-3 and LASCPC-01 cells upon PlexinD1 depletion (Appendix Fig. S4D–H). Conversely, overexpression of PlexinD1 in LNCaP cells led to more tumors formed (Vector: 6 tumors/8 injection sites; PlexinD1: 8 tumors/8 injection sites), faster tumor growth, higher tumor weight, and increased percentages of tumor-expressing Ki-67+ cells in xenografted mice compared with controls (Appendix Figs. S5A–E).

Since gaining the migratory ability and invading surrounding tissue for initiating metastasis are hallmarks of CRPC progression (Saad and Hotte, 2010), we next evaluated PlexinD1's role in regulating PCa cell migration and invasion. Using transwell assays, we demonstrated that both C4-2B$^{ENZR}$ and 22Rv1 cells lacking PlexinD1 exhibited decreased migration and invasion compared with controls (Fig. 5A). These findings were also confirmed in two highly metastatic CRPC cell lines, PC-3 and ARCaP$_M$, by transwell assays (Wu et al, 2017) (Appendix Fig. S6A). Wound-healing assays further revealed less cell migration of PlexinD1-knockdown compared with control PC-3 cells (Appendix Fig. S6B). Conversely, PlexinD1 overexpression led to increased migration of LNCaP cells by transwell assays. Using a 3D spheroid invasion assay, we also showed increased invasion of PlexinD1-overexpressing LNCaP cells as evidenced by more cells invading through the edge of the Matrigel drop compared with controls (Fig. 5B). Then, we examined several well-known markers of EMT implicated for providing cancer cells with the ability to migrate and invade (Zeisberg and Neilson, 2009). We demonstrated increases in E-cadherin (an epithelial marker) expression and decreases in N-cadherin (a mesenchymal marker) and Twist1 (an EMT-driving master regulator) expression in PlexinD1-knockdown C4-2B$^{ENZR}$ and 22Rv1 cells compared with controls, which was corroborated by similar observations in PC-3 cells upon PlexinD1 depletion (Fig. 5C; Appendix Fig. S6C). Consistently, we found opposite changes in the expression levels of these EMT markers in LNCaP cells upon PlexinD1 overexpression compared with control cells (Fig. 5C). GSEA also revealed a Hallmark EMT gene signature positively enriched in PlexinD1-overexpressing relative to control LNCaP cells (Fig. 5D).

Having demonstrated PlexinD1's pro-migratory and pro-invasive role in vitro, we sought to find out whether PlexinD1 could influence CRPC metastasis in vivo. To this end, we injected Luc and red fluorescence protein (RFP) dually tagged 22Rv1 cells expressing either a control or a *PLXND1* shRNA intracardially into immunodeficient male mice to allow rapid development of tumor metastasis and tracked tumor burden by bioluminescence imaging. We observed a significant reduction of metastatic tumor growth in mice inoculated with PlexinD1-knockdown cells compared to mice inoculated with control cells (Fig. 5E,F). Necropsy and ex vivo fluorescence imaging analysis further revealed that PlexinD1 silencing resulted in significantly fewer tumor cells grown at select organ and tissue sites, including the liver, adrenal glands, and bone, compared to controls (Fig. 5G). Examining liver and adrenal metastases by IHC demonstrated lower PlexinD1 expression and Ki-67+ cell percentages in PlexinD1-knockdown tumors relative to controls (Fig. 5H). Conversely, overexpression of PlexinD1 in Luc-tagged LNCaP cells with inherently weak metastatic

potential resulted in higher metastasis incidence and greater tumor burden when inoculated intracardially into mice compared with controls (Appendix Fig. S7A,B). At the endpoint, necropsy and ex vivo bioluminescence imaging analysis confirmed more prevalent distant metastases, primarily found in the liver and jawbone, caused by PlexinD1-overexpressing tumors cells relative to controls (Appendix Fig. S7C,D). Together, these results indicate that PlexinD1 plays a pivotal role in driving the aggressive phenotype of CRPC.

## PlexinD1 promotes cellular plasticity, stemness, and NE differentiation in PCa

Blockade of AR signaling in PCa is known to induce a stem-like and multilineage cellular state, which facilitates the development of therapy resistance (Beltran et al, 2019; Deng et al, 2022). To determine whether PlexinD1 regulates lineage plasticity, we first performed GSEA with the transcriptome profiling data from PlexinD1-overexpressing versus control LNCaP cells. GSEA revealed upregulation of gene signatures related to stem cell and basal epithelial phenotypes, accompanied by downregulation of signatures corresponding to a luminal epithelial phenotype, when PlexinD1 was overexpressed (Fig. 6A). To examine PlexinD1's effect on stemness, we showed that PlexinD1 depletion limited tumorsphere establishment as evidenced by both fewer and smaller spheres in C4-2B$^{ENZR}$ and 22Rv1 cells compared with controls. Conversely, forced expression of PlexinD1 enhanced LNCaP tumorsphere formation (Fig. 6B). Given the critical function of aldehyde dehydrogenase (ALDH) in promoting cancer stemness (Le Magnen et al, 2013), we also performed an ALDH assay and demonstrated reductions in ALDH activity in PlexinD1-knockdown C4-2B$^{ENZR}$ and 22Rv1 cells but increased ALDH activity in PlexinD1-overexpressing LNCaP cells compared with respective controls (Fig. 6C). Then, we examined protein expression levels of markers for stemness and multiple cell lineages. Consistent with the above findings, PlexinD1 depletion resulted in decreased expression of stemness (SOX2, NANOG, OCT3/4, and LIN28) and basal cell (CK5 and p63) markers along with heightened levels of luminal cell markers (AR, PSA, and CK8). The opposite expression changes of these markers were seen in PlexinD1-overexpressing versus control LNCaP cells (Fig. 6D).

Intriguingly, we found that PlexinD1 depletion reduced the expression of several canonical NE markers (SYP, CD56, NSE, and CHGA) in C4-2B$^{ENZR}$ and 22Rv1 cells, where C4-2B$^{ENZR}$ cells exhibit treatment-induced NE features as we reported recently (Bland et al, 2021) and 22Rv1 cells express dual AR/NE positivity (Huss et al, 2004; Sramkoski et al, 1999). This was paralleled by enhanced expression of these NE markers upon PlexinD1 overexpression in LNCaP cells (Fig. 6D). Concordantly, knockdown of PlexinD1 in two additional NE-like PCa cell lines, LASCPC-01 and NE1.8, also caused significant decreases in NE marker expression (Appendix Fig. S8A,B). Of note, NE1.8 cells grew extremely slowly upon PlexinD1 silencing and survived PlexinD1 knockdown with only one shRNA but not the other, suggesting a requisite role of PlexinD1 in maintaining the viability of these NE-like cells. Complementing the approach of NE marker assessment, we also showed that PlexinD1 silencing resulted in a dedifferentiated cell morphology, a measure of a NE-like phenotype (Bland et al, 2021), in both C4-2B$^{ENZR}$ and NE1.8 cells, with decreases in per-cell number of neurites and average neurite length compared with

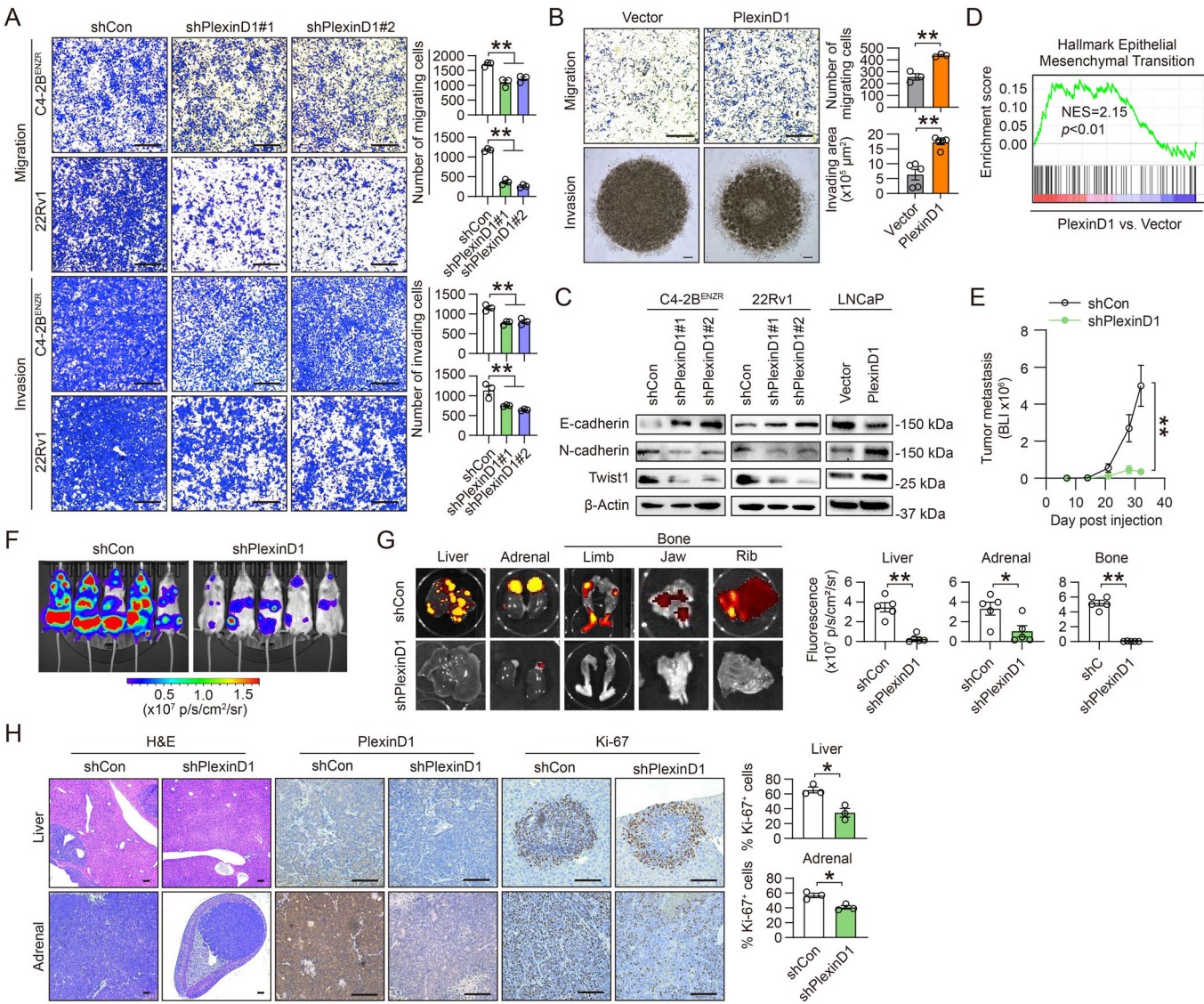

**Figure 5. PlexinD1 drives EMT, migration, invasion, and metastasis in PCa.**

(A) Representative images and quantification of transwell-based cell migration and invasion by control and PlexinD1-knockdown C4-2B[ENZR] and 22Rv1 cells (n = 3 biological replicates). Scale bars: 400 μm. (B) Representative images and quantitation of transwell-based cell migration and 3D spheroid cell invasion by control and PlexinD1-overexpressing LNCaP cells (n = 3 biological replicates). Scale bars: 400 μm. (C) Western blot of select EMT markers in control and PlexinD1-manipulated PCa cells. (D) GSEA plot of a Hallmark EMT gene set enriched in PlexinD1-overexpressing vs. control LNCaP cells. (E, F) Bioluminescence (BLI)-based growth curves (E) and endpoint BLI images (F) of Luc/RFP dually tagged control and PlexinD1-knockdown 22Rv1 tumors developed in an intracardiac xenograft model (n = 5 mice). (G) Representative fluorescence images and quantification of tumor metastasis developed at indicated organ sites by control and PlexinD1-knockdown 22Rv1 cells (n = 5 metastatic tumors). (H) Representative images of H&E and IHC staining of PlexinD1 and Ki-67 and quantification of % of Ki-67+ cells in control and PlexinD1-knockdown 22Rv1 tumors grown in mouse liver and adrenal glands (n = 3 metastatic tumors). Scale bars: 100 μm. Data information: In (A, B, E, G, H), data are presented as mean ± SEM. In (A), P values were determined by one-way ANOVA with Dunnett's multiple comparisons test. In (B, E, G, H), P values were determined by unpaired two-tailed Student's t-test. In (D), P values were determined by permutation test. *P < 0.05, **P < 0.01. Exact P values are listed in Appendix Table S4. Source data are available online for this figure.

controls (Fig. 6E; Appendix Fig. S8C). Since acquisition of a NE cell lineage has emerged as a key mechanism for PCa cells to escape ADT (Davies et al, 2018), we further sought to determine whether PlexinD1 could modulate ARSI response. We demonstrated that PlexinD1-overexpressing LNCaP cells displayed androgen-independent cell proliferation with comparable relative changes of proliferation rate under ENZ versus vehicle treatment over a 5-day observation period, in contrast to control cells that had a

notable reduction in relative changes of proliferation rate when exposed to ENZ (Fig. 6F). These results in aggregate suggested that PlexinD1 is essential for induction and maintenance of a NE phenotype.

To seek a clinical association of PlexinD1 with NE features, we interrogated publicly available PCa datasets and demonstrated uniform upregulation of *PLXND1* mRNA levels in patient tumor tissue [Trento/Cornell/Broad 2016 (Beltran et al, 2016) (Data ref:

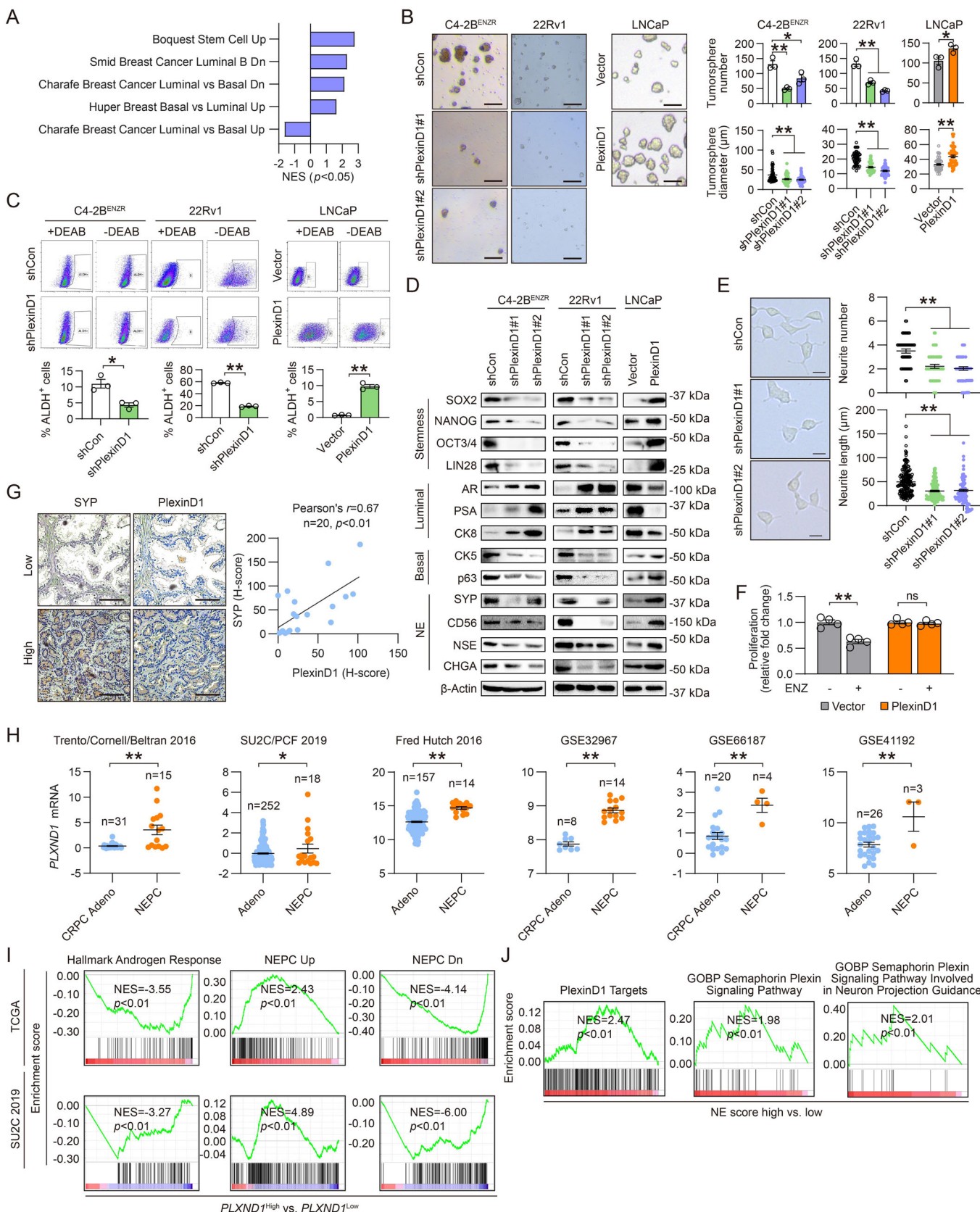

**Figure 6.  PlexinD1 promotes cellular plasticity, stemness, and NE differentiation in PCa.**

(A) GSEA of cell stemness and lineage-related gene sets enriched in PlexinD1-overexpressing vs. control LNCaP cells. (B) Representative images and quantification of number and size of tumorspheres formed by control or PlexinD1-manipulated PCa cells ($n = 3$ biological replicates). Scale bars: 100 μm. (C) Flow cytometric analysis of % of ALDH+ cells in control and PlexinD1-manipulated PCa cells, with DEAB-treated groups as negative controls ($n = 3$ biological replicates). (D) Western blot of stemness, luminal, basal, and NE markers as indicated in control and PlexinD1-manipulated PCa cells. (E) Representative images of control and PlexinD1-knockdown C4-2B$^{ENZR}$ cell morphology and quantification of per-cell number of neurites and neurite length in each group ($n = 50$ cells per group). A representative of 3 independent experiments is shown. Scale bars: 20 μm. (F) Cell proliferation assays of control and PlexinD1-overexpressing LNCaP cells upon ENZ treatment (20 μM, 5 days). Data represent the fold changes of cell proliferation on day 5 relative to cell seeding day (day 0) ($n = 4$ biological replicates), with fold changes in non-treated groups set as 1 for normalization of paired treated groups. (G) Representative PlexinD1 and SYP IHC staining in serial sections of tumor tissue from a CRPC patient cohort and corresponding Pearson correlation analysis of protein co-expression. Scale bars: 100 μm. (H) Comparisons of *PLXND1* mRNA levels in NEPC vs. Adeno or CRPC Adeno patient samples from multiple PCa clinical datasets as indicated. The $n$ of each group indicates the number of patient tumor samples included for comparisons. (I) GSEA plots of Hallmark Androgen Response and NEPC gene sets enriched in *PLXND1*-high vs. -low PCa patient samples from TCGA and SU2C/PCF 2019 cohorts in cBioPortal. (J) GSEA plots of PlexinD1 targets and semaphorin/plexin signaling-related gene sets enriched in NE score-high vs. -low cells of a CRPC patient tumor as in the scRNA-seq dataset GSE137829. Data information: In (B, C, E, F, H), data are represented as mean ± SEM. In (A, I, J), *P* values were determined by permutation test. In (B, E), *P* values were determined by one-way ANOVA with Dunnett's multiple comparisons test [for C4-2B$^{ENZR}$ and 22Rv1 in (B)] and unpaired two-tailed Student's *t*-test [for LNCaP in (B)]. In (C, H), *P* values were determined by unpaired two-tailed Student's *t*-test. In (F), *P* values were determined by one-way ANOVA with Tukey's multiple comparisons test. *$P < 0.05$, **$P < 0.01$; ns, not significant. Exact *P* values are listed in Appendix Table S4. Source data are available online for this figure.

Beltran et al, 2016), SU2C/PCF 2019, and Fred Hutch 2016 from cBioPortal] as well as patient-derived xenografts (PDXs) [MDA series: GSE32967 (Tzelepi et al, 2012) (Data ref: Tzelepi et al, 2012); LuCaP series: GSE66187 (Zhang et al, 2015) (Data ref: Zhang et al, 2015); and LTL series: GSE41192 (Lin et al, 2015) (Data ref: Lin et al, 2015)] of NE PCa (NEPC) compared with PCa and CRPC adenocarcinoma (Adeno) (Fig. 6H). GSEA also revealed a down-regulated "Hallmark Androgen Response" gene set and a strong correlation of genes differentially expressed in human NEPC in *PLXND1*-high relative to *PLXND1*-low PCa patient tumor samples from both TCGA and SU2C/PCF 2019 cohorts (Fig. 6I). Further, we performed transcriptomic analysis of an existing human CRPC single-cell RNA-seq (scRNA-seq) dataset (GSE137829), focusing specifically on patient #2 in this cohort whose epithelial tumor cells had an obvious NE differentiated phenotype (Dong et al, 2020) (Data ref: Dong et al, 2020). GSEA revealed a positively enriched "PlexinD1 Targets" gene signature, along with two GO gene sets related to semaphorin/plexin signaling, in epithelial cancer cells with high NE scores based on a set of 70 NEPC reference genes (Beltran et al, 2016) compared with those scored low on NE gene expression (Fig. 6J). The PlexinD1 targets were defined as 372 differentially expressed genes in PlexinD1-overexpressing versus control LNCaP cells. Together, these results suggested a plausible role for PlexinD1 in promoting cellular plasticity associated with stem-like and NE phenotypes.

## PlexinD1 activates ErbB3 and cMet to confer CRPC growth and cellular plasticity

Next, we sought to understand the molecular mechanism by which PlexinD1 confers growth advantages and cellular plasticity to CRPC cells. Plexins are known to activate multiple receptor tyrosine kinases (RTKs) on the cell surface to trigger downstream mitogenic signaling pathways to fuel cancer cells (Peacock et al, 2018; Toledano and Neufeld, 2023), which led us to speculate that PlexinD1 may utilize the same mechanism to provide growth and survival stimuli to CRPC cells. To test this hypothesis, we first performed GSEA demonstrating down-regulation of two RTK-related GO gene signatures, "Transmembrane Receptor Protein Kinase Activity" and "Transmembrane Receptor Protein Tyrosine Kinase Signaling Pathway" in PlexinD1-knockdown versus control 22Rv1 cells, which coincided with upregulation of these

two gene sets in *PLXND1*-high versus *PLXND1*-low PCa patient tumors from TCGA cohort (Fig. 7A). To search for potential RTKs that partner with PlexinD1 in CRPC cells, we carried out an unbiased proteomic screen using a RTK phosphorylation antibody array to detect relative phosphoprotein levels of 71 different RTKs in PlexinD1-silenced versus control C4-2B$^{ENZR}$ cells. We identified 21 RTKs downregulated upon PlexinD1 depletion and selected the top 10 RTKs with the most reductions in phosphorylation for validation by Western blots (Appendix Table S2). Our data revealed ErbB3 as the only RTK with consistent decreases of protein phosphorylation in both C4-2B$^{ENZR}$ and 22Rv1 cells subject to two different *PLXND1* shRNAs (Fig. 7B,C). ErbB3, also known as HER3, is a member of the HER/ErbB family of RTKs and was recently shown to promote ARSI resistance (Hynes and MacDonald, 2009; Zhang et al, 2020). ErbB3 has limited intrinsic kinase activity, and upon binding to neuregulin (NRG) ligands, it often uses ErbB2 as the preferred heterodimer partner for dimerization and activation of downstream signaling (Shi et al, 2010; Sierke et al, 1997; Sliwkowski et al, 1994). Other than ErbB3, we also demonstrated reduced ErbB2 protein phosphoprotein levels in PlexinD1-silenced C4-2B$^{ENZR}$ and 22Rv1 cells compared with controls. Using a candidate approach, we found that PlexinD1 knockdown also remarkably decreased cMet protein phosphorylation in both C4-2B$^{ENZR}$ and 22Rv1 cells. cMet, a RTK activated by its ligand, hepatocyte growth factor (HGF), was shown to be induced by castration and AR inhibition and able to promote resistance to ADT in PCa (Cannistraci et al, 2017; Verras et al, 2007). In addition, we demonstrated reductions in ERK and AKT protein phosphorylation, both often acting downstream of RTKs including ErbB3/ErbB2 and cMet (Arteaga and Engelman, 2014; Zhang et al, 2018), upon PlexinD1 depletion in C4-2B$^{ENZR}$ and 22Rv1 cells compared with controls (Fig. 7C). In line with PlexinD1's effect, we further showed that siRNA-mediated knockdown of Sema3C or Sema3E reduced, while addition of recombinant Sema3C or Sema3E proteins enhanced, the phosphoprotein levels of ErbB3, ErbB3, cMet, ERK, and AKT in both C4-2B$^{ENZR}$ and 22Rv1 cells (Appendix Fig. S9A,B). These data in aggregate suggested ErbB3 and cMet as possible RTK mediators of PlexinD1 function in CRPC cells.

To investigate how PlexinD1 activates RTKs, we first measured the expression of principal ligands for ErbB3 and cMet. Intriguingly, our data revealed no significant changes in *NRG1*, *NRG2* (ErbB3 ligands), and *HGF* mRNA levels between PlexinD1-silenced and control C4-2B$^{ENZR}$ and 22Rv1 cells (Fig. 7D), suggesting that PlexinD1 is unlikely

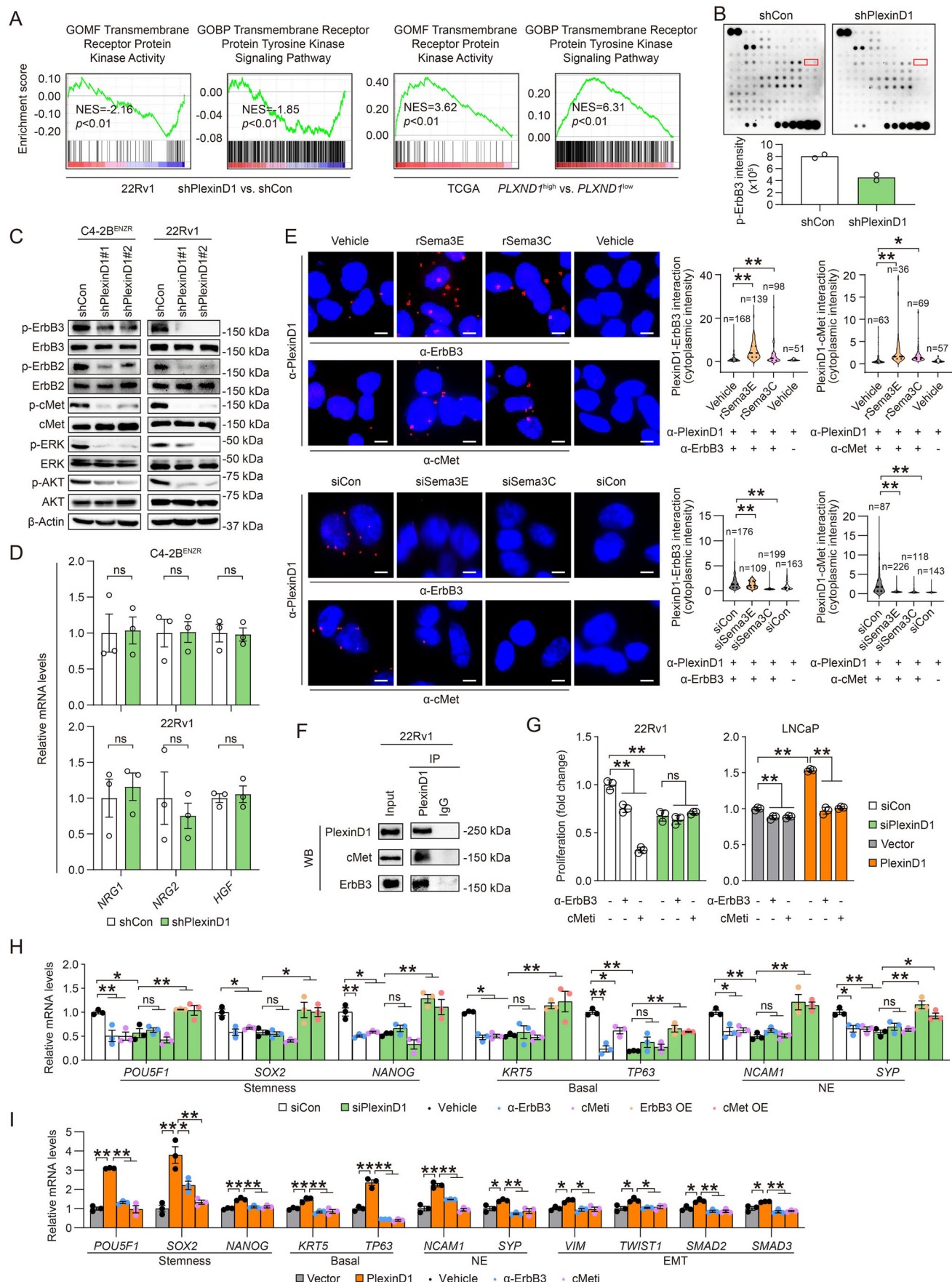

◄ **Figure 7. PlexinD1 activates ErbB3 and cMet in PCa.**

(A) GSEA plots of transmembrane RTK-related gene sets enriched in PlexinD1-knockdown vs. control 22Rv1 cells, and *PLXND1*-high vs. -low PCa patient samples in TCGA cohort. (B) Representative images of a RTK phosphorylation antibody array and quantification of p-ErbB3 levels in control and PlexinD1-knockdown C4-2B$^{ENZR}$ cells ($n = 2$ technical replicates). The raw values from measurement of p-ErbB3 spot intensity after background subtraction are presented. (C) Western blot of p-ErbB3, p-ErbB2, p-cMet, p-ERK, p-AKT, and their total protein forms in control and PlexinD1-knockdown C4-2B$^{ENZR}$ and 22Rv1 cells. (D) qPCR of ErbB3 (*NRG1* and *NRG2*) and cMet (*HGF*) ligand mRNA levels in control and PlexinD1-knockdown C4-2B$^{ENZR}$ and 22Rv1 cells ($n = 3$ biological replicates). (E) Representative PLA images and quantification of PlexinD1-ErbB3/cMet interaction by per-cell cytoplasmic fluorescence intensity in C4-2B$^{ENZR}$ cells treated with recombinant Sema3E/Sema3C proteins (200–500 ng/ml, 4 h) or *SEMA3E*/*SEMA3C* siRNA (10 μM, 48 h). PlexinD1 antibody incubation alone served as negative controls. The $n$ of each group indicates the number of cells examined for quantification. Scale bars: 5 μm. (F) Co-IP assays of PlexinD1-ErbB3/cMet interaction in 22Rv1 cells. IgG was used in the IP step as negative control. Five percent of input was blotted as positive control. (G) Cell proliferation assays of 22Rv1 cells pre-treated with an ErbB3 neutralizing antibody (100 ng/ml, 24 h) or SGX-523 (5 μM, 24 h) and then subjected to *PLXND1* siRNA addition followed by a 5-day observation period, and control and PlexinD1-overexpressing LNCaP cells upon treatment with an ErbB3 neutralizing antibody or SGX-523 during a 5-day observation period. Data represent the fold changes of cell proliferation on day 5 relative to siRNA treatment day (day 1) for 22Rv1 cells or treatment day (day 1) for LNCaP cells ($n = 3$ biological replicates), with fold changes in non-treated groups of control cells set as 1 for normalization of other groups. (H) qPCR of stemness, basal, and NE markers as indicated in 22Rv1 cells receiving an ErbB3 neutralizing antibody (100 ng/ml, 24 h), SGX-523 (5 μM, 24 h), or transient transfection of ErbB3/cMet expression plasmids followed by *PLXND1* siRNA treatment for another 48 h ($n = 3$ biological replicates). (I) qPCR of stemness, basal, NE, and EMT markers as indicated in control and PlexinD1-overexpressing LNCaP cells upon treatment with 100 ng/ml ErbB3 neutralizing antibody or 5 μM SGX-523 for 24 h ($n = 3$ biological replicates). Data information: In (B, D, E, G, H, I), data are presented as mean ± SEM. In (A), *P* values were determined by permutation test. In (B, D), *P* values were determined by unpaired two-tailed Student's *t*-test. In (E), *P* values were determined by one-way ANOVA with Dunnett's multiple comparisons test. In (G, H, I), *P* values were determined by two-way ANOVA with Tukey's multiple comparisons test. *$P < 0.05$, **$P < 0.01$; ns, not significant. Exact *P* values are listed in Appendix Table S4. Source data are available online for this figure.

to induce ErbB3 and cMet in a ligand-dependent manner. Then, we pursued the likelihood of PlexinD1 activating these RTKs via direct protein-protein interaction, which is a proven mechanism for plexins to signal through RTKs in cancer including PCa (Casazza et al, 2010; Peacock et al, 2018). To this end, we performed a PLA and demonstrated the formation of PlexinD1-ErbB3 and PlexinD1-cMet protein complexes in C4-2B$^{ENZR}$ cells as evidenced by fluorescent dots that were absent when a PlexinD1 antibody was incubated alone. Quantitative analysis of the fluorescence restricted to the cytoplasm revealed increases in PlexinD1-RTK interaction ranging from 2–4 folds upon addition of recombinant Sema3E or Sema3C proteins and 28–86% drops of PlexinD1-RTK interaction upon treatment with *SEMA3E* or *SEMA3C* siRNA compared with respective controls (Fig. 7E), suggesting physical interaction between PlexinD1 and ErbB3/cMet with interaction strength dependent on Sema3 ligands. A co-immunoprecipitation (co-IP) assay also confirmed PlexinD1 interaction with ErbB3 and cMet in 22Rv1 cells (Fig. 7F). A direct PlexinD1-ErbB3 interaction was further demonstrated by a co-IP assay with a mix of both recombinant proteins in solution (Appendix Fig. S10).

To determine whether ErbB3/ErbB2 and cMet mediate PlexinD1's role in the regulation of CRPC growth and cellular plasticity, we demonstrated that prior treatment with an ErbB3 blocking antibody, the ErbB2 inhibitor tucatinib, the ErbB2-targeted antibody trastuzumab, or the cMet inhibitor SGX-523, abolished the suppressive effect of PlexinD1 knockdown on the proliferation of 22Rv1 cells compared with controls (Fig. 7G; Appendix Fig. S11A). These data supported a model wherein cells with impaired ErbB3/ErbB2 or cMet activity tend to be insensitive to PlexinD1 intervention for expected phenotypic changes if PlexinD1 exerts effects through ErbB3/ErbB2 or cMet. On the other hand, individual blockade of ErbB3, ErbB2, and cMet all reduced PlexinD1-induced proliferation of LNCaP cells to control levels (Fig. 7G; Appendix Fig. S11B). To assess the contributions of ErbB3/ErbB2 and cMet to PlexinD1's effect on cellular plasticity of CRPC cells, we showed that PlexinD1 knockdown failed to repress the mRNA expression of stemness, basal cell, and NE markers upon pre-treatment with an ErbB3 blocking antibody, tucatinib,

trastuzumab, or SGX-523 in 22Rv1 cells, while enforced expression of ErbB3 or cMet restored these marker expressions as attenuated by PlexinD1 silencing (Fig. 7H; Appendix Fig. S11C). Conversely, interference with ErbB3, ErbB2, or cMet reversed PlexinD1-activated mRNA expression of stemness, basal cell, NE, and EMT markers in LNCaP cells (Fig. 7I; Appendix Fig. S11D). Collectively, these results support the idea that PlexinD1 drives CRPC growth and plasticity through transactivation of ErbB3 and cMet.

## PlexinD1 induces noncanonical Gli1-dictated Hedgehog signaling

Next, we carried out pathway analysis of gene expression profiling data generated in control and PlexinD1-knockdown C4-2B$^{ENZR}$ cells. Our analysis revealed the transcription factor Gli1 as the most significantly downregulated upstream regulator (absolute value of z-score of ≥2 and $p < 0.05$) accounting for the transcriptional changes among those observed in PlexinD1-silenced cells (Fig. 8A). Gli1 is a transcriptional effector at the terminal end of the Hedgehog (Hh) signaling pathway whose aberrant activation is associated with a multitude of cancers including PCa (Eichenmuller et al, 2009; Karhadkar et al, 2004; Kubo et al, 2004; Sheng et al, 2004; Watkins et al, 2003). Our STRING analysis further identified Gli1 as a hub protein integrating direct and indirect interactions with multiple lineage-specific markers and drivers of stemness, EMT, and NE phenotype (Fig. 8B). To determine whether Gli1 is altered by PlexinD1, we examined both Gli1 expression and transcriptional activity. We showed that Gli1 protein levels remained stable upon either overexpression or depletion of PlexinD1 in PCa cells (Fig. 8C). Using a Gli-Luc reporter in which eight copies of Gli-binding sites upstream of the Luc gene drive Luc expression to indicate Gli transcriptional activity, we found decreases in Gli-Luc activity upon PlexinD1 depletion in both C4-2B$^{ENZR}$ and 22Rv1 cells and an increase in Gli-Luc activity when PlexinD1 was overexpressed in LNCaP cells compared with respective controls (Fig. 8D). This was corroborated by prevalent reductions in the mRNA expression of several Gli1 target genes (*PTCH1*, *GLI2*, *HHIP*, *BCL2*, *FOXM1*, *IGFBP6*, *CCND1*, and

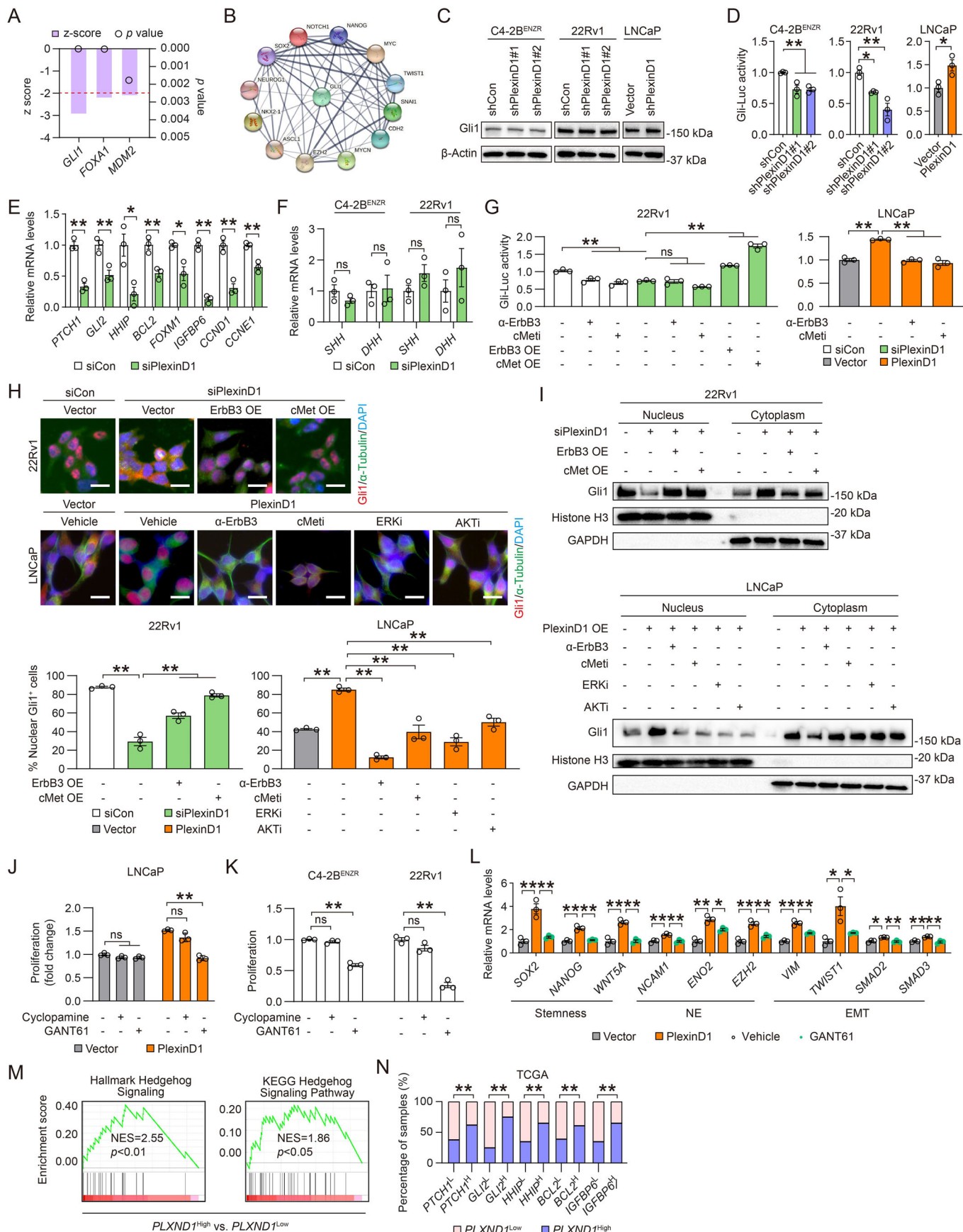

**Figure 8. PlexinD1 induces noncanonical Gli1-dictated Hedgehog signaling in PCa.**

(A) IPA upstream regulator analysis of RNA-seq data from PlexinD1-knockdown vs. control C4-2B^ENZR cells, with significantly downregulated hits shown (absolute value of z-score of ≥2, $p < 0.05$). (B) STRING analysis of Gli1 linked to multiple stemness, NE, and EMT marker and/or driver genes. (C) Western blot of Gli1 in control and PlexinD1-manipulated PCa cells. (D) Determination of Gli-Luc reporter activity in control and PlexinD1-manipulated PCa cells ($n = 3$ biological replicates). (E) qPCR of Gli1 target genes in control and PlexinD1-knockdown C4-2B^ENZR cells ($n = 3$ biologial replicates). (F) qPCR of Hh ligands in control and PlexinD1-knockdown C4-2B^ENZR and 22Rv1 cells ($n = 3$ biological replicates). (G) Determination of Gli-Luc reporter activity in 22Rv1 cells receiving an ErbB3 neutralizing antibody (100 ng/ml, 24 h), SGX-523 (5 μM, 24 h), or transient transfection of ErbB3/cMet expression plasmids followed by *PLXND1* siRNA treatment for another 48 h ($n = 3$ biological replicates), and in control and PlexinD1-overexpressing LNCaP cells upon treatment with 100 ng/ml ErbB3 neutralizing antibody or 5 μM SGX-523 for 24 h ($n = 3$ biological replicates). (H) Representative images of Gli1 (red color) and α-tublin (a cytoplasm marker, green color) co-IF images and quantification of Gli1+ cells in the nuclei of control and PlexinD1-manipulated 22Rv1 and LNCaP cells under the conditions as in (G) ($n = 3$ biological replicates). Scale bars: 10 μm. (I) Western blot of Gli1 in nuclear and cytoplasmic fractions of control and PlexinD1-manipulated 22Rv1 and LNCaP cells under the conditions as in (G). (J) Cell proliferation assays of control and PlexinD1-overexpressing LNCaP cells upon treatment with 10 μM cyclopamine or 5 μM GANT61 for 5 days ($n = 3$ biological replicates). (K) Cell proliferation assays of C4-2B^ENZR and 22Rv1 cells upon treatment with 10 μM cyclopamine or 5 μM GANT61 for 5 days ($n = 3$ biological replicates). (L) qPCR of stemness, NE and EMT markers in control and PlexinD1-overexpressing LNCaP cells under treatment with GANT61 (5 μM, 24 h) ($n = 3$ biological replicates). (M) GSEA plots of two Hh signaling-related gene sets enriched in *PLXND1*-high vs. -low PCa patient samples from TCGA cohort. (N) Chi-square analysis of distribution of select Gli1 target gene mRNA expression (L, low; H, high) in *PLXND1*-low and -high PCa patient samples from TCGA cohort. Data information: In (D-H, J-L), data are presented as mean ± SEM. In (A), P values were determined by Fisher's exact test. In (D-F, K), P values were determined by one-way ANOVA with Dunnett's multiple comparisons test [for C4-2B^ENZR and 22Rv1 in (D)] and unpaired two-tailed Student's *t*-test [for LNCaP in (D)]. In (G, H, J, L), P values were determined by one-way ANOVA with Tukey's multiple comparisons test. In (M), P values were determined by permutation test. In (N), P values were determined by chi-square test. *$P < 0.05$, **$P < 0.01$; ns, not significant. Exact P values are listed in Appendix Table S4. Source data are available online for this figure.

*CCNE1*) in PlexinD1-knockdown C4-2B^ENZR cells compared with controls (Fig. 8E). Congruent with PlexinD1's effect on Gli1 transcriptional activity, we further showed that siRNA-mediated knockdown of Sema3C or Sema3E reduced, while treatment with recombinant Sema3C or Sema3E proteins enhanced, Gli-Luc reporter activity as well as Gli1 target gene expression (Appendix Fig. S12A,B).

Then, we interrogated how PlexinD1 activates Gli1. To this end, we first attempted to discern whether PlexinD1 induces Gli1 through canonical or noncanonical Hh signaling. Since canonical activation of Hh signaling proceeds through binding of Hh ligands to the transmembrane receptor Patched 1 (Ptch1), which relieves the downstream depression of Smoothened (Smo) to facilitate Gli translocation to the nucleus (McMillan and Matsui, 2012), we sought to determine the effect of PlexinD1 on expression of Hh ligands that are often overproduced in PCa for activation of Hh-Gli signaling (Almazan-Moga et al, 2017; Takabatake et al, 2019). Our analysis revealed no significant changes in mRNA levels of Hh ligands including Sonic and Desert Hedgehogs (*SHH* and *DHH*) between PlexinD1-knockdown and control C4-2B^ENZR and 22Rv1 cells, with Indian Hedgehog (*IHH*) undetectable in these cells (Fig. 8F). This observation lessened the likelihood of PlexinD1 reliance on canonical Hh signaling to activate Gli1, which in turn led us to explore whether PlexinD1 induces Gli1 in a noncanonical way. One noncanonical mechanism of activation of Gli transcription factors occurring independently of Smo involves crosstalk between Hh-Gli and mitogenic signaling pathways, such as MEK-ERK and PI3K-AKT signaling, which can be triggered by various upstream activated RTKs (Pietrobono et al, 2019). This coincided with our observations that PlexinD1 stimulated an ErbB3/cMet-ERK/AKT signaling cascade, and prompted us to investigate whether this signaling cascade could constitute a noncanonical mechanism for PlexinD1-dictated Gli1 activation. First, we examined how Gli1 transcriptional activity responds to ErbB3/cMet in the PlexinD1's context. We demonstrated that pre-treatment with an ErbB3 blocking antibody or SGX-523 blunted the suppressive effect of PlexinD1 knockdown on Gli-Luc activity and that overexpression of ErbB3 or cMet reactivated Gli-Luc activity as repressed by *PLXND1* siRNA in 22Rv1 cells. Blockade of ErbB3 or cMet also resulted in decreases of PlexinD1-induced Gli-Luc

activity to the control level in LNCaP cells (Fig. 8G). To assess whether PlexinD1 influences Gli1 nuclear translocation, a mechanism reported for modulation of Gli1 transcriptional activity in response to mitogenic signaling in cancer (Stecca et al, 2007), we performed a double immunofluorescence (IF) assay to visualize the nuclear localization of Gli1 in the PlexinD1's context using α-tubulin as a cytoplasm marker. Our analysis demonstrated that PlexinD1 knockdown markedly impaired Gli1 nuclear accumulation in 22Rv1 cells, which was restored by overexpression of ErbB3 or cMet. Conversely, blocking endogenous ErbB3, cMet, ERK, or AKT individually all reversed PlexinD1-induced Gli1 nuclear accumulation with simultaneously enhanced cytoplasmic localization of Gli1 in LNCaP cells (Fig. 8H). To determine the effect of PlexinD1 ligands on Gli1 nuclear localization, we showed that siRNA-mediated knockdown of Sema3C or Sema3E decreased, while treatment with recombinant Sema3C or Sema3E proteins stimulated, nuclear accumulation of endogenous Gli1 protein in C4-2B^ENZR cells by an IF assay (Appendix Fig. S12C). Alternatively, we introduced a FLAG-tagged Gli1 expression plasmid into 22Rv1 cells, and using an IF assay with an anti-FLAG antibody we observed less or more nuclear accumulation of exogenous Gli1 protein upon treatment with *SEMA3C*/*SEMA3E* siRNA or recombinant proteins, respectively, compared to the control (Appendix Fig. S12D). Using a complementary approach, we isolated nuclear and cytoplasmic cell extracts to examine Gli1 protein expression levels in individual subcellular fractions. We demonstrated a lower nuclear Gli1 protein level and a higher cytoplasmic Gli1 protein level upon PlexinD1 knockdown in 22Rv1 cells, which was reversed by overexpression of ErbB3 or cMet. Conversely, we observed a notable increase of Gli1 protein localization in the nuclei of PlexinD1-overexpressing LNCaP cells, accompanied by decreased Gli1 presence in the cytoplasm, which was reversed by individual inhibition of ErbB3, cMet, ERK, or AKT (Fig. 8I). Similar to PlexinD1's effect on nuclear Gli1 expression, we further demonstrated that treating cells with *SEMA3C*/*SEMA3E* siRNA or recombinant proteins, respectively, repressed or enhanced the expression levels of Gli1 protein in the nuclei of C4-2B^ENZR cells (Appendix Fig. S12E). These data in aggregate indicate that PlexinD1 drives Gli1 transcriptional activity and nuclear translocation in an ErbB3/cMet-ERK/AKT pathway-dependent manner.

To further investigate whether Gli1 mediates PlexinD1 function in PCa cells, we treated control and PlexinD1-overexpressing LNCaP cells with cyclopamine, a Smo inhibitor (Chen et al, 2002), or GANT61, a small-molecule inhibitor of Gli1 for abrogating Gli1-mediated transcription at the nuclear level (Lauth et al, 2007). Despite no changes observed for the proliferation of PlexinD1-low LNCaP cells in response to both cyclopamine and GANT61, GANT61 treatment reversed PlexinD1-induced proliferation of LNCaP cells to the control level, whereas cyclopamine had minimal effect on the proliferation of PlexinD1-overexpressing LNCaP cells (Fig. 8J). This was paralleled by the observation that GANT61 but not cyclopamine significantly repressed the proliferation of both PlexinD1-high C4-2B$^{ENZR}$ and 22Rv1 cells (Fig. 8K). These data indicate that PlexinD1 confers PCa cell growth advantages through noncanonical activation of Gli1. Moreover, we showed that GANT61 treatment also reversed the expression of multiple stemness, NE, and EMT markers as upregulated by PlexinD1 in LNCaP cells (Fig. 8L).

Lastly, we sought to determine the connection of PlexinD1 with Hh-Gli1 signaling in the clinical setting. We performed GSEA with genes differentially expressed in *PLXND1*-high relative to *PLXND1*-low PCa patient tumors from TCGA cohort. Our analysis revealed that two well-established gene signatures related to Hh signaling, "Hallmark Hedgehog Signaling" and "KEGG Hedgehog Signaling Pathway", were upregulated in *PLXND1*-high versus *PLXND1*-low tumors (Fig. 8M). This was corroborated by higher mRNA levels of several Gli1 target genes (*PTCH1*, *GLI2*, *HHIP*, *BCL2*, and *IGFBP6*) in *PLXND1*-high versus *PLXND1*-low PCa patient tumors from TCGA cohort (Fig. 8N). Together, these results demonstrated that PlexinD1 activates an ErbB3/cMet-ERK/AKT-Gli1 pathway to promote CRPC growth and plasticity.

## Targeting PlexinD1 via D1SP inhibits CRPC growth, migration, and invasion

Having demonstrated the therapeutic targeting potential of PlexinD1 through a shRNA-mediated gene silencing approach for restricting the aggressive phenotype of CRPC cells and tumors, we sought to develop a therapeutic protein inhibitor of PlexinD1. Following a published protocol for designing plexin protein inhibitors (Peacock et al, 2018), we engineered a PlexinD1:Fc decoy protein named D1SP, comprising the PlexinD1 Sema domain and adjacent PSI domain fused to a G4S linker sequence, a hinge, and a human IgG1 Fc domain. A signal peptide sequence was further fused to the N terminus of D1SP to facilitate protein translocation to the cell membrane (Peng et al, 2019) (Fig. 9A). The D1SP recombinant protein was expressed in lentiviral-transduced CHO-K1 cells, effectively secreted for purification, and detected in both whole cell lysate and conditioned medium using immuno-blotting with a hIgGFc-specific antibody. D1SP has an original size of 86 kDa, but due to the glycosylation that is predicted to occur on multiple positions of D1SP, its real size is anticipated to shift up, which could fall in the range of 100 kDa as we observed (Fig. 9B). Then, we conjugated an IgG-PE antibody to D1SP, which was used to treat 22Rv1 cells at various doses for a cell binding assay. Our flow cytometric analysis revealed that D1SP bound to 22Rv1 cells in a dose-dependent manner (Fig. 9C). To examine whether D1SP blocks PlexinD1 interaction with its ligands. We mixed D1SP with recombinant Sema3C or Sema3E proteins in solution and

demonstrated direct binding of D1SP to both Sema3C and Sema3E by co-IP assays (Appendix Fig. S13A). We also found direct interaction of D1SP with recombinant PlexinD1 protein by a co-IP assay when both were incubated in solution (Appendix Fig. S13B). Next, we showed that D1SP dose-dependently restored Sema3E-repressed migration of HUVEC endothelial cells, a previously established measurement of ligand-induced PlexinD1-dependent function (Casazza et al, 2010) (Appendix Fig. S13C). To further determine whether D1SP disrupts the binding of Sema3 ligands to PlexinD1, we performed a PLA and demonstrated that D1SP markedly inhibited the interactions between PlexinD1 and Sema3C and between PlexinD1 and Sema3E (Fig. 9D).

To evaluate the therapeutic utility of D1SP for CRPC treatment, we exposed C4-2B$^{ENZR}$ and 22Rv1 cells to D1SP concurrently with or without recombinant Sema3E protein. Despite a modest but statistically significant induction of cell proliferation upon Sema3E, D1SP substantially inhibited the proliferation of C4-2B$^{ENZR}$ and 22Rv1 cells in both the absence and presence of Sema3E to a similar extent (Fig. 9E). To determine the effect of D1SP in non-tumorigenic and non-CRPC cells, we showed that D1SP reduced the proliferation of C4-2B$^{ENZR}$ and DU145 cells in both time- and dose-dependent manners, in contrast to parallel observations of no changes in the proliferation of HUVEC, RWPE-1, and LNCaP cells (Appendix Fig. S13D,E). Based on the demonstrated interference of D1SP with Sema ligand binding to PlexinD1, we further compared the anti-proliferative effect of D1SP versus targeting Sema3 ligands using gene silencing approaches. We demonstrated that siRNA-mediated knock-down of Sema3C or Sema3E reduced the proliferation of C4-2B$^{ENZR}$ and 22Rv1 cells to a similar extent, with more reductions observed upon concurrent silencing of both Sema3 ligands. Parallel treatment with D1SP achieved the growth inhibitory efficacy comparable to the Sema3C/Sema3E co-silencing approach in both C4-2B$^{ENZR}$ and 22Rv1 cells (Appendix Fig. S13F). Using transwell assays, we also showed that D1SP inhibited the migration and invasion of C4-2B$^{ENZR}$ and 22Rv1 cells (Fig. 9F). Moreover, using organoids derived from LuCaP 147CR, 49, and 173.1 PCa PDX tumors, which have castration-resistant and NE features (Nguyen et al, 2017), we demonstrated notable growth inhibition of all three organoid models upon D1SP treatment (Fig. 9G). To further examine the impact of D1SP on the PlexinD1-dependent mechanism in CRPC cells, we performed a PLA and demonstrated that D1SP impaired the formation of PlexinD1-ErbB3 and PlexinD1-cMet endogenous protein complexes, with up to 82% and 80% drops of cytoplasmic fluorescence intensity, respectively, in both C4-2B$^{ENZR}$ and 22Rv1 cells (Appendix Fig. S13G). Then, we showed that D1SP dose-dependently reduced the phosphoprotein levels of ErbB3, ErbB2, cMet, ERK, and AKT in C4-2B$^{ENZR}$ and 22Rv1 cells (Appendix Fig. S13H). We also showed decreased phosphorylation levels of these proteins upon D1SP treatment in DU145 cells but not in LNCaP cells, which corresponded to the relatively high versus low expression levels of PlexinD1 protein in DU145 and LNCaP cells, respectively. Of note, we found nearly undetectable levels of cMet protein in LNCaP cells under androgen-replete conditions as consistent with previous reports (Knudsen et al, 2002; Liu et al, 2013) (Appendix Fig. S13I).

To further examine the potential therapeutic effect of D1SP on CRPC growth in vivo, we implanted 22Rv1 cells subcutaneously into immunodeficient male mice and administered D1SP or vehicle directly into tumors once every 3 days for a 2-week observation period. Our results showed that D1SP reduced tumor volume and

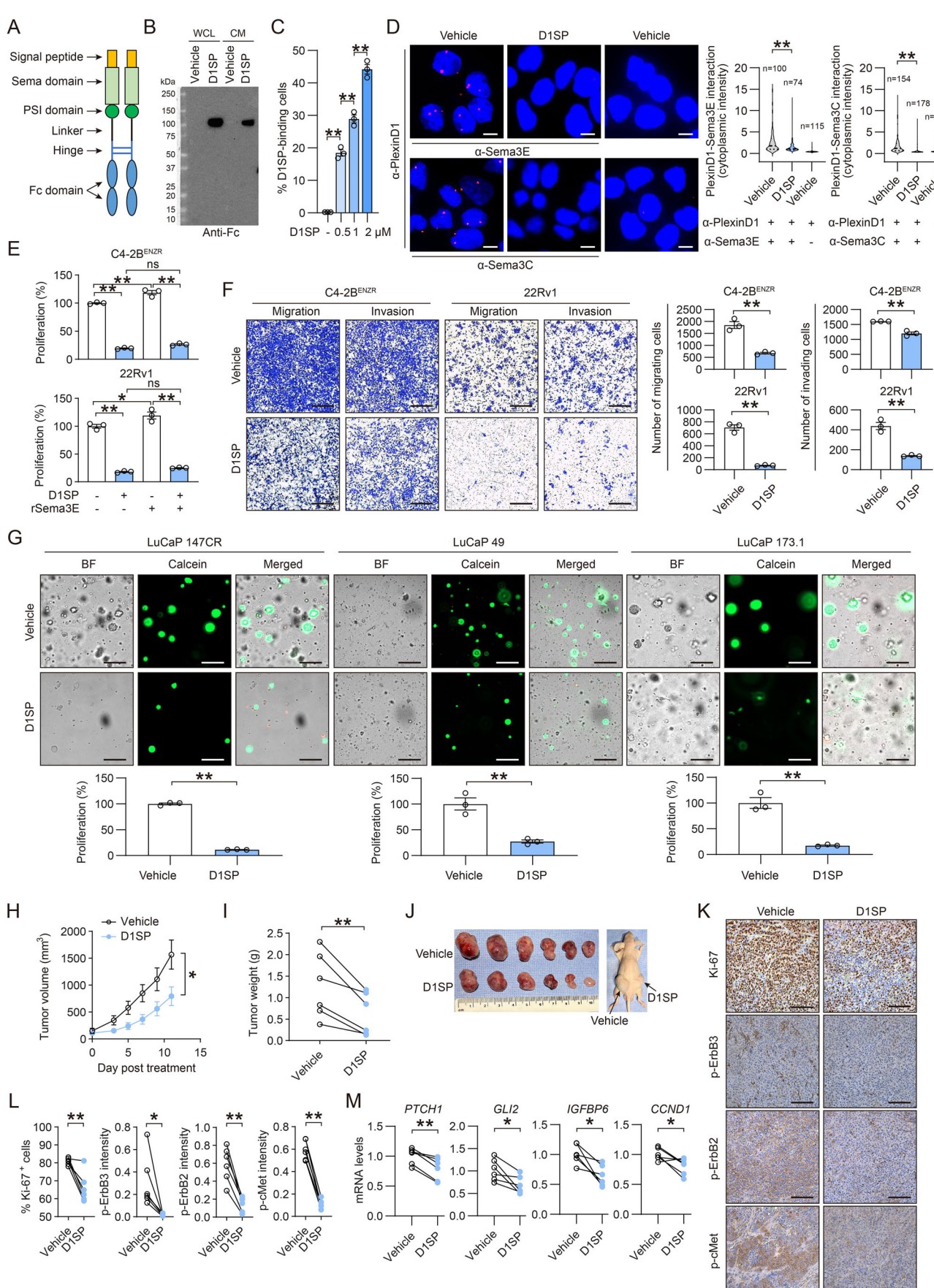

**Figure 9.  D1SP inhibits ErbB3/cMet-Gli1 signaling and CRPC aggressive behavior in vitro and in vivo.**

(A) Graphic depicting D1SP as a recombinant PlexinD1 decoy protein. (B) Western blot of D1SP in whole cell lysate (WCL) and conditioned medium (CM) of D1SP-expressing CHO-K1 cells using a hIgGFc-specific antibody. (C) Flow cytometric analysis of binding of IgG-PE antibody conjugated D1SP at various doses in 22Rv1 cells ($n = 3$ biological replicates). (D) Representative PLA images and quantification of PlexinD1-Sema3E/Sema3C interaction by per-cell cytoplasmic fluorescence intensity in 22Rv1 cells treated with D1SP (1 µM, 2 h) or PBS as a vehicle. PlexinD1 antibody incubation alone served as negative controls. The $n$ of each group indicates the number of cells examined for quantification. Scale bars: 5 µm. (E) Cell viability assays of C4-2B$^{ENZR}$ and 22Rv1 cells stimulated with 200 ng/µl recombinant Sema3E protein and then subjected to treatment with 1 µM D1SP for 7 days ($n = 3$ biological replicates). (F) Transwell-based cell migration and invasion assays of C4-2B$^{ENZR}$ and 22Rv1 cells under 1 µM D1SP treatment ($n = 3$ biological replicates). Scale bars: 400 µm. (G) Representative brightfield and fluorescence microscopic images and quantification of LuCaP 147CR, 49 and 173.1 PCa PDX-derived live organoids after incubation with 1 µM D1SP or PBS as a vehicle for 10 days ($n = 3$ biological replicates). Scale bars: 100 µm. (H–J) Tumor growth curves (H), endpoint tumor weights (I), and anatomic tumor images (J) of s.c. 22Rv1 xenografts grown in male nude mice ($n = 6$ tumors) and receiving intratumoral injection of D1SP (30 µg/tumor, 2–3 times per week) and saline on the right and left flanks of mice, respectively. (K, L) Representative images (K) and quantification (L) of IHC staining of Ki-67, p-ErbB3, p-ErbB2, and p-cMet ($n = 6$ tumors) in control and D1SP-treated 22Rv1 tumors. Scale bars: 100 µm. (M) qPCR of select Gli1 target genes in control and D1SP-treated 22Rv1 tumors ($n = 6$ tumors). Data information: In (C–H), data are presented as mean ± SEM. In (C, D), $P$ values were determined by one-way ANOVA with Dunnett's multiple comparisons test. In (E), $P$ values were determined by one-way ANOVA with Tukey's multiple comparisons test. In (F–H), $P$ values were determined by unpaired two-tailed Student's $t$-test. In (I, L, M), $P$ values were determined by paired two-tailed Student's $t$-test. *$P < 0.05$, **$P < 0.01$; ns, not significant. Exact $P$ values are listed in Appendix Table S4. Source data are available online for this figure.

weight compared with controls (Fig. 9H–J). Examining tumor samples by IHC, D1SP-treated tumors had lower percentages of Ki-67+ cells and less protein expression of p-ErbB3, p-ErbB2, and p-cMet compared with control tumors (Fig. 9K,L). Further analysis of tumor samples also revealed prevalent reductions in the mRNA expression of multiple Gli1 target genes in D1SP-treated tumors compared with controls (Fig. 9M).

## Discussion

In this study, we identified PlexinD1 as a pivotal executor driving an aggressive phenotype of PCa and defined a PlexinD1-dictated mechanism conferring PCa growth advantages, stemness, and lineage plasticity. PlexinD1 emerged from our unbiased transcriptomic profiling analysis of lineage-related androgen-dependent PCa and mCRPC cells, which faithfully reflect the molecular and phenotypic alterations occurring alongside PCa lethal disease progression in the clinic. PlexinD1 was also found in a 95 neural lineage gene signature likely to account for the PCa's aggressive properties, as identified recently from a combination of our C4-2B$^{ENZR}$ mCRPC cell model and a CRPC/NEPC patient dataset (Ning et al, 2022). Our findings suggest that PlexinD1 can regulate multiple types of CRPC cell behavior, including proliferation, migration, invasion, metastasis, and therapy resistance. Part of our findings are consistent with prior observations of PlexinD1's role in PCa or other types of cancer as reported by other groups (Casazza et al, 2010; Rehman et al, 2016). Our conclusion that PlexinD1 confers NE lineage plasticity and associated ENZ resistance corroborates a recent study by Chen et al, which revealed that PlexinD1 is critical for maintaining NEPC growth and differentiation using multiple NEPC models including the same C4-2B$^{ENZR}$ cell line as we used as a treatment-induced NEPC cell model (Chen et al, 2024). In addition, we showed that PlexinD1 can function in PCa regardless of AR status, androgen responsiveness, and disease subtype. Together, these findings implicate PlexinD1 as an attractive therapeutic target potentially against a wide range of aggressive and lethal PCa variants including CRPC and NEPC.

The plexin receptors and their ligands, Sema axon guidance molecules, are often inappropriately expressed in cancers including PCa (Herman and Meadows, 2007; Peacock et al, 2018; Wong et al, 2007). Blanc et al provided the initial observation of increased PlexinD1 expression in human PCa tumors compared with benign tissue (Blanc et al, 2011). Our study reinforced and expanded on previous findings to demonstrate that higher PlexinD1 expression in PCa patient tumor tissue is associated with high Gleason grades, hormone therapy failure, NE differentiation, and metastasis in multiple independent cohorts. Intriguingly, we showed that overexpression of PlexinD1 without modulation of Sema ligand levels was sufficient to promote PCa cell proliferation, migration, invasion, ENZ resistance, stemness, and plasticity, which could be possibly attributed to receptor clustering for mimicking ligand binding, a mechanism demonstrated by PlexinB1 previously (Giordano et al, 2002). On the other hand, we found upregulation of Sema3C and Sema3E in parallel with PlexinD1 in our C4-2B$^{ENZR}$ mCRPC cell model, suggesting that the elevated levels of these Sema ligands may enhance PlexinD1 function in CRPC. Indeed, we showed that these Sema3 ligands could recapitulate PlexinD1's effect on cell growth and activation of downstream ErbB3/cMet-ERK/AKT-noncanonical Hh/Gli1 signaling in PCa cells. We also demonstrated that PlexinD1 transactivates RTKs in a Sema-dependent manner to confer cell growth and plasticity. In addition, Tam et al showed that ectopic expression of Sema3C in RWPE-1 cells promoted the upregulation of EMT and stem markers, cell plasticity, migration and invasion in vitro, and cell dissemination in vivo (Tam et al, 2017b), which is similar to the PlexinD1-induced phenotype in PCa cells from our observations. These data collectively suggest a receptor-driven model with contributions from both the receptor and ligand for PlexinD1 to operate in CRPC.

We demonstrated that PlexinD1 is transcriptionally suppressed by AR via direct AR interaction with *PLXND1* promoter, which is a likely mechanism for the upregulation of PlexinD1 observed in PCa tumor clinical samples post-hormone therapy as well as in AR-negative CRPC and NEPC cells. We also found that AR signaling negatively regulates the transcription and expression of *SEMA3C* and *SEMA3E*, as evidenced by the findings that androgen depletion with CSS medium promoted, while R1881 reduced, their mRNA levels in several androgen-responsive AR+ PCa cell lines (LNCaP, LAPC4, and VCaP), which corroborated the upregulation of these Sema3 ligands in AR-repressed C4-2B$^{ENZR}$ mCRPC cells. But interestingly, Tam et al showed that R1881 increased, while ENZ treatment decreased, *SEMA3C* mRNA levels in LNCaP cells, which was driven by AR through an activating ARE within the *SEMA3C* intron 2 (Tam et al, 2017a). These conflicting results may be due in part to potential differing characteristics like the androgen

responses of specific genes in LNCaP cells used in different studies, which could be influenced by multiple factors such as cell sources, passages and culture methods, necessitating further clarifications from additional investigations. On the other hand, we demonstrated the ability of PlexinD1 to induce a stem-like and multilineage cellular state with repression of the AR+ luminal cell lineage and activation of basal, NE, and mesenchymal phenotypes. Thus, the AR-PlexinD1 reciprocal interaction may constitute a possible vicious cycle triggered by ARSIs to favor accumulative upregulation of PlexinD1 towards the eventual development of PlexinD1-driven plastic, therapy-resistant PCa variants. Mechanistically, we showed that PlexinD1 transactivates an ErbB3/cMet-ERK/AKT-noncanonical Hh/Gli1 signaling cascade to promote PCa cell growth and plasticity. Our findings on PlexinD1's effect on Hh-Gli1 signaling in PCa cells are also consistent with recent similar observations made outside the context of malignancies, where knockdown of multiple Plexins including PlexinD1 was shown to reduce SMO agonist-induced activation of Gli1 in mouse embryonic fibroblasts (Pinskey et al, 2022). Hh signaling has been demonstrated to modulate stemness and cellular plasticity in different types of organs and cells, such as the brain, in both normal physiology and disease states (Cazet et al, 2018; Magistri et al, 2017; Navon et al, 2020; Yao et al, 2016). Specifically in PCa, loss of SMO, a key canonical transducer of Hh signaling, was demonstrated as a molecular event occurring during transition from prostate adenocarcinoma into NEPC (Wang et al, 2021). Complementing this study, our findings provided evidence establishing the role of noncanonical Hh-Gli1 signaling under PlexinD1-RTKs crosstalk in driving cellular plasticity in PCa. Nevertheless, future studies are warranted to elucidate the precise epigenetic mechanism by which Gli1 regulates PlexinD1-driven cellular plasticity in PCa cells.

We presented substantial evidence supporting that targeting PlexinD1 by D1SP, a PlexinD1 decoy soluble receptor, effectively restricted CRPC growth in multiple experimental models including cell lines, PDX-derived organoids, and cell line-derived xenografts. We showed that D1SP directly binds to PlexinD1 and its ligands and interferes with PlexinD1-RTK interaction and the subsequent signaling cascade. We also demonstrated D1SP's growth inhibitory efficacy in PlexinD1-high CRPC cells (DU145, C4-2B$^{ENZR}$, and 22Rv1) but not in PlexinD1-low non-CRPC cells (LNCaP) and normal cells (RWPE-1 and HUVEC), which could be possibly attributed to the much higher expression levels of both PlexinD1 and its downstream RTK effectors, such as cMet as shown previously (Knudsen et al, 2002), for conveying PlexinD1-triggered growth-stimulating signals in CRPC cells compared with non-CRPC and normal cells. The ineffective inhibition of the growth and RTK signaling in LNCaP cells by a similar PlexinD1-based decoy receptor was also reported previously (Peacock et al, 2018). In addition, we noted some differing results between ours and the previous study by Peacock et al mainly regarding the experiments involving DU145 cells. Peacock et al showed that siRNA-mediated knockdown of PlexinD1 did not reduce Sema3C-induced proliferation of DU145 cells and D1SP had no inhibitory activity of Sema3C-induced activation of RTK signaling in DU145 cells, whereas we made the opposite observations. We argue that this discrepancy may result from different expression levels of PlexinD1 protein in the DU145 cells used in the different studies, which could be influenced by multiple factors such as cell sources, passages, and culture methods. Compared to respective PlexinD1-

low LNCaP cells, we demonstrated a significantly higher expression level of PlexinD1 protein in our DU145 cells, while a lower level of PlexinD1 protein was shown in the DU145 cells used by Peacock et al. The varying PlexinD1 expression levels are likely to account for the contrasting responses of different DU145 cells to PlexinD1 blockade. Nevertheless, coupling our study with others may reveal a correlation between D1SP's effectiveness and PlexinD1 expression levels in prostate cancer cells, suggesting the potential therapeutic utility of D1SP specifically in PlexinD1-high prostate cancer. Lastly, despite our findings demonstrating D1SP's in vivo anti-CRPC efficacy, we recognize the translational limitations of our study. We used intratumoral injections to deliver high-dose D1SP directly to tumors for assessment of its therapeutic effect, which is not an ideal approach to prove its amenability for pharmacological intervention in vivo and evaluate potential adverse effects or signs of toxicity. Future efforts are thus required for better examination of D1SP's in vivo anti-CRPC efficacy using intraperitoneal injection as previously validated by other Plexin protein inhibitors (Peacock et al, 2018), which would aid in rigorously and comprehensively evaluating the therapeutic benefit of D1SP in a translational perspective.

In summary, we identified PlexinD1 as a critical driver of aggressive PCa phenotypes and a novel facilitator of PCa lineage plasticity. Our study also suggests that targeted blockade of PlexinD1 activity and signaling via a protein inhibitor could be a potential therapy against lethal PCa variants such as CRPC.

# Methods

**Reagents and tools table**

| Reagent/Resource | Reference or Source | Identifier or Catalog Number |
|---|---|---|
| **Experimental models** | | |
| LNCaP | ATCC | CRL-1740 |
| PC-3 | ATCC | CRL-1435 |
| DU145 | ATCC | HTB-81 |
| 22Rv1 | ATCC | CRL-2505 |
| VCaP | ATCC | CRL-2876 |
| LASCPC-01 | ATCC | CRL-3356 |
| NCI-H660 | ATCC | CRL-5813 |
| NE1.8 | ATCC | PTA-3569 |
| RWPE-1 | ATCC | CRL-3607 |
| 293T | ATCC | CRL-3216 |
| HUVEC | ATCC | CRL-2854 |
| LAPC4 | Dr. Michael Freeman Lab (Cedars-Sinai Medical Center, Los Angeles, CA, USA) | N/A |
| ARCaP$_M$ | Dr. Leland Chung Lab (Cedars-Sinai Medical Center, Los Angeles, CA, USA) | N/A |
| C4-2B$^{ENZR}$ | Dr. Allen Gao Lab (UC Davis, Sacramento, CA, USA) | N/A |

| Reagent/Resource | Reference or Source | Identifier or Catalog Number |
|---|---|---|
| J:NU (nude) mice | Jackson Laboratory | 007850 |
| NOD.Cg-*Prkdc*scid *Il2rg*tm1wjl/SzJ (NSG) mice | Jackson Laboratory | 005557 |
| Patient-derived xenograft (PDX) | Dr. Eva Corey Lab (University of Washington, Seattle, WA, USA) | LuCaP 49, 147CR and 173.1 |
| **Recombinant DNA** | | |
| pLVX-AcGFP1-N1 | Takara Bio | 632154 |
| pLVX-PLXND1 | This study | N/A |
| pCMV-VSV-G | Addgene | 8454 |
| pCMV delta R8.2 | Addgene | 12263 |
| pGL4.26 | Promega | E8441 |
| PSA-Luc | Dr. Gerhard Coetzee Lab (Van Andel Institute, Grand Rapids, MI, USA) | N/A |
| Gli-Luc | Dr. Hiroshi Sasaki Lab (RIKEN Center for Developmental Biology, Kobe, Japan) | N/A |
| pCMV-SPORT6-ERBB3 | Horizon Discovery | MHS6278-202759664 |
| pLenti-MetGFP | Addgene | 37560 |
| pCMV-3×FLAG-hGli1 | Addgene | 84922 |
| **Antibodies** | | |
| PlexinD1 (WB: 1/1000; PLA: 1/50; co-IP: 1/50) | Cell Signaling Technology | 92470 |
| PlexinD1 (IHC: 1/50) | Origene | TA351543 |
| PlexinD1 (PLA: 1/50) | R&D Systems | MAB4160 |
| Sema3C (WB: 1/500; PLA: 1/200) | Proteintech | 19242-1-AP |
| Sema3E (WB: 1/500; PLA: 1/200) | Thermo Fisher Scientific | PA5-56140 |
| AR (WB: 1/1000; ChIP: 1/100) | Cell Signaling Technology | 5153 |
| PSA (WB: 1/1000) | Santa Cruz Biotechnology | sc-7638 |
| CHGA (WB: 1/1000) | Proteintech | 60135 |
| SYP (WB: 1/1000; IHC: 1/50) | Santa Cruz Biotechnology | sc-17750 |
| CD56 (WB: 1/500) | Santa Cruz Biotechnology | sc-7326 |
| NSE (WB: 1/1000) | Santa Cruz Biotechnology | sc-21738 |
| E-cadherin (WB: 1/500) | Santa Cruz Biotechnology | sc-8426 |
| N-cadherin (WB: 1/500) | Santa Cruz Biotechnology | sc-7939 |

| Reagent/Resource | Reference or Source | Identifier or Catalog Number |
|---|---|---|
| Twist1 (WB: 1/1000) | Novus Biologicals | NBP2-37364 |
| SOX2 (WB: 1/1000) | Santa Cruz Biotechnology | sc-365823 |
| NANOG (WB: 1/200) | Santa Cruz Biotechnology | sc-134218 |
| OCT3/4 (WB: 1/500) | Santa Cruz Biotechnology | sc-5279 |
| LIN28 (WB: 1/1000) | Santa Cruz Biotechnology | sc-67266 |
| CK8 (WB: 1/1000) | Developmental Studies Hybridoma Bank at the University of Iowa | TROMA-I |
| CK5 (WB: 1/500) | Santa Cruz Biotechnology | sc-32721 |
| p63 (WB: 1/500) | Santa Cruz Biotechnology | sc-25268 |
| p-ErbB3 (Tyr1289) (WB: 1/1000; IHC: 1/1000) | Cell Signaling Technology | 2842 |
| ErbB3 (WB: 1/1000; PLA: 1/100) | Cell Signaling Technology | 12708 |
| p-ErbB2 (Tyr1221/1222) (WB: 1/1000; IHC: 1/400) | Cell Signaling Technology | 2243 |
| ErbB2 (WB: 1/1000) | Cell Signaling Technology | 4290 |
| p-cMet (Tyr1234/1235) (WB: 1/1000; IHC: 1/200) | Cell Signaling Technology | 3077 |
| cMet (WB: 1/1000; PLA: 1/1000) | Cell Signaling Technology | 8198 |
| p-ERK (WB: 1/1000) | Santa Cruz Biotechnology | sc-7383 |
| ERK (WB: 1/1000) | Santa Cruz Biotechnology | sc-514302 |
| p-AKT (Ser473) (WB: 1/1000) | Cell Signaling Technology | 4060 |
| AKT (WB: 1/1000) | Cell Signaling Technology | 4691 |
| Gli1 (WB: 1/2000) | Proteintech | 66905-1-Ig |
| Gli1 (WB: 1/1000; IF: 1/50) | Santa Cruz Biotechnology | sc-20687 |
| α-Tubulin (IF: 1/50) | Santa Cruz Biotechnology | sc-5286 |
| Histone H3 (WB: 1/2000) | Cell Signaling Technology | 4499 |
| Acetyl-Histone H3 (ChIP: 1/50) | Cell Signaling Technology | 9649 |
| GAPDH (WB: 1/1000) | Santa Cruz Biotechnology | sc-47724 |
| BRN2 (WB: 1/1000) | Cell Signaling Technology | 12137 |
| Ki-67 (IHC: 1/400) | Cell Signaling Technology | 9027 |

| Reagent/Resource | Reference or Source | Identifier or Catalog Number |
|---|---|---|
| Human IgG Fc (WB: 1/1000; co-IP: 1/50) | Sigma-Aldrich | I2136 |
| FLAG (IF: 1/200) | Sigma-Aldrich | F1804 |
| β-Actin (WB: 1/1000) | Santa Cruz Biotechnology | sc-69879 |
| Goat anti-mouse IgG-HRP (WB: 1/5000) | Santa Cruz Biotechnology | sc-2031 |
| Goat anti-rabbit IgG-HRP (WB: 1/5000) | Cell Signaling Technology | 7074 |
| Mouse anti-goat IgG-HRP (WB: 1/5000) | Santa Cruz Biotechnology | sc-2354 |
| Goat anti-mouse IgG (H&L) Alexa Fluor 488 (IF: 1/1000) | Thermo Fisher Scientific | A32723 |
| Goat anti-rabbit IgG (H&L) Alexa Fluor 555 (IF: 1/1000) | Thermo Fisher Scientific | A32732 |
| **Oligonucleotides and other sequence-based reagents** | | |
| qPCR primers | This study | Appendix Table S3 |
| Human *PLXND1* siRNA | Horizon Discovery | L-014121-01-0010 |
| Human *SEMA3C* siRNA | Santa Cruz Biotechnology | sc-44091 |
| Human *SEMA3E* siRNA | Santa Cruz Biotechnology | sc-61520 |
| Human *BRN2* siRNA | Santa Cruz Biotechnology | sc-29837 |
| Non-targeting control siRNA | Santa Cruz Biotechnology | sc-37007 |
| **Chemicals, Enzymes and other reagents** | | |
| B-27 supplement | Thermo Fisher Scientific | 17504044 |
| Bovine serum albumin | Fisher Scientific | BP1600-100 |
| Crystal violet | Fisher Scientific | C581-25 |
| Cyclopamine | Selleck Chemicals | S1146 |
| D1SP protein | This study | N/A |
| DAPI solution | Thermo Fisher Scientific | 62248 |
| D-Luciferin, sodium salt | Gold Biotechnology | LUCNA-100 |
| *EcoR*I | New England Biolabs | R0101S |
| Enzalutamide | Selleck Chemicals | S1250 |
| β-Estradiol | Sigma-Aldrich | E2758 |
| Formaldehyde | Fisher Scientific | BP531-500 |
| GANT61 | Selleck Chemicals | S8075 |
| L-Glutamine | Corning | 25-005-CI |
| Halt protease and phosphatase inhibitor cocktail | Thermo Fisher Scientific | PI78443 |

| Reagent/Resource | Reference or Source | Identifier or Catalog Number |
|---|---|---|
| Hydrocortisone | Sigma-Aldrich | H0888 |
| Ipatasertib | MedChemExpress | HY-15186 |
| Lipofectamine 3000 transfection reagent | Thermo Fisher Scientific | L3000015 |
| Low melt agarose | IBI Scientific | IB70051 |
| Matrigel basement membrane matrix | Corning | 354234 |
| M-MLV reverse transcriptase | Promega | M1705 |
| Nitrotetrazolium blue chloride | Abcam | ab146262 |
| Puromycin dihydrochloride | Thermo Fisher Scientific | A1113803 |
| Recombinant human bFGF protein | Thermo Fisher Scientific | 13256-029 |
| Recombinant human EGF protein | Thermo Fisher Scientific | PHG0314 |
| Recombinant human ErbB3 protein | R&D Systems | 10368-RB-050 |
| Recombinant human insulin solution | Sigma-Aldrich | I9278 |
| Recombinant human Sema3C protein | R&D Systems | 5570-S3-050 |
| Recombinant human Sema3E protein | R&D Systems | 3239-S3B-025 |
| Recombinant human PlexinD1 protein | R&D Systems | 4160-PD-050 |
| SGX-523 | MedChemExpress | HY-12019 |
| Sodium selenite | Sigma-Aldrich | S5261 |
| Transferrin | Sigma-Aldrich | T2036 |
| Trastuzumab | MedChemExpress | HY-P9907 |
| Triton X-100 | Fisher Scientific | BP151-500 |
| TrypLE enzyme | Thermo Fisher Scientific | 12563029 |
| Tucatinib | MedChemExpress | HY-16069 |
| Tween 20 | Fisher Scientific | BP337-500 |
| U0126 | Selleck Chemicals | S1102 |
| *Xho*I | New England Biolabs | R0146S |
| Y-27632 | MedChemExpress | HY-10583 |
| **Software** | | |
| ImageJ | NIH | https://imagej.nih.gov/ij/ |
| Image Lab 6.1 | Bio-Rad Laboratories | https://www.bio-rad.com/en-us/product/image-lab-software |
| HALO | Indica Labs | https://indicalab.com/halo/ |
| GSEA v4.0.3 | Broad Institute | https://www.gsea-msigdb.org/ |

| Reagent/Resource | Reference or Source | Identifier or Catalog Number |
|---|---|---|
| GraphPad Prism 9 | GraphPad | https://www.graphpad.com/ |
| **Other** | | |
| DMEM | Corning | 10-013-CV |
| DMEM/F12 medium | Thermo Fisher Scientific | 12634010 |
| RPMI-1640 medium | Corning | 10-040-CV |
| Pheno red-free RPMI-1640 medium | Corning | 17-105-CV |
| Prostate epithelial cell growth basal medium | Lonza | CC-3165 |
| Keratinocyte SFM | Thermo Fisher Scientific | 17005042 |
| Vascular cell basal medium | ATCC | PCS-100-030 |
| Fetal bovine serum | Atlanta Biologicals | S12450H |
| Charcoal-stripped serum | Atlanta Biologicals | S11650H |
| Penicillin/streptomycin | Corning | 30-002-CI |
| Endothelial Cell Growth Kit-BBE | ATCC | PCS-100-040 |
| Prostate cancer tissue microarray | US Biomax | PR807c |
| Prostate cancer tissue microarray | Prostate Cancer Biorepository Network (PCBN) NYU site | N/A |
| Subcloning efficiency DH5α competent cells | Thermo Fisher Scientific | 18-265-017 |
| pLKO.1-puro non-mammalian shRNA control lentiviral particles | Sigma-Aldrich | SHC002V |
| pLKO.1-puro human *PLXND1* shRNA#1 lentiviral particles | Sigma-Aldrich | SHCLNV_TRCN0000061550 |
| pLKO.1-puro human *PLXND1* shRNA#2 lentiviral particles | Sigma-Aldrich | SHCLNV_TRCN0000061552 |
| ALDEFLUOR Kit | Stem Cell Technologies | 01700 |
| CellTiter-Glo Luminescent Cell Viability Assay | Promega | G7571 |
| CellTiter-Glo 3D Cell Viability Assay Kit | Promega | G9681 |
| Dual-Luciferase Reporter Assay | Promega | E1910 |
| Duolink In Situ Red Starter Kit | Sigma-Aldrich | DUO92101 |
| Human HGF ELISA Kit | RayBiotech | ELH-HGF |

| Reagent/Resource | Reference or Source | Identifier or Catalog Number |
|---|---|---|
| Human RTK Phosphorylation Array | RayBiotech | AAJ-PRTK-1-2 |
| NE-PER Nuclear and Cytoplasmic Extraction Reagents | Thermo Fisher Scientific | 78833 |
| LIVE/DEAD Viability/Cytotoxicity Kit | Thermo Fisher Scientific | L3224 |
| Pierce BCA Protein Assay Kit | Thermo Fisher Scientific | 23225 |
| Protein G magnetic beads | Cell Signaling Technology | 9006 |
| QIAquick Gel Extraction Kit | Qiagen | 28706 |
| QIAquick PCR Purification Kit | Qiagen | 28104 |
| Quick Ligation Kit | New England Biolabs | M2200S |
| QuickChange II XL Site-Directed Mutagenesis Kit | Agilent Technologies | E200524 |
| RNeasy Mini Kit | Qiagen | 74104 |
| SimpleChIP Enzymatic Chromatin IP Kit | Cell Signaling Technology | 9002S |

## Clinical specimens

The TMA for PlexinD1 evaluation in Fig. 2A,B was obtained from PCa tissue array PR807c (US Biomax) containing 50 cases of prostate adenocarcinoma and 30 cases of normal prostate tissue with a single core and 1.5-mm-diameter size. The TMA for PlexinD1 evaluation and association with clinical outcomes in Fig. 2D was obtained from the NYU site of the Prostate Cancer Biorepository Network (PCBN), which contains 86 cases of primary adenocarcinoma with duplicate cores per patient. The primary ($n = 11$), bone metastatic ($n = 19$), and castration-resistant ($n = 20$) PCa TMAs for evaluating PlexinD1 and its correlation with SYP in Figs. 2F and 6G were constructed by the Biobank of Taipei General Veterans Hospital, as reviewed and approved by the Institutional Review Board of Taipei General Veterans Hospital (No. 2015-06-006C) with informed consent obtained from all patients. The experiments conformed to the principles set out in the WMA Declaration of Helsinki and the Department of Health and Human Services Belmont Report.

## Cell lines and cell culture

The human PCa LNCaP, 22Rv1, VCaP, PC-3, DU145, LASCPC-01, NCI-H660, NE1.8, human normal prostate epithelial RWPE-1, human normal endothelial HUVEC, and human embryonic kidney 293T cell lines were obtained from the American Type Culture Collection (ATCC). The human PCa ARCaP$_M$ cell line was provided by Dr. Leland Chung (Cedars-Sinai Medical Center).

The human PCa LAPC4 cell line was provided by Dr. Michael Freeman (Cedars-Sinai Medical Center). The ENZ-resistant C4-2B cell line, C4-2B$^{ENZR}$, was generated as described previously (Liu et al, 2015). All human cell lines were authenticated by short tandem repeat profiling recently, tested for *Mycoplasma* regularly by the MycoProbe Mycoplasma Detection Kit (R&D Systems), and used with the number of cell passages below 10. 293T and VCaP cells were cultured in DMEM medium (Corning) supplemented with 10% fetal bovine serum (FBS, Atlanta Biologicals) and 1% penicillin/streptomycin (Corning). LNCaP, C4-2B$^{ENZR}$, 22Rv1, LAPC4, PC-3, DU145, and ARCaP$_M$ cells were cultured in RPMI-1640 medium (Corning) supplemented with 10% FBS and 1% penicillin/streptomycin. C4-2B$^{ENZR}$ cells were cultured further in the continuous presence of 20 μM ENZ. LASCPC-01 and NCI-H660 cells were cultured in RPMI-1640 medium supplemented with 10% FBS, 1% penicillin/streptomycin, 5 μg/ml insulin (Sigma-Aldrich), 0.01 mg/ml transferrin (Sigma-Aldrich), 30 nM sodium selenite (Sigma-Aldrich), 10 nM hydrocortisone (Sigma-Aldrich), 10 nM β-estradiol (Sigma-Aldrich), and 4 mM L-glutamine (Corning). NE1.8 cells were cultured in phenol red-free RPMI-1640 medium (Corning) supplemented with 10% charcoal-stripped serum (CSS, Atlanta Biologicals) and 1% penicillin/streptomycin. RWPE-1 cells were cultured in Keratinocyte SFM (Thermo Fisher Scientific). HUVEC cells were cultured in Vascular Cell Basal Medium (ATCC) supplemented with 0.2% bovine brain extract, 5 ng/ml rhEGF, 10 mM L-glutamine, 0.75 unit/mL heparin sulfate, 1 μg/ml hydrocortisone, 50 μg/ml ascorbic acid, and 2% FBS as included in the Endothelial Cell Growth Kit-BBE (ATCC).

## Plasmids and reagents

A human *PLXND1* lentiviral expression construct was generated by inserting the human *PLXND1* coding region at *XhoI*/*EcoRI* sites in pLVX-AcGFP1-N1 vector (Takara Bio) containing a puromycin-resistant gene. The human *PLXND1* expression construct as provided by Dr. Luca Tamagnone (University of Torino, Italy) was used as a template for the subcloning described above. The human *ERBB3* expression construct was purchased from Horizon Discovery. The pLenti-MetGFP expression construct was provided by Dr. David Rimm and obtained from Addgene. The human FLAG-tagged *GLI1* expression construct was provided by Dr. Martin Fernandez-Zapico and obtained from Addgene. A human 221-bp *PLXND1* promoter luciferase reporter construct, *PLXND1* promoter-Luc, was generated by inserting the corresponding *PLXND1* promoter sequence (3251–3471 upstream of transcription start site) upstream of a minimal promoter and the *Firefly* luciferase gene of pGL4.26 vector (Promega). Primer sequences for cloning the sequence from LNCaP genomic DNA were forward 5′-CGGGGTACCCTGGATTATCACCT ATTTCACATTTGTC-3′ and reverse 5′-CCGCTCGAGTTTAATA GAGAGAAGGTCTCAAACTCAG-3′. The Gli-Luc reporter (Sasaki et al, 1997) was provided by Dr. Hiroshi Sasaki (RIKEN Center for Developmental Biology, Japan). The pRL-TK *Renilla* luciferase reporter was purchased from Promega. Human *PLXND1* and non-target control shRNA lentiviral particles were purchased from Sigma-Aldrich. Human *PLXND1* siRNA pool was purchased from Horizon Discovery. Human *SEMA3C*, *SEMA3E*, and *BRN2* siRNA pools and non-target control siRNA were purchased from Santa Cruz Biotechnology. R1881 was purchased from Sigma-Aldrich. ENZ, SGX-523, and Ipatasertib were purchased from MedChemExpress. U0126,

cyclopamine, and GANT61 were purchased from Selleck Chemicals. An ErbB3 neutralizing antibody and human recombinant Sema3E, Sema3C, PlexinD1, and ErbB3 proteins were purchased from R&D Systems. Tucatinib and trastuzumab were purchased from MedChemExpress.

## Biochemical analyses

Total RNA was isolated using a RNeasy Mini Kit (Qiagen) and reverse-transcribed to cDNA by M-MLV (Promega) following the manufacturers' instructions. Subsequently, qPCR was conducted using SYBR Green PCR Master Mix and run with the Applied Biosystems QuantStudio 3 Real-Time PCR System (Thermo Fisher Scientific). PCR conditions included an initial denaturation step of 10 min at 95 °C, followed by 40 cycles of PCR consisting of 15 s at 95 °C and 1 min at 60 °C. PCR data were analyzed by the $2^{-\Delta\Delta CT}$ method (Livak and Schmittgen, 2001). Details on the primers used for qPCR are provided in Appendix Table S3. For immunoblots, cells were extracted with RIPA buffer supplemented with a protease/phosphatase inhibitor cocktail (Thermo Fisher Scientific). Nuclear and cytoplasmic proteins were extracted using the NE-PER Nuclear and Cytoplasmic Extraction Kit (Thermo Fisher Scientific) according to the manufacturer's manual. Blots were performed as described previously (Wu et al, 2009) using different primary antibodies. The human receptor tyrosine kinase phosphorylation antibody array (RayBiotech) was performed following the manufacturer's instructions.

## Generation of stable overexpression and knockdown cells

Stable shRNA-mediated PlexinD1 knockdown was achieved by infecting cells with lentivirus expressing *PLXND1* shRNA TRCN0000061550 (shPlexinD1#1, mainly used in this study and usually dubbed "shPlexinD1") or TRCN0000061552 (shPlexinD1#2), followed by 2-week selection with 2 μg/ml puromycin to establish stable cell lines. A non-target control shRNA (shCon) was used as a control for stable knockdown cells. Lentivirus production was performed for stably overexpressing PlexinD1 in PCa cells. Briefly, 293T cells were co-transfected with a *PLXND1*-expressing lentiviral construct, pCMV delta R8.2 (Addgene), and pCMV-VSV-G (Addgene) in a 4:2:1 ratio using Lipofectamine 3000 reagent (Thermo Fisher Scientific) following the manufacturer's instructions. The medium was changed 12 h after transfection. The medium containing lentivirus was harvested 48 h after transfection. PCa cells were infected with lentivirus in the presence of 8 μg/ml polybrene followed by 2-week selection with 2 μg/ml puromycin. An empty lentiviral construct was used as a control for stable overexpression of PlexinD1 in cells.

## Analyses of cell proliferation, colony formation, and tumorsphere formation

To determine cell proliferation, cells were seeded in 96-well plates (1000 cells/well) and grown up to 7 days. Cell proliferation was determined by CellTiter-Glo (Promega) following the manufacturer's instructions. For determining anchorage-independent colony formation, cells were suspended in culture medium containing 0.3% agarose at a density of 400 cells/well and placed on top of solidified 0.6% agarose layer in 6-well

plates. Cells were grown until visible colonies formed and then stained with nitrotetrazolium blue chloride (Abcam) following the manufacturer's instructions. For determining tumorsphere formation, cells were seeded on ultra-low attachment 24-well plates (Corning) at a density of 2000 cells/well in serum-free prostate epithelial basal medium (Lonza) supplemented with B-27 supplement (Thermo Fisher Scientific), 20 ng/ml EGF (Thermo Fisher Scientific), 10 ng/ml bFGF (Thermo Fisher Scientific), 5 µg/ml insulin, and 0.4% BSA (Sigma-Aldrich). The medium was replenished every 3 days until spheres arose 2–3 weeks after plating. Colonies and spheres were imaged with a ChemiDoc Touch Imager (Bio-Rad) and the number of colonies or spheres containing ≥50 cells were enumerated with ImageJ software (NIH).

## Analyses of cell migration, invasion, and wound healing ability

To determine cell migration and invasion, 6.5-mm transwell inserts (8-µM pore size) were used, which were further coated with growth factor-reduced Matrigel (Corning) specifically for invasion assays. $5 \times 10^4$ cells were serum starved overnight before seeding to eliminate the interference of proliferative effect with cell migration and invasion. The inserts were incubated in medium containing 10% FBS as a chemoattractant. After 24–48 h, cells that passed through the membrane and translocated to the lower surface of the insert filter were fixed and stained with 1% crystal violet solution. Assays were quantified by counting the number of stained nuclei in 3–5 independent fields in each transwell by ImageJ software. For 3D invasion assays, $5 \times 10^4$ cells in 10 µl 100% Matrigel were seeded in a 6-well plate and the medium was replenished every 3 days. The gel drops containing cells were imaged on days 0 and 7 using an inverted microscope (Leica). The invading cells extending outside of the drop was measured by ImageJ software. For wound healing assays, $3 \times 10^6$ PC-3 cells were seeded in 6-well plates and incubated overnight to allow cells to attach. One linear scratch was generated in each well using 200 µl pipette tips once a 100% confluency of adherent cells was achieved. The wound area was imaged at 0 and 15 h and quantified using ImageJ software. The percentage of wound area was graphed at 15 h.

## Generation of recombinant D1SP protein

The D1SP sequence was generated by fusing a signal peptide sequence, human PlexinD1 Sema domain (aa 49–547) and PSI domain (aa 548–601), a G4S linker, and a hinge sequence to human IgG1 followed by codon optimization and synthesized by Biointron Biological. The target sequence was cloned into pcDNA3.4 vector at *Not*I/*Xba*I sites and expressed in CHO-K1 cells. The recombinant D1SP protein was subtracted from the supernatant and purified with Protein A resins. The QC analysis of the purified protein was conducted by SDS-PAGE, SEC-HPLC, and endotoxin test.

## Analyses of PDX-derived organoid growth

The LuCaP 147CR, 49, and 173.1 PDX models were obtained from the University of Washington. Male 4- to 6-week-old male NSG mice were purchased from Jackson Laboratory and used to expand PDX tumor tissue in vivo. The LuCaP 147CR tumor tissue was cut into ~5–10 mm³ pieces and implanted subcutaneously into both flanks of mice receiving surgical castration two weeks before implantation, while the LuCaP 173.1 and 49 tumor pieces were implanted subcutaneously into intact mice. Six weeks later, tumors were collected from the corresponding donor mice and subjected to digestion with collagenase type I in advanced DMEM/F12 medium supplemented with 10 µM Rho-associated coiled-coil containing protein kinase (ROCK) inhibitor Y-27632. Resuspend pellets were digested with TrypLE enzyme in the presence of Y-27632 for ~15 mins at 37 °C and passed through a 40-µm cell strainer to remove tissue debris and obtain single-cell suspensions. Cells were then seeded in 48-well plates ($1 \times 10^4$ cells/well) with growth factor-reduced Matrigel for organoid culture following a published protocol (Ning et al, 2022). For determining D1SP effect on organoid growth, established organoids were cultured in the complete medium containing 1 µM D1SP or PBS as control for 10 days with medium replenished every 2–3 days. The viability of organoids was measured using a Calcein-AM dye-based LIVE/DEAD Viability/Cytotoxicity Kit (Thermo Fisher Scientific) following the manufacturer's instructions.

## Animal studies

All animal studies received prior approval from the Institutional Animal Care and Use Committee (IACUC) at Washington State University (No. 6635) and complied with IACUC recommendations. Male 5-week-old NGS or athymic nude mice were purchased from Jackson Laboratory, housed in the animal research facility at Washington State University, and fed a normal chow diet. For determining the effect of PlexinD1 on PCa tumor development and growth, $4 \times 10^6$ control and PlexinD1-overexpressing/knockdown tumor cells were mixed 1:1 with Matrigel for bilateral subcutaneous injection into NSG or nude mice. The tumor length (L), width (W), and height (H) were measured every 2–3 days, and tumor volumes were calculated as ½ L × W × H. At the endpoint, tumors were collected, weighed, and processed for subsequent analyses. For determining the effect of PlexinD1 on PCa tumor metastasis, $1 \times 10^6$ control and PlexinD1-overexpressing/knockdown tumor cells stably expressing Luc and RFP were injected into the left ventricle of NSG mice for rapid development of tumor metastasis. Whole-body bioluminescence from mice was measured weekly after the injection by an IVIS SpectrumCT In Vivo Imaging System (PerkinElmer). At the endpoint, after mice were sacrificed, bone and visceral organs were harvested, exposed to fluorescence imaging using the IVIS instrument, and processed for subsequent analyses. For determining the effect of D1SP on CRPC tumor growth, $4 \times 10^6$ 22Rv1 cells were mixed 1:1 with Matrigel for bilateral subcutaneous injection in nude mice. When palpable tumors were formed about one week after tumor inoculation, mice were given 30 µl of D1SP at the dose of 1 µg/µl via intratumoral injection into tumors grown on the right flanks, or 30 µl of PBS intratumorally delivered into tumors grown on the left flanks as a paired control, 2–3 times a week. The tumor volumes were measured every 2–3 days and calculated as described above. At the endpoint, tumors were collected, weighed, and processed for subsequent analyses.

## Proximity ligation assays

Cells were seeded on chamber slides, fixed with 4% formaldehyde for 10 min at room temperature after receiving different treatments as needed, and washed twice with PBS. Different primary antibodies were incubated in blocking solution at 4 °C overnight. Assays were then performed with the Duolink In Situ Red Starter Kit Mouse/Rabbit (Duolink, Sigma-Aldrich) according to the manufacturer's instructions using anti-mouse MINUS and

anti-rabbit PLUS proximity ligation assay probes. Images were acquired by a Zeiss Axio Imager M2 upright microscope using a ×40 objective and analyzed for cytoplasmic fluorescence per cell with HALO software (Indica Labs).

## Analysis of *PLXND1* promoter

Site-directed mutagenesis was used to mutate the *PLXND1* AREs inserted in the pGL4.26 vector, with a WT construct used as a template. Mutagenesis was carried out using a QuickChange II XL Site-Directed Mutagenesis Kit (Agilent Technologies) following the manufacturer's instructions. The primer sequences used for mutating the individual *PLXND1* AREs were forward 5′-GGCAAGTGCTACTGGTTTTCCATC-CATAGTAATGATGTAAAGCTTCCTTGA-3′ and reverse 5′-TCAA GGAAGCTTTACATCATTACTATGGATGGAAAACCAGTAGCACT TGCC-3′ (ARE1), forward 5′-CTTGCCAAAACCAGAAAGTAACGGC AAATGGGAAGAAAACAAAGCGCTATCAGTACAT-3′ and reverse 5′-ATGTACTGATAGCGCTTTGTTTTCTTCCCATTTGCCGTTACT TTCTGGTTTTGGCAAG-3′ (ARE2), and forward 5′-GTACAAAGA AAACAAAGCGCTATCAATGGGTTCTAGCACGCTGATGTCAAC AGCC-3′ and reverse 5′-GGCTGTTGACATCAGCGTGCTAGAACCC ATTGATAGCGCTTTGTTTTCTTTGTAC-3′ (ARE3). Mutated nucleotides were verified by DNA sequencing prior to experimental use.

## Luciferase reporter assays

For determining the effect of AR on a *PLXND1* promoter-Luc reporter, LNCaP cells were transfected with WT or mutant *PLXND1* ARE-Luc together with pRL-TK and treated with 10 nM R1881 for 6 h. Cells were then harvested and cell lysates were assayed for relative luciferase activity by a Dual-Luciferase Reporter Assay (Promega) following the manufacturer's instructions. For determining the effects of Sema3C/Sema3E, PlexinD1, and ErbB3/cMet on a Gli-Luc reporter, cells were co-transfected with Gli-Luc and pRL-TK followed by treatment with *SEMA3C/SEMA3E* siRNA, Sema3C/Sema3E recombinant proteins, an ErbB3 neutralizing antibody, or SGX-523 in the absence or presence of *PLXND1* siRNA, or additional transfection of an ErbB3 or cMet expression construct along with *PLXND1* siRNA, followed by determination of relative luciferase activity within a 48-h incubation period.

## Chromatin immunoprecipitation-qPCR assays

ChIP-qPCR assays were used to determine the association of endogenous AR and H3K9ac proteins with the AREs identified on the *PLXND1* promoter region in LNCaP cells, which were cultured in phenol red-free medium containing 2% CSS for 24 h and then exposed to R1881 or ethanol for another 24 h, by a SimpleChIP Enzymatic Chromatin IP Kit (Cell Signaling) according to the manufacturer's instructions. Briefly, the chromatin was crosslinked with nuclear proteins, enzymatically digested with micrococcal nuclease followed by sonication, and immunoprecipitated with anti-AR, anti-H3K9ac, or a normal IgG as a negative control for IP. The immunoprecipitates were pelleted with agarose beads, purified, and subjected to qPCR with primers specifically targeting the ARE-centric *PLXND1* promoter region or a known AR-bound *KLK3* promoter region as a positive control. Primer sequences for the ARE-encompassing *PLXND1* promoter region were forward 5′-CCAGA

AAGTAACGGCAAGTA-3′ and reverse 5′-GTCTCAAACTCAGGGCT GTT-3′. Primer sequences for the AR-bound *KLK3* promoter region were forward 5′-GCCTGGATCTGAGAGAGATATCATC-3′ and reverse 5′-ACACCTTTTTTTTTTCTGGATTGTTG-3′.

## ALDH assays

An ALDEFLUOR kit (StemCell Technologies) optimized for interaction with human ALDH1A1 was used to identify ALDH1A1+ cells following the manufacturer's instructions. Briefly, the brightly fluorescent ALDH1A1-expressing cells were detected using a flow cytometer (Beckman Coulter). Side-scatter and forward-scatter profiles were used to reduce cell doublets. Specific ALDH1A1 activity was determined based on the difference between the presence and absence of the ALDEFLUOR inhibitor diethylaminobenzaldehyde (DEAB).

## Co-immunoprecipitation assays

PlexinD1-ErbB3 and PlexinD1-cMet interaction in 22Rv1 cells was determined by co-IP assays using Protein G magnetic beads. Briefly, cells were lysed in ice-cold IP lysis/wash buffer. One milligram of total protein lysates was incubated in IP lysis/wash buffer containing a PlexinD1 antibody or a control IgG at 4 °C overnight. Twenty-five microliters of Protein G magnetic beads were added into the cell lysate solution followed by a 2-h incubation at 4 °C. After incubation, the beads were washed 5 times with the IP lysis/wash buffer, and proteins were eluted from immunoprecipitates and then subjected to immunoblots with antibodies against PlexinD1, ErbB3, or cMet. For determining direct interaction between PlexinD1 and ErbB3, 10 μg each of PlexinD1 and ErbB3 recombinant proteins were co-incubated in 300 μl of IP lysis/wash buffer at 4 °C for 4 h. A PlexinD1 antibody was incubated with Protein G magnetic beads at room temperature for 2 h. The antibody-bound beads were then added to the protein mixture and incubated overnight at 4 °C. Subsequently, the beads were washed five times with IP lysis/wash buffer, and bound proteins were eluted for immunoblots with antibodies specific to PlexinD1 and ErbB3. For determining D1SP interaction with PlexinD1 and its ligands, 10 μg each of PlexinD1, Sema3C, or Sema3E proteins were incubated with 10 μg D1SP in IP buffer at 4 °C for 4 h. The protein mixture was then incubated with Fc-antibody bound beads as described above, with proteins eluted for immunoblots with antibodies against PlexinD1, Sema3C, or Sema3E.

## Immunohistochemical and immunofluorescence analyses

IHC analyses of clinical specimens and xenograft tumors and IF analysis of cultured cells were performed using antibodies against PlexinD1, Ki-67, SYP, p-ErbB3, p-ErbB2, p-cMet, Gli1, or α-tubulin following a published protocol (Wu et al, 2014). Serial sections of TMAs were used for IHC staining of two proteins separately as in Fig. 6G. The H-scores for PlexinD1 and SYP IHC staining in clinical specimens, percentages of Ki-67+ cells and per-cell intensity of PlexinD1, p-ErbB3, p-ErbB2, and p-cMet IHC staining in xenograft tumors, and intensity of nuclear Gli1

## The paper explained

**Problem**

Aggressive variants of prostate cancer, especially castration-resistant prostate cancer after failure of hormone therapy, often result in treatment resistance, metastasis, and death, lacking a cure or effective treatments.

**Results**

We identified the axon guidance semaphorin receptor PlexinD1 as a crucial driver of the aggressive phenotype in metastatic castration-resistant prostate cancer cells. High PlexinD1 expression is manifested in human prostate tumors associated with unfavorable clinical outcomes. PlexinD1 induces a variety of malignant behaviors in castration-resistant prostate cancer in vitro and in vivo, and confers cellular stemness and plasticity for evading therapy. Mechanistically, PlexinD1 activates an ErbB3/cMet-ERK/AKT-noncanonical Hh/Gli1 signaling cascade to facilitate prostate cancer growth and plasticity. Importantly, a PlexinD1 decoy receptor, D1SP, was shown to effectively restrict castration-resistant prostate cancer growth in multiple preclinical experimental models.

**Impact**

Our findings suggest PlexinD1 as a potentially attractive and actionable therapeutic target against a wide range of aggressive and lethal prostate cancer variants. Our work also provides the strong mechanistic rationale and pharmacological evidence for further developing D1SP as a potential prostate cancer therapy.

expression in PCa cells were analyzed by HALO software after areas of interest were defined using manual tissue segmentation.

## RNA-seq, ChIP-seq, scRNA-seq, and bioinformatic analyses

The total RNA of LNCaP and C4-2B$^{ENZR}$ cells, control and PlexinD1-knockdown C4-2B$^{ENZR}$ and 22Rv1 cells, and control and PlexinD1-overexpressing LNCaP cells was extracted by a RNeasy Mini Kit and underwent DNase digestion following the manufacturer's instructions. RNA-seq was performed on an Illumina HiSeq 4000 at Novogene. Bowtie 2 v2.1.0 was used for mapping to the human genome hg19 transcript set. RSEM v1.2.15 was used to calculate the count and estimate the gene expression level. Trimmed mean of M-values (TMM) method in the edge R package was used for gene expression normalization. The PCa clinical datasets used for examination of PlexinD1 expression between different disease states and correlation studies were downloaded from Oncomine, cBioPortal, and Gene Expression Omnibus (GEO) databases. The AR ChIP-seq dataset (GSE125245) was downloaded from the GEO database. Integrative Genomics Viewer v2.13.2 was used to visualize AR-binding peaks within *PLXND1* genomic region. The scRNA-seq dataset (GSE137829) was downloaded from the GEO database and analyzed for individual cell's NE score based on the expression levels of 14 NE genes (*ASCL1*, *CHGA*, *CHGB*, *SYP*, *FOXA2*, *NKX2-1*, *ENO2*, *MYCN*, *POU3F2*, *NCAM1*, *INSM1*, *EZH2*, *SOX2*, and *SIAH2*) using the ssGSEA (single sample GSEA) method which is implemented in the GSVA R package. GSEA v4.3.3 was used to evaluate the enrichment of different gene sets from the molecular signature database (MSigDB v2023.2 Hs) or curated based on related studies (Wang et al, 2022).

## Statistical analysis

Sample sizes were determined based on previous publications. No data were excluded from the analysis. No randomization procedures were needed and thus used for allocating mice to treatment. The investigators were not blinded to allocation during experiments and outcome assessment. Statistical analysis was performed with Graph-Pad. Comparisons between Kaplan-Meier curves were analyzed using the log-rank test. Correlations between groups were determined by Pearson correlation. All other comparisons were analyzed by paired and unpaired two-tailed Student's $t$ test, one-way analysis of variance (ANOVA) with Tukey's or Dunnett's multiple-comparison test, two-way ANOVA with Tukey's or Sidak's multiple-comparison test, or chi-square test. A $P$ value less than 0.05 was considered statistically significant. Exact $P$ values are listed in Appendix Table S4.

## Data availability

The RNA-seq data generated in this study are available in the NCBI GEO database with accession number GSE253766.

The source data of this paper are collected in the following database record: biostudies:S-SCDT-10_1038-S44321-024-00186-z.

## Peer review information

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

## Acknowledgements

This work was supported by Department of Defense Prostate Cancer Research Program grant W81XWH-19-1-0279, NIH/NCI grants R37CA233658, R01CA258634 and R01CA279528, and WSU startup funds to BJW; and Department of Defense Prostate Cancer Research Program grants W81XWH-18-2-0013, W81XWH-18-2-0015, W81XWH-18-2-0016, W81XWH-18-2-0017, W81XWH-18-2-0018, and W81XWH-18-2-0019 to the Prostate Cancer Biorepository Network. The establishment and characterization of LuCaP PDX models were supported by the Pacific Northwest Cancer SPORE (P50CA097186), the Department of Defense Prostate Cancer Biorepository Network (W81XWH-14-2-0183), and NCI grant (P01CA163227). We gratefully acknowledge resources and support from the Flow Cytometry, Histology, Imaging, and Microscopy core facilities at WSU Health Sciences in Spokane. We thank Gary Mawyer for editorial assistance.

## Author contributions

**Jing Wei**: Conceptualization; Data curation; Formal analysis; Investigation; Methodology; Writing—review and editing. **Jing Wang**: Conceptualization; Data curation; Formal analysis; Investigation; Methodology; Writing—review and editing. **Wen Guan**: Data curation; Formal analysis. **Jingjing Li**: Investigation; Methodology. **Tianjie Pu**: Investigation; Methodology. **Eva Corey**: Resources; Writing—review and editing. **Tzu-Ping Lin**: Resources. **Allen C Gao**: Resources. **Boyang Jason Wu**: Conceptualization; Supervision; Funding acquisition; Investigation; Writing—original draft; Project administration; Writing—review and editing.

Source data underlying figure panels in this paper may have individual authorship assigned. Where available, figure panel/source data authorship is listed in the following database record: biostudies:S-SCDT-10_1038-S44321-024-00186-z.

## Disclosure and competing interests statement

EC received sponsored research funding from Sanofi, Gilead, AbbVie, Genentech, Janssen Research, Astra Zeneca, GSK, Bayer Pharmaceuticals, Forma Pharmaceuticals, Foghorn, Kronos, and MacroGenis. The other authors declare no competing interests.

