## [Peer Review File · EMBO Molecular Medicine]

PlexinD1 is a Driver and a Therapeutic Target in Advanced Prostate Cancer

Jing Wei, Jing Wang, Wen Guan, Jingjing Li, Tianjie Pu, Eva Corey, Tzu-Ping Lin, Allen gao, and Boyang Wu

Corresponding authors: Boyang Wu (boyang.wu@wsu.edu) , Jing Wang (jing_wang1@dfci.harvard.edu)

Review Timeline:

Submission Date:	9th Apr 24
Editorial Decision:	30th Apr 24
Revision Received:	21st Nov 24
Editorial Decision:	2nd Dec 24
Editor's Correspondence:	5th Dec 24
Authors' Correspondence:	5th Dec 24
Correspondence:	6th Dec 24
Revision Received:	11th Dec 24
Accepted:	

Editor: Lise Roth

Transaction Report:

30th Apr 2024

Dear Dr. Wu,

We have now heard back from the referees who agreed to evaluate your manuscript. As you will see below, the reviewers raise substantial concerns on your work, which unfortunately preclude its publication in EMM in its current form.

As you will see below, the reviewers find that the question addressed by the study is of potential interest, however they remain unconvinced that some of the major conclusions are sufficiently supported by the data and raise partially overlapping concerns related to the mechanism at play, the potential role of ligands/semaphorins, the unknown effects of PlexinD1 targeting on non-tumorigenic or non-CRPC cells, and discrepancies with previously published work.

If you feel you can satisfactorily address all points listed by the referees, you may wish to submit a revised version of your manuscript. After discussion with my colleagues, we nevertheless agreed that while the comment from referee #1 on intratumoral vs. systemic injection is very relevant, you could address it by discussing the translational limitations of your work and toning down your conclusions.

Please attach a covering letter giving details of the way in which you have handled each of the points raised by the referees. A revised manuscript will once again be subject to review, and we cannot guarantee at this stage that the eventual outcome will be favorable.

We are expecting your revised manuscript within three to six months, if you anticipate any delay, please contact us.

We require:

- 1) A .docx formatted version of the manuscript text (including legends for main figures, EV figures and tables). Please make sure that the changes are highlighted to be clearly visible.
- 2) Individual production quality figure files as .eps, .tif, .jpg (one file per figure). For guidance, download the 'Figure Guide PDF' (<https://www.embopress.org/page/journal/17574684/authorguide#figureformat>).
- 3) At EMBO Press we ask authors to provide source data for the main figures. Our source data coordinator will contact you to discuss which figure panels we would need source data for and will also provide you with helpful tips on how to upload and organize the files.
- 4) A .docx formatted letter INCLUDING the reviewers' reports and your detailed point-by-point responses to their comments. As part of the EMBO Press transparent editorial process, the point-by-point response is part of the Review Process File (RPF), which will be published alongside your paper.
- 5) A complete author checklist, which you can download from our author guidelines (<https://www.embopress.org/page/journal/17574684/authorguide#submissionofrevisions>). Please insert information in the checklist that is also reflected in the manuscript. The completed author checklist will also be part of the RPF.
- 6) Please note that all corresponding authors are required to supply an ORCID ID for their name upon submission of a revised manuscript.
- 7) It is mandatory to include a 'Data Availability' section after the Materials and Methods. Before submitting your revision, primary datasets produced in this study need to be deposited in an appropriate public database, and the accession numbers and database listed under 'Data Availability'. Please remember to provide a reviewer password if the datasets are not yet public (see <https://www.embopress.org/page/journal/17574684/authorguide#dataavailability>). In case you have no data that requires deposition in a public database, please state so in this section. Note that the Data Availability Section is restricted to new primary data that are part of this study.
- 8) For data quantification: please specify the name of the statistical test used to generate error bars and P values, the number

(n) of independent experiments (specify technical or biological replicates) underlying each data point and the test used to calculate p-values in each figure legend. The figure legends should contain a basic description of n, P and the test applied. Graphs must include a description of the bars and the error bars (s.d., s.e.m.). Please provide exact p values.

13) Author contributions: CRedit has replaced the traditional author contributions section because it offers a systematic machine readable author contributions format that allows for more effective research assessment. Please remove the Authors Contributions from the manuscript and use the free text boxes beneath each contributing author's name in our system to add specific details on the author's contribution. More information is available in our guide to authors.

16) As part of the EMBO Publications transparent editorial process initiative (see our Editorial at <http://embomolmed.embopress.org/content/2/9/329>), EMBO Molecular Medicine will publish online a Review Process File (RPF) to accompany accepted manuscripts.

In the event of accepted acceptance, this file will be published in conjunction with your paper and will include the anonymous referee reports, your point-by-point response and all pertinent correspondence relating to the manuscript. Let us know whether you agree with the publication of the RPF and as here, if you want to remove or not any figures from it prior to publication. Please note that the Authors checklist will be published at the end of the RPF.

EMBO Molecular Medicine has a "scooping protection" policy, whereby similar findings that are published by others during

review or revision are not a criterion for rejection. Should you decide to submit a revised version, I do ask that you get in touch after three months if you have not completed it, to update us on the status.

I look forward to receiving your revised manuscript.

Yours sincerely,

Lise Roth

***** Reviewer's comments *****

Referee #1 (Comments on Novelty/Model System for Author):

The study is interesting, novel, and potentially relevant for translational medicine. However, a number of mechanistic aspects should be elucidated with new experiments (detailed below) in order to consolidate the proposed working model and promote the applicability of these findings in future preclinical or clinical studies.

Referee #1 (Remarks for Author):

In this study, Wang et al. have studied the role of semaphorin receptor PlexinD1 in advanced prostate cancer. PlexinD1 was previously reported to mediate cancer progression, in different tumor types and experimental models, in response to its main ligand Sema3E or other semaphorin family members. Here the authors found PlexinD1 (as well as its ligands) to be upregulated in a transcriptomic screening comparing parental androgen-dependent LnCaP cells and their enzalutamide-resistant derivatives. PlexinD1 expression was further studied in human prostate cancer samples, and its mechanistic role was investigated in prostate cancer cell lines, by exploiting in vitro experiments and in vivo murine models. From the mechanistic point of view, PlexinD1 signaling has been previously associated with ErbB2 activation; here the authors found ErbB2 and ErbB3 regulation by PlexinD1 leading to a non-canonical activation of the Hedgehog pathway.

The study is very interesting, providing evidence for a previously unknown mechanism responsible for enzalutamide resistance in prostate cancer cells. However, some mechanistic aspects need to be clarified. Above all the role of PlexinD1 ligands, signaling in autocrine and paracrine manner in the tumor microenvironment.

1. Data in Figure 3 suggest that androgen receptor signaling suppresses PLXND1 expression at promoter level. Since this receptor is crucially regulated by the availability of semaphorin ligands, it would be important to clarify if androgens also regulate the transcription of e.g. Sema3E or Sema3C (found to be upregulated as well in enzalutamide-resistant cells).
2. Figure 4 shows that the proliferation of prostate cancer cells is dependent on PlexinD1 expression. Surprisingly, this also applies to LnCaP cells, reported to express very low PlexinD1. This finding requires clarification, since PlexinD1 is furthermore expressed by a variety of normal cells and tissues in the body and a widespread inhibition of cell viability would not be advisable in translational perspective.
3. On the same line, the impact of PlexinD1-decoy receptor D1-SP on proliferation (validated in Fig. 9 of this study) should also be investigated in LnCaP cells as well as in non-tumoral cells (e.g. HUVEC), by comparison to enzalutamide-resistant prostate cancer cells. Moreover, cell growth inhibition should be documented by dose-response and time-course experiments (beyond what minimally shown in Fig. 9D).
4. Further concerning Fig. 9D, the results indicate a strong basal effect of PlexinD1-decoy soluble receptor, which is consistent with the blockade of endogenous autocrine signaling circuits mediated by PlexinD1 ligands. However, the role of such PlexinD1-activating molecules should be necessary addressed by complementary gene silencing approaches, at least targeting the main known ligands Sema3E and Sema3C.
5. Intratumoral injection of high-dose D1-SP is not an ideal approach to prove its amenability for pharmacological intervention in

vivo and rule out potential adverse effects or signs of toxicity. Intraperitoneal injection could represent an alternative approach, previously validated in literature.

6. PlexinD1 overexpression in LnCaP cells enhanced migration/invasion (Fig. 5B), as well as cell proliferation (Fig. 8J) in vitro. Moreover, since PlexinD1-overexpressing LnCaP are shown to upregulate stem cell markers and tumorsphere forming ability (in Fig. 6), it would be relevant to assay whether they also exhibit enhanced tumorigenic and metastatic capacity in mice.

7. PlexinD1 signaling in androgen-independent prostate cancer cells is found to be mediated by ErbB3 signaling. Since ErbB3 is known to act in association with ErbB2, which was previously found to be activated by PlexinD1 in other cancer cells (Casazza et al., 2010), the potential involvement of this oncogenic tyrosine receptor in androgen-independent prostate cancer should be mechanistically elucidated. Most of all, by testing ErbB2 kinase inhibitory drugs and antibodies commonly applied in the clinical setting, which could be particularly relevant in a translational perspective.

8. The authors postulate that upregulated PlexinD1 can activate ErbB3 and Met receptors in a RTK cognate ligand-independent manner, as previously reported in literature. However, even in the presence of plexin-RTK complexes, prior studies consistently show RTK activation in response to plexin ligands, the semaphorins (including in the case of PlexinD1). This aspect is almost totally neglected in the present manuscript, while it should be thoroughly investigated. It is not sufficient to show a modest semaphorin-dependent increase in PlexinD1-RTK proximity (as in Fig. 7E) to speak about Sema-dependent transactivation of RTKs (line 522). Instead, the authors should assay the impact of PlexinD1 ligands on ErbB3 and/or Met phosphorylation, and downstream Hedgehog pathway activation, in prostate cancer cells.

9. Unlike what is stated in the manuscript, the evidence of D1-SP blocking PlexinD1 interaction with its ligands (as provided in Fig. 9C) is not particularly clear or convincing in quantitative terms. It is recommended, first, to validate D1-SP as a decoy receptor, capable of specifically capturing recombinant PlexinD1 ligands in solution. Next, a (previously validated) ligand-induced PlexinD1 functional assay should be used to confirm the inhibitory activity of D1-SP.

10. Data shown in Fig. 9J need to be quantified and associated with statistical analysis.

Referee #2 (Comments on Novelty/Model System for Author):

Various xenograft models used.

Referee #2 (Remarks for Author):

In this study, the authors show consistent data supporting the hypothesis that PlexinD1 is a driver and a potential therapeutic target for advanced PCa, including neuroendocrine prostate cancer (NEPC). Using a variety of in vitro and in vivo models, plexinD1 was shown to promote PCa cell growth, migration and stemness. These effects were shown to be mediated by PlexinD1 activation of ErbB3 and cMet, which trigger ERK/AKT pathways and noncanonical activation of Gli1. Findings were supported by patient data that showed correlation between higher PlexinD1 expression and adverse outcomes. Furthermore, the authors developed a PlexinD1 inhibitor (D1SP) which restricted CRPC growth in preclinical models. Overall, the manuscript provides convincing data linking PlexinD1 to advanced prostate cancer and is a potential therapeutic target. Attention to the following experiments would strengthen the impact of this manuscript:

1. Effects of targeting PlexinD1, either through genetic perturbation or pharmacologically, on non-tumorigenic or non-CRPC cells is important to understand potential overall toxic effects.

2. In figure 1, the authors show mRNA expression of PLXND1 in a panel of prostate cancer cells, it would be helpful to know the expression of the protein in this panel.

3. Conclusions from the experiment shown on figure 7G would be strengthened if data would be corroborated on LNCaP overexpressing Plexin D1.

4. The authors have shown signaling pathway initiated by Plexin D1 using shRNAs. This could have potential pitfalls as signaling is usually a rapid process, while KD of a gene takes significantly longer. Signaling through Plexin D1 should be confirmed by stimulating with the ligand. Because cells endogenously express the ligand, it might be necessary to do a previous KD of the ligand.

5. On figure 8J, it would be helpful to know the effects of GANT61 on empty vector LNCaP cells.

6. Authors should cite and discuss the following preprint:

Res Sq [Preprint]. 2024 Mar 27:rs.3.rs-4095949. doi: 10.21203/rs.3.rs-4095949/v1. Plexin D1 emerges as a novel target in the development of neural lineage plasticity in treatment-resistant prostate cancer. Chengfei Liu 1, Bo Chen, Pengfei Xu 2, Joy Yang 1, Christopher Nip 2, Leyi Wang 2, Yuqiu Shen 2, Shu Ning 1, Yufeng Shang 2, Eva Corey 3, Allen C Gao 1, Jason Gestwicki 4, Qiang Wei 5, Liangren Liu. PMID: 38585965 PMCID: PMC10996809 DOI: 10.21203/rs.3.rs-4095949/v1

Referee #3 (Comments on Novelty/Model System for Author):

This study provides robust and convincing data, to justify the claims. The work is important to the field and of interest to more than specialists (includes semaphorin/Plexin signalling, prostate cancer, androgen deprivation therapy resistance). Some of the claims are not very novel however, as a similar phenotype has been shown for the ligands of PlexinD1 (the protein concerned in the paper under review).

Similarities to previous papers:

- Sema3C (a ligand for PlexinD1) expression levels increase in castration-resistant prostate cancer (CRPC) and functions to promote cancer cell growth and resistance to androgen receptor pathway inhibition, while SEMA3C inhibition (with a decoy receptor to PlexinB1) delays CRPC and enzalutamide-resistant progression. (Peacock et al. 2018)
- Expression of Sema3E and its receptor Plexin D1 correlates with the metastatic progression of human tumors. PlexinD1 (and Sema3E) depletion reduces metastatic potential of xenografts of lung and breast cancer cell lines. (Casazza et al 2010)
- Plexins promote HH signalling at level of Gli1 (mainly concerned with A-type plexins and MEFs) (Pinskey et al. 2022). (this has not been cited)
- Sema3C promotes EMT in prostate cancer cells (Tam et al. 2017), AR transcription regulates Sema3C in a GATA-dependent manner (Tam et al. 2016). (this has not been cited)

Some other experiments would strengthen the paper, specified in comments for authors.

Referee #3 (Remarks for Author):

In this manuscript the authors provide convincing evidence for a role for PlexinD1 in the progression of CRPC. Transcription of PlexinD1 is downregulated by AR while repression of AR signalling in response to androgen deprivation therapy leads to an increase in expression. Upregulation of PlexinD1 promotes an aggressive phenotype of increased migration, invasion, proliferation, EMT, stemness and neuroendocrine differentiation and tumour growth and metastasis in vivo, in a variety of prostate cancer cell lines. These findings are backed up by clinical evidence. PlexinD1 activates ErbB3 and cMet and promotes translocation of Gli1 to the nucleus and noncanonical Hedgehog signalling. Inhibition of PlexinD1 with a decoy receptor inhibits CRPC growth, migration and invasion.

The work is important and clinically relevant. Nevertheless, the following points should be addressed:

1. Figure 1. The hypothesis is that AR down regulates PlexinD1, and so LNCaP, which expresses high AR, produces low PlexinD1, while cell lines expressing no AR (such as PC3, DU145) have higher levels of PlexinD1 mRNA. However, 22Rv1 expresses both AR and high PlexinD1 mRNA levels. How can this be explained?

In contrast to results in Figure 1 of the manuscript, Peacock et al. find that LNCaP and 22RV1 have similar levels of PlexinD1 protein and PC3 and DU145 have very low PlexinD1 expression. A western blot comparing PlexinD1, Sema3C and Sema3E protein levels in all the prostate cancer cell lines used, including LNCaP and 22RV1 would be helpful (the cell lines may differ somewhat between labs) to clarify this concern.

2. Figure 3. shows convincingly that PlexinD1 expression is suppressed by AR signalling in LNCaP and LAPC4 cells - R1881 inhibits PlexinD1 expression, while enzalutamide treatment increases expression of PlexinD1. PlexinD1 has repressive androgen response element (AREs) in its promoter.

These findings are interesting in relation to those of Tam et al. (2016) who found that R1881 increases, while enzalutamide treatment decreases, Sema3C expression (a ligand of PlexinD1), due to an activating ARE in the promoter of Sema3C.

One wonders why AR binding to the promoters of receptor and ligand genes AREs have opposite effects? In this scenario, is PlexinD1 perhaps acting in a ligand-independent manner or is it activated by Sema3E binding? This paper should be referred to and discussed.

3. The issue above raises the question as to the role of ligand binding in PlexinD1 activation. Experiments should be performed to establish this role. (ie expression vs activation)

No ligand was added in the experiments involving PlexinD1 knock down in comparison to non-silenced controls suggesting that PlexinD1 works in the absence of ligand. However, serum present in the medium contains some semaphorins and endogenous SemaE is expressed by many prostate cancer cell lines. Furthermore, ectopic overexpression of Plexins have been shown to mimic ligand binding (Giordano et al.). Peacock et al show that Sema3C activates PlexinB1 in LNCaP and that inhibition of PlexinD1 has little effect on Sema3C-induced growth). Could the effects shown in the manuscript under review, at endogenous levels of PlexinD1, be induced by Sema3E, another ligand for PlexinD1? To resolve this issue, experiments should be performed to establish the effect of Sema3E and Sema3C on the aggressive phenotype of the prostate cancer cell lines used (in serum free medium), in the presence and absence of PlexinD1.

4. Supplementary Figure 1. In contrast to the current manuscript, Peacock et al showed that siRNA-mediated silencing of Plexin D1 did not reduce proliferation of DU145.

5. Figure 6. Tam et al.(sci rep2017) show that ectopic expression of SEMA3C in RWPE-1 promotes a similar phenotype to that shown in this manuscript upon overexpression of PlexinD1 (upregulation of EMT and stem markers, cell plasticity, migration and invasion in vitro and cell dissemination in vivo). This should be referred to.

6. Figure 7. Is the interaction of PlexinD1 with ErbB3 direct? Co-IP of purified proteins would resolve this issue.

7. Figure 8H, these images are not convincing. Better images are required showing markers of cell membrane or cytoplasm in addition.

To confirm the nuclear translocation of Gli1 upon activation of PlexinD1, experiments should be performed to investigate the

effect of SemaE and Sema3C stimulation of cells on the nuclear localisation of ectopically expressed tagged-Gli1. Western blots of cell fractionation - the marker for cytoplasmic proteins should be shown in the nuclear extract and the marker for nuclear proteins should be shown for the cytoplasmic proteins, to ensure the purity of the samples. A Link between Plexins and HH signalling and Gli 1 has been shown before (Pinsky 22): 'Plexins promote Hedgehog signaling through their cytoplasmic GAP activity' - MEFs treated with siRNA oligos for Plxna3, Plxnb2, and Plxnd1 showed reduced activation of Gli1. This paper should be mentioned.

8. Figure 9.

Western blots of the purified D1SP protein from the cell lysate and conditioned medium should be shown to ensure that the protein is of the correct size.

A dose response to added D1SP protein should be shown in the RTK experiments.

How does the D1SP work? Does it bind to PlexinD1 or its ligands, or prevent RTK/PlexinD1 interaction?

As mentioned in this manuscript, Peacock et al used a similar decoy receptor. They showed that, compared to B1SP, D1SP was an ineffective inhibitor of LNCaP cell growth and that D1SP showed no inhibitory activity of SEMA3C-induced activation of RTK signaling in both LNCaP (Appendix Fig S5E) or DU145 cells. This is somewhat at odds to the new manuscript, which shows that D1SP inhibits proliferation and aggressive behaviour of 22rv1 and c4-2b enz cells and growth and ERB3 and MET phosphorylation in xenografted 22rv1 tumours. These differing results should be discussed.

Does D1SP inhibit RTK phosphorylation in 22rv1, DU14 and or LNCaP cells?

9. Discussion. The authors state that: 'overexpression of PlexinD1 without modulation of Sema ligand levels was sufficient to promote PCa cell proliferation, migration, invasion, ENZ resistance, stemness, and plasticity' - overexpression has been shown to mimic ligand binding, probably due to clustering of receptors, in in vitro ectopic expression experiments (eg Giordano et al.).

Minor changes

Figure 3I, the figure key too small

References.

- Peacock JW, Takeuchi A, Hayashi N, Liu L et al. SEMA3C drives cancer growth by transactivating multiple receptor tyrosine kinases via Plexin B1. *EMBO Mol Med.* 2018
- Tam, K.J., Hui, D.H.F., Lee, W.W. et al. Semaphorin 3 C drives epithelial-to-mesenchymal transition, invasiveness, and stem-like characteristics in prostate cells. *Sci Rep* 7, 11501 (2017). <https://doi.org/10.1038/s41598-017-11914-6>
- Tam K. J, Dalal K., Hsing M., et al Androgen receptor transcriptionally regulates semaphorin 3C in a GATA2-dependent manner. *Oncotarget.* 2017; 8: 9617-9633.
- Pinsky JM, Hoard TM, Zhao XF, et al. Plexins promote Hedgehog signaling through their cytoplasmic GAP activity. *Elife.* 2022
- Casazza A, Finisguerra V, Capparuccia L, et al. Sema3E-Plexin D1 signaling drives human cancer cell invasiveness and metastatic spreading in mice. *J Clin Invest.* 2010 Aug;120(8):2684-98.
- Giordano S, Corso S, Conrotto P, et al. The semaphorin 4D receptor controls invasive growth by coupling with Met. *Nat Cell Biol.* 2002 Sep;4(9):720-4.

We are grateful for the reviewers' overall appreciation of our work and constructive comments. We have included additional experimental data and discussion in our revision and provided a detailed point-by-point response to the reviewers' comments, including the locations of the incorporated changes highlighted in red in the revised manuscript.

Referee #1 (Comments on Novelty/Model System for Author):

"The study is interesting, novel, and potentially relevant for translational medicine. However, a number of mechanistic aspects should be elucidated with new experiments (detailed below) in order to consolidate the proposed working model and promote the applicability of these findings in future preclinical or clinical studies."

Response: We thank the reviewer for recognizing the novelty and potential translational relevance of our work and for raising very constructive comments, which increased the strength of our manuscript, and we hope our detailed response below will be satisfactory to the reviewer.

Referee #1 (Remarks for Author):

"In this study, Wang et al. have studied the role of semaphorin receptor PlexinD1 in advanced prostate cancer. PlexinD1 was previously reported to mediate cancer progression, in different tumor types and experimental models, in response to its main ligand Sema3E or other semaphorin family members. Here the authors found PlexinD1 (as well as its ligands) to be upregulated in a transcriptomic screening comparing parental androgen-dependent LnCaP cells and their enzalutamide-resistant derivatives. PlexinD1 expression was further studied in human prostate cancer samples, and its mechanistic role was investigated in prostate cancer cell lines, by exploiting in vitro experiments and in vivo murine models. From the mechanistic point of view, PlexinD1 signaling has been previously associated with ErbB2 activation; here the authors found ErbB2 and ErbB3 regulation by PlexinD1 leading to a non-canonical activation of the Hedgehog pathway."

The study is very interesting, providing evidence for a previously unknown mechanism responsible for enzalutamide resistance in prostate cancer cells. However, some mechanistic aspects need to be clarified. Above all the role of PlexinD1 ligands, signaling in autocrine and paracrine manner in the tumor microenvironment."

Response: We thank the reviewer for considering our work interesting. We performed new experiments to address the reviewer's concerns on the mechanistic aspects as detailed below.

"1. Data in Figure 3 suggest that androgen receptor signaling suppresses PLXND1 expression at promoter level. Since this receptor is crucially regulated by the availability of semaphorin ligands, it would be important to clarify if androgens also regulate the transcription of e.g. Sema3E or Sema3C (found to be upregulated as well in enzalutamide-resistant cells)."

Response: We performed new experiments demonstrating androgen regulation of the transcription and gene expression of SEMA3E and SEMA3C, as evidenced by the findings that androgen depletion with the use of charcoal-stripped serum supplemented medium increased, while treatment with the synthetic androgen R1881 decreased, SEMA3E and SEMA3C mRNA expression (**Fig. 3A**). We have added the relevant text in the revised manuscript (**page 5 line 178 – page 6 line 182**).

"2. Figure 4 shows that the proliferation of prostate cancer cells is dependent on PlexinD1 expression. Surprisingly, this also applies to LnCaP cells, reported to express very low PlexinD1. This finding requires clarification, since PlexinD1 is furthermore expressed by a variety of normal cells and tissues in the body and a widespread inhibition of cell viability would not be advisable in translational perspective."

Response: Based on the relatively low PlexinD1 expression in LNCaP cells as shown in New Fig. 1G and Appendix Fig S1A, we chose to overexpress PlexinD1 in LNCaP cells to study the function and mechanism of PlexinD1 in prostate cancer in our study. As shown in Fig. 4, we demonstrated that overexpression of PlexinD1 accelerated the proliferation and anchorage-independent colony formation of LNCaP cells compared with controls. We clarified this point in the revised manuscript (**page 8 line 259-261**).

“3. On the same line, the impact of PlexinD1-decoy receptor D1-SP on proliferation (validated in Fig. 9 of this study) should also be investigated in LnCaP cells as well as in non-tumoral cells (e.g. HUVEC), by comparison to enzalutamide-resistant prostate cancer cells. Moreover, cell growth inhibition should be documented by dose-response and time-course experiments (beyond what minimally shown in Fig. 9D).”

Response: As suggested, we performed new experiments demonstrating that D1SP inhibited the proliferation of C4-2B^{ENZR} and DU145 cells in both dose- and time-dependent manners, paralleled by no changes in the proliferation of LNCaP cells and non-tumorigenic HUVEC and RWPE-1 (a normal prostate epithelial cell line) cells (**New Appendix Figs. S13D and S13E**). This could be attributed to the relatively higher expression levels of both PlexinD1 and PlexinD1 downstream effector kinases such as cMet, as revealed in our study and others, for conveying PlexinD1-triggered mitogenic signals in CRPC cells (C4-2B^{ENZR} and DU145) compared with non-CRPC (LNCaP) and normal (HUVEC and RWPE-1) cells. We have added the relevant text in the revised manuscript (**page 16 line 560-564 and page 19 line 667-674**).

“4. Further concerning Fig. 9D, the results indicate a strong basal effect of PlexinD1-decoy soluble receptor, which is consistent with the blockade of endogenous autocrine signaling circuits mediated by PlexinD1 ligands. However, the role of such PlexinD1-activating molecules should be necessary addressed by complementary gene silencing approaches, at least targeting the main known ligands Sema3E and Sema3C.”

Response: As suggested, we performed new experiments demonstrating that siRNA-mediated knockdown of Sema3E and Sema3C reduced the proliferation of C4-2B^{ENZR} and 22Rv1 cells to a similar extent, with more reductions observed upon concurrent silencing of both Sema3 ligands. Also, we observed that parallel treatment with D1SP achieved growth inhibitory efficacy comparable to the Sema3C/3E co-silencing approach in both C4-2B^{ENZR} and 22Rv1 cells (**New Appendix Fig. S13F**). We have added the relevant text in the revised manuscript (**page 16 line 564-570**).

“5. Intratumoral injection of high-dose D1-SP is not an ideal approach to prove its amenability for pharmacological intervention in vivo and rule out potential adverse effects or signs of toxicity. Intraperitoneal injection could represent an alternative approach, previously validated in literature.”

Response: We thank the reviewer for raising this valuable point from a translational angle. We intended to provide a proof of principle demonstrating D1SP's potential in vivo anti-CRPC efficacy, using an intratumoral delivery approach due to the limited amount of D1SP we could afford for an in vivo test given both time and budget constraints. We fully recognize the translational limitations of this approach and agree that future efforts using intraperitoneal injection as previously validated by other Plexin protein inhibitors are required to rigorously and comprehensively evaluate the therapeutic benefit of D1SP1 in a translational perspective. We thus toned down our conclusions by suggesting targeted blockade of PlexinD1 via the protein inhibitor D1SP as a “potential” therapy against CRPC in both the Abstract and Discussion section of the revised manuscript (**page 2 line 49, and page 20 line 686-693 and 697**).

“6. PlexinD1 overexpression in LnCaP cells enhanced migration/invasion (Fig. 5B), as well as cell proliferation (Fig. 8J) in vitro. Moreover, since PlexinD1-overexpressing LnCaP are shown to upregulate stem cell markers and tumorsphere forming ability (in Fig. 6), it would be relevant to assay whether they also exhibit enhanced tumorigenic and metastatic capacity in mice.”

Response: As suggested, we performed new sets of experiments to study the effect of PlexinD1 overexpression in LNCaP cells on tumorigenic and metastatic capacity in mice. As shown in **New Appendix Figs. S5A-S5E**, overexpression of PlexinD1 in LNCaP cells led to more tumors formed (PlexinD1: 8 tumors/8 injection sites vs. Vector: 6 tumors/8 injection sites), faster tumor growth, higher tumor weight, and increased percentages of tumor-expressing Ki-67+ cells in s.c. xenografted mice compared with controls. As shown in **New Appendix Figs. S7A-S7D**, overexpression of PlexinD1 in Luc-tagged LNCaP cells (known to have inherently weak metastatic potential) resulted in higher metastasis incidence and greater tumor burden when inoculated intracardially into mice compared with controls, as determined by bioluminescence imaging. At the endpoint, necropsy and ex vivo bioluminescence imaging analysis confirmed more prevalent distant metastasis, primarily in the liver and jawbone, caused by PlexinD1-overexpressing tumor cells relative to controls. We have added the relevant text in the revised manuscript (**page 8 line 275-278 and page 9 line 309-314**).

“7. PlexinD1 signaling in androgen-independent prostate cancer cells is found to be mediated by ErbB3 signaling. Since ErbB3 is known to act in association with ErbB2, which was previously found to be activated by PlexinD1 in other cancer cells (Casazza et al., 2010), the potential involvement of this oncogenic tyrosine receptor in androgen-independent prostate cancer should be mechanistically elucidated. Most of all, by testing ErbB2 kinase inhibitory drugs and antibodies commonly applied in the clinical setting, which could be particularly relevant in a translational perspective.”

Response: As suggested, we performed new experiments demonstrating that the ErbB2 kinase inhibitory drug (tucatinib) and antibody (trastuzumab) as used in the clinic had similar effects to ErbB3 inactivation on PlexinD1-dependent cell proliferation and lineage marker expression. As shown in **New Appendix Fig. S11A**, we demonstrated that prior treatment with tucatinib or trastuzumab abolished the suppressive effect of PlexinD1 knockdown on the proliferation of 22Rv1 cells compared with controls. Conversely, we found that tucatinib or trastuzumab treatment reduced PlexinD1 overexpression-induced proliferation of LNCaP cells to control levels (**New Appendix Fig. S11B**). We also demonstrated that PlexinD1 knockdown failed to repress the mRNA expression of stemness, basal cell, and neuroendocrine (NE) markers upon pre-treatment with tucatinib or trastuzumab in 22Rv1 cells (**New Appendix Fig. S11C**). Further, we showed that interfering with ErbB2 with tucatinib or trastuzumab reversed PlexinD1-activated mRNA expression of stemness, basal cell, NE, and EMT markers in LNCaP cells (**New Appendix Fig. S11D**). We have added the relevant text in the revised manuscript (**page 12 line 426-430 and page 13 line 432-440**).

“8. The authors postulate that upregulated PlexinD1 can activate ErbB3 and Met receptors in a RTK cognate ligand-independent manner, as previously reported in literature. However, even in the presence of plexin-RTK complexes, prior studies consistently show RTK activation in response to plexin ligands, the semaphorins (including in the case of PlexinD1). This aspect is almost totally neglected in the present manuscript, while it should be thoroughly investigated. It is not sufficient to show a modest semaphorin-dependent increase in PlexinD1-RTK proximity (as in Fig. 7E) to speak about Sema-dependent transactivation of RTKs (line 522). Instead, the authors should assay the impact of PlexinD1 ligands on ErbB3 and/or Met phosphorylation, and downstream Hedgehog pathway activation, in prostate cancer cells.”

Response: As suggested, we performed new experiments to examine the impact of PlexinD1 ligands, Sema3C and Sema3E, on ErbB3/cMet and downstream Hedgehog pathway in prostate cancer cells. We demonstrated that siRNA-mediated knockdown of Sema3C or Sema3E reduced, while addition of recombinant Sema3C or Sema3E proteins enhanced, the phosphorylation of ErbB3, cMet, and their associated kinases including ErbB2, ERK, and AKT in both C4-2B^{ENZR} and 22Rv1 cells (**New Appendix Figs. S9A and S9B**). To determine the effect of PlexinD1 ligands on Gli1 transcriptional activity, we showed that siRNA-mediated knockdown of Sema3C or Sema3E reduced, while treatment with recombinant Sema3C or Sema3E proteins enhanced, Gli-Luc reporter activity as well as expression of several Gli1 target genes (**New Appendix Figs. S12A and S12B**). To determine the effect of PlexinD1 ligands on Gli1 nuclear localization, we showed that siRNA-mediated knockdown of Sema3C or Sema3E decreased, while

treatment with recombinant Sema3C or Sema3E proteins stimulated, nuclear accumulation of endogenous Gli1 protein in C4-2B^{ENZ^R} cells by an immunofluorescence assay (**New Appendix Fig. S12C**). We further showed that treating cells with SEMA3C/SEMA3E siRNA or recombinant proteins respectively repressed or enhanced the expression levels of Gli1 protein in the nuclei of C4-2B^{ENZ^R} cells (**New Appendix Fig. S12E**). We have added the relevant text in the revised manuscript (**page 12 line 402-405, page 13 line 460-463, page 14 line 492-495, page 15 line 505-507, and page 18 line 624-626**).

“9. Unlike what is stated in the manuscript, the evidence of D1-SP blocking PlexinD1 interaction with its ligands (as provided in Fig. 9C) is not particularly clear or convincing in quantitative terms. It is recommended, first, to validate D1-SP as a decoy receptor, capable of specifically capturing recombinant PlexinD1 ligands in solution. Next, a (previously validated) ligand-induced PlexinD1 functional assay should be used to confirm the inhibitory activity of D1-SP.”

Response: As suggested, we performed new experiments to more rigorously examine the disruptive effect of D1SP on PlexinD1 interaction with its ligands. As shown in **New Appendix Fig. S13A**, we mixed D1SP with recombinant Sema3C or Sema3E proteins in solution and demonstrated direct binding of D1SP to both Sema3s by co-IP assays using an IgG FC-specific antibody to precipitate D1SP out of solution. Next, we showed that D1SP dose-dependently restored Sema3E-repressed migration of HUVEC endothelial cells, a previously established measurement of ligand-induced PlexinD1-dependent function (refer to Figs. 5A and 5B, Casazza et al., J Clin Invest, 2010, 120:2684-2698, PMID:20664171) (**New Appendix Fig. S13C**). We have added the relevant text in the revised manuscript (**page 16 line 546-548 and 550-552**).

“10. Data shown in Fig. 9J need to be quantified and associated with statistical analysis.”

Response: We added quantification and statistical analysis of IHC staining data for Ki-67, p-ErbB3, p-ErbB2, and p-cMet as **New Fig. 9L** and included relevant text in the revised manuscript (**page 17 line 588-590**).

Referee #2 (Comments on Novelty/Model System for Author):

“Various xenograft models used.”

Response: We appreciate the reviewer’s recognition of this strength.

Referee #2 (Remarks for Author):

“In this study, the authors show consistent data supporting the hypothesis that PlexinD1 is a driver and a potential therapeutic target for advanced PCa, including neuroendocrine prostate cancer (NEPC). Using a variety of in vitro and in vivo models, plexinD1 was shown to promote PCa cell growth, migration and stemness. These effects were shown to be mediated by PlexinD1 activation of ErbB3 and cMet, which trigger ERK/AKT pathways and noncanonical activation of Gli1. Findings were supported by patient data that showed correlation between higher PlexinD1 expression and adverse outcomes. Furthermore, the authors developed a PlexinD1 inhibitor (D1SP) which restricted CRPC growth in preclinical models. Overall, the manuscript provides convincing data linking PlexinD1 to advanced prostate cancer and is a potential therapeutic target. Attention to the following experiments would strengthen the impact of this manuscript:”

Response: We thank the reviewer for considering our data convincing and for raising very constructive comments, which increased the strength of our manuscript, and we hope our detailed response below will be satisfactory to the reviewer.

“1. Effects of targeting PlexinD1, either through genetic perturbation or pharmacologically, on non-tumorigenic or non-CRPC cells is important to understand potential overall toxic effects.”

Response: We performed new experiments to determine the effect of targeting PlexinD1 by both gene silencing and pharmacological approaches in non-tumorigenic HUVEC (a human normal endothelial cell line) and RWPE-1 (a human normal prostate epithelial cell line) cells and non-CRPC LNCaP (an androgen/AR-dependent human prostate cancer cell line) cells, along with CRPC C4-2B^{ENZ^R} cells. We demonstrated that siRNA-mediated knockdown of PlexinD1, with siRNA target sequences validated in C4-2B^{ENZ^R} and HUVEC cells (**New Appendix Fig. S3A**), did not interfere with the proliferation of HUVEC, RWPE-1, and LNCaP cells, which was paralleled with decreased proliferation of C4-2B^{ENZ^R} cells (**New Appendix Fig. S3B**). We also showed that D1SP caused no changes in the proliferation of HUVEC, RWPE-1 and LNCaP cells, in contrast to parallel dose- and time-dependent reductions in the proliferation of C4-2B^{ENZ^R} and DU145 (an AR- CRPC cell line) cells (**New Appendix Figs. S13D and S13E**). The discrepancy in the anti-proliferative efficacy of PlexinD1 blockade between CRPC and non-tumorigenic/non-CRPC cells could be due to the relatively higher expression levels of both PlexinD1 and PlexinD1 downstream effector kinases such as cMet, as revealed in our study and others, conveying PlexinD1-triggered mitogenic signals in CRPC cells (C4-2B^{ENZ^R} and DU145) compared with non-tumorigenic (HUVEC and RWPE-1) and non-CRPC (LNCaP) cells. We have added the relevant text in the revised manuscript (**page 7 line 249 – page 8 line 253, page 16 line 560-564, and page 19 line 667-674**).

“2. In figure 1, the authors show mRNA expression of PLXND1 in a panel of prostate cancer cells, it would be helpful to know the expression of the protein in this panel.”

Response: As suggested, we performed a new experiment comparing PlexinD1 protein expression in a panel of human prostate cancer cell lines as **New Fig. 1G**. Please note that unequal amounts of total cell lysates from different cell lines were loaded intentionally due to the large discrepancy in PlexinD1 protein levels among different cell lines as indicated in the figure legend. We moved old Fig. 1G into the Appendix as New Appendix Fig. S1A. We have added the relevant text in the revised manuscript (**page 4 line 141 – page 5 line 146 and page 33 line 1020-1024**).

“3. Conclusions from the experiment shown on figure 7G would be strengthened if data would be corroborated on LNCaP overexpressing Plexin D1.”

Response: As suggested, we performed new experiments demonstrating that individual blockade of ErbB3 and cMet reduced the PlexinD1-induced proliferation of LNCaP cells to control levels, which has been included in **New Fig. 7G**. We have also added relevant text in the revised manuscript (**page 13 line 432-433**).

“4. The authors have shown signaling pathway initiated by Plexin D1 using shRNAs. This could have potential pitfalls as signaling is usually a rapid process, while KD of a gene takes significantly longer. Signaling through Plexin D1 should be confirmed by stimulating with the ligand. Because cells endogenously express the ligand, it might be necessary to do a previous KD of the ligand.”

Response: We thank the reviewer for bringing out this insightful point. As suggested, we performed new experiments to examine the effect of PlexinD1 ligands, Sema3C and Sema3E, on the ErbB3/ErbB2/cMet-ERK/AKT-noncanonical Hedgehog/Gli1 signaling pathway triggered by PlexinD1 in prostate cancer cells. We demonstrated that siRNA-mediated knockdown of Sema3C or Sema3E reduced, while addition of recombinant Sema3C or Sema3E proteins enhanced, the phosphorylation of ErbB3, cMet, and their associated kinases including ErbB2, ERK, and AKT in both C4-2B^{ENZ^R} and 22Rv1 cells (**New Appendix Figs. S9A and S9B**). To determine the effect of PlexinD1 ligands on Gli1 transcriptional activity, we showed that siRNA-mediated knockdown of Sema3C or Sema3E reduced, while treatment with recombinant Sema3C or Sema3E proteins enhanced, Gli-Luc reporter activity as well as expression of several Gli1

target genes (**New Appendix Figs. S12A and S12B**). To determine the effect of PlexinD1 ligands on Gli1 nuclear localization, we showed that siRNA-mediated knockdown of Sema3C or Sema3E decreased, while treatment with recombinant Sema3C or Sema3E proteins stimulated, nuclear accumulation of endogenous Gli1 protein in C4-2B^{ENZR} cells by an immunofluorescence assay (**New Appendix Fig. S12C**). We further showed that treating cells with SEMA3C/SEMA3E siRNA or recombinant proteins respectively repressed or enhanced the expression levels of Gli1 protein in the nuclei of C4-2B^{ENZR} cells (**New Appendix Fig. S12E**). Of note, we were able to observe the above-mentioned effects of added ligands on the PlexinD1-elicited signaling pathway without prior silencing of endogenous ligands. We have added the relevant text in the revised manuscript (**page 12 line 402-405, page 13 line 460-463, page 14 line 492-495, page 15 line 505-507, and page 18 line 624-626**).

"5. On figure 8J, it would be helpful to know the effects of GANT61 on empty vector LNCaP cells."

Response: We included the control LNCaP cell groups coupled with GANT61 or cyclopamine treatment in **New Fig. 8J**. We demonstrated no changes in the proliferation of PlexinD1-low LNCaP cells in response to both cyclopamine and GANT61, while GANT61 but not cyclopamine treatment reversed PlexinD1-induced proliferation of LNCaP cells to the control level. We have added the relevant text in the revised manuscript (**page 15 line 514-517**).

"6. Authors should cite and discuss the following preprint:

Res Sq [Preprint]. 2024 Mar 27:rs.3.rs-4095949. doi: 10.21203/rs.3.rs-4095949/v1. Plexin D1 emerges as a novel target in the development of neural lineage plasticity in treatment-resistant prostate cancer. Chengfei Liu 1, Bo Chen, Pengfei Xu 2, Joy Yang 1, Christopher Nip 2, Leyi Wang 2, Yuqiu Shen 2, Shu Ning 1, Yufeng Shang 2, Eva Corey 3, Allen C Gao 1, Jason Gestwicki 4, Qiang Wei 5, Liangren Liu. PMID: 38585965 PMCID: PMC10996809 DOI: 10.21203/rs.3.rs-4095949/v1"

Response: We have cited and discussed this study (now published as Chen et al., *Oncogene*, 2024, 43:2325-2337, PMID: 38877132) in our revised manuscript (**page 7 line 217-219 and page 17 line 605-608**).

Referee #3 (Comments on Novelty/Model System for Author):

"This study provides robust and convincing data, to justify the claims. The work is important to the field and of interest to more than specialists (includes semaphorin/Plexin signalling, prostate cancer, androgen deprivation therapy resistance).

Some of the claims are not very novel however, as a similar phenotype has been shown for the ligands of PlexinD1 (the protein concerned in the paper under review).

Similarities to previous papers:

- Sema3C (a ligand for PlexinD1) expression levels increase in castration-resistant prostate cancer (CRPC) and functions to promote cancer cell growth and resistance to androgen receptor pathway inhibition, while SEMA3C inhibition (with a decoy receptor to PlexinB1) delays CRPC and enzalutamide-resistant progression. (Peacock et al. 2018)*
- Expression of Sema3E and its receptor Plexin D1 correlates with the metastatic progression of human tumors. PlexinD1 (and Sema3E) depletion reduces metastatic potential of xenografts of lung and breast cancer cell lines. (Casazza et al 2010)*
- Plexins promote HH signalling at level of Gli1 (mainly concerned with A-type plexins and MEFs) (Pinskey et al. 2022). (this has not been cited)*
- Sema3C promotes EMT in prostate cancer cells (Tam et al. 2017), AR transcription regulates Sema3C in a GATA-dependent manner (Tam et al. 2016). (this has not been cited)*

Some other experiments would strengthen the paper, specified in comments for authors.

Response: We thank the reviewer for considering our data robust and convincing and raising very constructive comments, which increased the strength of our manuscript, and we hope our detailed response will be satisfactory to the reviewer.

Referee #3 (Remarks for Author):

“In this manuscript the authors provide convincing evidence for a role for PlexinD1 in the progression of CRPC. Transcription of PlexinD1 is downregulated by AR while repression of AR signalling in response to androgen deprivation therapy leads to an increase in expression. Upregulation of PlexinD1 promotes an aggressive phenotype of increased migration, invasion, proliferation, EMT, stemness and neuroendocrine differentiation and tumour growth and metastasis in vivo, in a variety of prostate cancer cell lines. These findings are backed up by clinical evidence. PlexinD1 activates ErbB3 and cMet and promotes translocation of Gli1 to the nucleus and noncanonical Hedgehog signalling. Inhibition of PlexinD1 with a decoy receptor inhibits CRPC growth, migration and invasion.

The work is important and clinically relevant. Nevertheless, the following points should be addressed:”

Response: We thank the reviewer for recognizing the importance and clinical relevance of our work. We have performed new experiments and added clarifications to address the reviewer’s concerns as detailed below.

“1. Figure 1. The hypothesis is that AR down regulates PlexinD1, and so LNCaP, which expresses high AR, produces low PlexinD1, while cell lines expressing no AR (such as PC3, DU145) have higher levels of PlexinD1 mRNA. However, 22Rv1 expresses both AR and high PlexinD1 mRNA levels. How can this be explained?”

In contrast to results in Figure 1 of the manuscript, Peacock et al. find that LNCaP and 22RV1 have similar levels of PlexinD1 protein and PC3 and DU145 have very low PlexinD1 expression. A western blot comparing PlexinD1, Sema3C and Sema3E protein levels in all the prostate cancer cell lines used, including LNCaP and 22RV1 would be helpful (the cell lines may differ somewhat between labs) to clarify this concern.”

Response: We thank the reviewer for bringing out this intriguing point. As suggested, we performed new Western blotting experiments to compare PlexinD1, Sema3C, and Sema3E protein expression levels in multiple human prostate cancer cell lines (including LNCaP and 22Rv1) and a human normal prostate epithelial cell line RWPE-1. Our new data showed that PlexinD1, Sema3C, and Sema3E have similar protein expression patterns in these cell lines, with more abundance in most CRPC cell lines than in RWPE-1 and androgen-dependent prostate cancer cell lines (**New Fig. 1G**). Please note that unequal amounts of total cell lysates from different cell lines were loaded intentionally due to the large discrepancy in target protein levels among different cell lines as indicated in figure legend. Of note, our new data also demonstrated that PlexinD1 protein is most highly expressed in AR+ 22Rv1 cells among all the prostate cancer cell lines examined, which was confirmed in a recent study (refer to Fig. 1H, Chen et al., *Oncogene*, 2024, 43:2325-2337, PMID: 38877132). This could be explained in part by the loss of canonical AR signaling as evidenced by very low levels of PSA expression and enrichment of neuroendocrine markers in 22Rv1 cells as reported in a recent preprint (refer to Supplementary Figs. 5B and 5C, Thaper et al., *bioRxiv*, 2022, doi: <https://doi.org/10.1101/2022.05.04.490172>; or <https://www.biorxiv.org/content/10.1101/2022.05.04.490172v1>). Given the higher level of PlexinD1 mRNA in 22Rv1 cells compared with other AR+ prostate cancer cell lines like LNCaP (old Fig. 1G, and now New Appendix Fig. S1A), we speculated that additional transcriptional regulators may outcompete AR repression of PlexinD1 to upregulate PlexinD1 expression particularly in 22Rv1 cells. Following this idea, we

interrogated the Cancer Cell Line Encyclopedia (CCLE), a database including gene expression data for human cancer cell lines, and identified BRN2, a neural transcription factor known as an AR-suppressed driver of neuroendocrine differentiation (Bishop et al., *Cancer Discov*, 2017, 7:54-71, PMID: 27784708), expressed with a notably higher mRNA level in 22Rv1 cells than in other common prostate cancer cell lines (e.g., LNCaP, PC-3, and DU145), which was confirmed by the recent preprint (refer to Supplementary Fig. S5A, Thaper et al., bioRxiv, 2022 as cited above). We also demonstrated substantially higher expression levels of BRN2 protein in 22Rv1 and C4-2B^{ENZR} than in other prostate cancer cell lines (**New Appendix Fig. S1C**), which is in accord with the PlexinD1 protein expression pattern in these cell lines (New Fig. 1G). Further, we showed that BRN2 knockdown reduced PlexinD1 mRNA levels in 22Rv1 cells, while overexpression of BRN2 enhanced PlexinD1 mRNA levels in LNCaP cells, compared to respective controls (**New Appendix Fig. S1D**). These data suggested BRN2 as a potential transcriptional regulator likely to contribute to the particular abundance of PlexinD1 in 22Rv1 cells. We have added the relevant text in the revised manuscript (**page 4 line 141 – page 5 line 148, page 7 line 216-233, and page 33 line 1020-1024**).

As shown in **New Fig. 1G**, we found that 22Rv1, PC-3, and DU145 cells do express higher levels of PlexinD1 protein than LNCaP cells to various extents using our cell lines, which is similar to the PlexinD1 mRNA expression pattern in these cell lines (old Fig. 1G, now New Appendix Fig. S1A). These cell lines were purchased from ATCC and authenticated by short tandem repeat profiling recently. Of note, our conclusion that 22Rv1 cells have a much higher level of PlexinD1 protein than LNCaP cells corroborated the findings from the recent study (refer to Fig. 1H, Chen et al., *Oncogene*, 2024, 43:2325-2337, PMID: 38877132) showing that 22Rv1 cells expressed substantially more PlexinD1 protein than C4-2B cells (a LNCaP-derived cell line). Examining Appendix Fig. S3A in the study by Peacock et al., 22Rv1 cells also appear to have a modest increase in PlexinD1 protein expression compared with LNCaP cells. Nevertheless, we agree with the reviewer that the discrepancy in PlexinD1 protein expression in the same prostate cancer cell lines observed between our study and Peacock et al. could be due to differing characteristics, including specific protein expressions, of the same cell lines between different labs, possibly influenced by multiple factors such as cell sources, passages, and culture methods.

“2. Figure 3. shows convincingly that PlexinD1 expression is suppressed by AR signalling in LNCaP and LAPC4 cells - R1881 inhibits PlexinD1 expression, while enzalutamide treatment increases expression of PlexinD1. PlexinD1 has repressive androgen response element (AREs) in its promoter.

These findings are interesting in relation to those of Tam et al. (2016) who found that R1881 increases, while enzalutamide treatment decreases, Sema3C expression (a ligand of PlexinD1), due to an activating ARE in the promoter of Sema3C.

One wonders why AR binding to the promoters of receptor and ligand genes AREs have opposite effects? In this scenario, is PlexinD1 perhaps acting in a ligand-independent manner or is it activated by Sema3E binding? This paper should be referred to and discussed.”

Response: We performed new experiments demonstrating that, similar to PlexinD1, AR signaling negatively regulates the transcription and mRNA expression of SEMA3C and SEMA3E. Androgen depletion with charcoal-stripped serum-supplemented medium promoted, while R1881 reduced their mRNA levels in several androgen-responsive AR⁺ prostate cancer cell lines (LNCaP, LAPC4, and VCaP) (**New Fig. 3A**), which corroborated the upregulation of these Sema3 ligands in AR-repressed C4-2B^{ENZR} cells as we observed previously (Fig. 1E). We also noted the study by Tam et al. where R1881 increased, while enzalutamide decreased, Sema3C mRNA levels in LNCaP cells, which was driven by AR through an activating ARE within the Sema3C intron 2. We argue that these conflicting results may be due in part to potential differing characteristics like androgen responses of specific genes in the LNCaP cells used in different studies, which could be influenced by multiple factors such as cell sources, passages, and culture methods, and we stress the necessity for further clarifications from additional investigations. We have

added the relevant text in the revised manuscript (page 5 line 178 – page 6 line 182 and page 18 line 635-645).

“3. The issue above raises the question as to the role of ligand binding in PlexinD1 activation. Experiments should be performed to establish this role. (ie expression vs activation)

No ligand was added in the experiments involving PlexinD1 knock down in comparison to non-silenced controls suggesting that PlexinD1 works in the absence of ligand. However, serum present in the medium contains some semaphorins and endogenous SemaE is expressed by many prostate cancer cell lines. Furthermore, ectopic overexpression of Plexins have been shown to mimic ligand binding (Giordano et al.). Peacock et al show that Sema3C activates PlexinB1 in LNCaP and that inhibition of PlexinD1 has little effect on Sema3C -induced growth). Could the effects shown in the manuscript under review, at endogenous levels of PlexinD1, be induced by Sem3E, another ligand for PlexinD1? To resolve this issue, experiments should be performed to establish the effect of Sema3E and Sema3C on the aggressive phenotype of the prostate cancer cell lines used (in serum free medium), in the presence and absence of PlexinD1.”

Response: We thank the reviewer for raising this critical point. As suggested, we performed new experiments to determine the effect of Sema3C and Sema3E as PlexinD1 ligands on CRPC cell proliferation. To this end, we treated control and PlexinD1-knockdown cells with recombinant Sema3C or Sema3E proteins in serum-free medium. Our new data demonstrated that both Sema3 ligands induced the proliferation of control C4-2B^{ENZR} and 22Rv1 cells, with a stronger effect by Sema3E than Sema3C, while these inductions were reversed upon PlexinD1 knockdown (**New Appendix Fig. S2**). Of note, Sema3C but not Sema3E still caused modest but statistically significant increases in the proliferation of both cell lines in the absence of PlexinD1, which could be likely due to activation of other Plexins such as PlexinB1, which is known to mediate Sema3C's pro-proliferative effect in CRPC cells (Peacock et al., EMBO Mol Med, 2018, 10:219-238, PMID: 29348142). Based on our new data, we concluded that both Sema3C and Sema3E could utilize PlexinD1 to promote CRPC proliferation, with more reliance on PlexinD1 for Sema3E. We have added the relevant text in the revised manuscript (page 7 line 240-249 and page 18 line 624-626). We have also cited the reference pointed out by the reviewer (Giordano et al.) in the revised manuscript.

We noted that siRNA-mediated knockdown of PlexinD1 had little effect on Sema3C-induced growth of DU145 cells as shown in Fig. 2A of the study by Peacock et al. We argue that this could be due to the very low expression level of PlexinD1 protein detected in their DU145 cells as shown in Appendix Figure S3A of their study. By contrast, we demonstrated very high expression levels of PlexinD1 protein in both C4-2B^{ENZR} and 22Rv1 cells, thus rendering the Sema3C-induced growth of these cells responsive to PlexinD1 depletion in our study.

“4. Supplementary Figure 1. In contrast to the current manuscript, Peacock et al showed that siRNA-mediated silencing of Plexin D1 did not reduce proliferation of DU145.”

Response: As mentioned above to address point #3 from this reviewer, we noted these differing results between ours and the study by Peacock et al. regarding the experiments involving DU145 cells. We suggest that this discrepancy results from the different expression levels of PlexinD1 protein in the DU145 cells used in the different studies, which could be influenced by multiple factors such as cell sources, passages, and culture methods. Compared to respective PlexinD1-low LNCaP cells, we demonstrated a significantly higher expression level of PlexinD1 protein in our DU145 cells (New Fig. 1G), while a lower level of PlexinD1 protein was shown in the DU145 cells used by Peacock et al. (Appendix Fig. S3A). The varying PlexinD1 expression levels likely account for the contrasting responses of different DU145 cells to PlexinD1 blockade. We have included a detailed discussion of these differing observations in the revised manuscript (page 19 line 674 – page 20 line 684).

“5. Figure 6. Tam et al.(sci rep2017) show that ectopic expression of SEMA3C in RWPE-1 promotes a

similar phenotype to that shown in this manuscript upon overexpression of PlexinD1 (upregulation of EMT and stem markers, cell plasticity, migration and invasion in vitro and cell dissemination in vivo). This should be referred to.”

Response: We have cited this reference and added the relevant text in the revised manuscript (**page 18 line 627-630**).

“6. Figure 7. Is the interaction of PlexinD1 with ErbB3 direct? Co-IP of purified proteins would resolve this issue.”

Response: As suggested, we performed a new co-IP experiment with a mix of both PlexinD1 and ErbB3 recombinant proteins in solution and demonstrated their direct interaction (**New Appendix Fig. S10**). We have added the relevant text in the revised manuscript (**page 12 line 422-424**).

“7. Figure 8H, these images are not convincing. Better images are required showing markers of cell membrane or cytoplasm in addition.

To confirm the nuclear translocation of Gli1 upon activation of PlexinD1, experiments should be performed to investigate the effect of SemaE and Sema3C stimulation of cells on the nuclear localisation of ectopically expressed tagged-Gli1.

Western blots of cell fractionation - the marker for cytoplasmic proteins should be shown in the nuclear extract and the marker for nuclear proteins should be shown for the cytoplasmic proteins, to ensure the purity of the samples.

A Link between Plexins and HH signalling and Gli 1 has been shown before (Pinsky 22): 'Plexins promote Hedgehog signaling through their cytoplasmic GAP activity' - MEFs treated with siRNA oligos for Plxna3, Plxnb2, and Plxnd1 showed reduced activation of Gli1. This paper should be mentioned.”

Response: As suggested, we performed new co-IF experiments to visualize the nuclear localization of endogenous Gli1 upon manipulation of PlexinD1 levels using α -tubulin as a cytoplasm marker. Our new data demonstrated similar results as reported previously (**New Fig. 8H**). We also performed new experiments to determine the effect of Sema3C and Sema3E on the nuclear translocation of exogenously transfected FLAG-tagged Gli1 in 22Rv1 cells using an IF assay with an anti-FLAG antibody. Our new data demonstrated less or more nuclear accumulation of FLAG-tagged Gli1 protein upon treatment with SEMA3C/SEMA3E siRNA or recombinant proteins respectively, compared to the control (**New Appendix Fig. S12D**). We have also included an updated **Fig. 8I** with both cytoplasmic and nuclear protein markers shown on the same membranes in the revised manuscript. Lastly, we have cited the reference (Pinsky 22) as pointed out by the reviewer and added/modified the relevant text in the revised manuscript (**page 14 line 485-492, page 14 line 495 – page 15 line 505, and page 19 line 651-654**).

“8. Figure 9.

Western blots of the purified D1SP protein from the cell lysate and conditioned medium should be shown to ensure that the protein is of the correct size.

A dose response to added D1SP protein should be shown in the RTK experiments.

How does the D1SP work? Does it bind to PlexinD1 or its ligands, or prevent RTK/PlexinD1 interaction? As mentioned in this manuscript, Peacock et al used a similar decoy receptor. They showed that, compared to B1SP, D1SP was an ineffective inhibitor of LNCaP cell growth and that D1SP showed no inhibitory activity of SEMA3C-induced activation of RTK signaling in both LNCaP (Appendix Fig S5E) or DU145 cells. This is somewhat at odds to the new manuscript, which shows that D1SP inhibits proliferation and

aggressive behaviour of 22rv1 and c4-2b enz cells and growth and ERB3 and MET phosphorylation in xenografted 22rv1 tumours. These differing results should be discussed.

Does D1SP inhibit RTK phosphorylation in 22rv1, DU14 and or LNCaP cells?"

Response: As suggested, we performed new Western blotting experiments demonstrating D1SP migrated at a size of ~100 kDa in both whole cell lysate and conditioned medium using a human IgGFC-specific antibody for D1SP recognition (**New Fig. 9B**). D1SP has an original size of 86 kDa, but due to the glycosylation that is predicted to occur on multiple positions of D1SP, its real size is anticipated to shift up, which could fall in the range of 100 kDa as we observed. We also performed new sets of experiments to clarify D1SP's potential mechanism of action. Our new data demonstrated direct binding of D1SP to PlexinD1, Sema3C, and Sema3E by co-IP assays of a mix of individual pairs of recombinant proteins in solution (**New Appendix Figs. S13A and S13B**). Our new data also demonstrated that D1SP impaired the formation of PlexinD1-ErbB3 and PlexinD1-cMet endogenous protein complexes in both C4-2B^{ENZR} and 22Rv1 cells by PLA assays (**New Appendix Fig. S13G**). Further, our new data showed that D1SP treatment reduced the phosphoprotein levels of ErbB3, ErbB2, cMet, ERK, and AKT in PlexinD1-high C4-2B^{ENZR}, 22Rv1 and DU145 cells but not that of PlexinD1-low LNCaP cells (**New Appendix Figs. S13H and S13I**). We have added the relevant text in the revised manuscript (**page 15 line 539 – page 16 line 544, page 16 line 546-550, page 16 line 573 – page 17 line 583, and page 19 line 666-667**).

We also compared our data with the observations of Peacock et al. regarding the effect of D1SP in different prostate cancer cells. Peacock et al. demonstrated D1SP as an ineffective inhibitor of the growth and RTK signaling in LNCaP cells (refer to Appendix Figs. S5C and S5E in the study by Peacock et al.), which is consistent with our new findings (**New Appendix Figs. S13D and S13E – to address Reviewer 1 point #3 and Reviewer 2 point #1, see above, and New Appendix Fig. S13I**). However, Peacock et al. observed no inhibitory activity of Sema3C-induced activation of RTK signaling by D1SP treatment in DU145 cells (refer to Appendix Fig. S5F in the study by Peacock et al.), while we found decreased RTK signaling by D1SP treatment in DU145 cells (New Appendix Fig. S13I). As mentioned in the response to point #4 from this reviewer above, we argue that this discrepancy may result from the different expression levels of PlexinD1 protein in the DU145 cells as used in different studies, as possibly influenced by multiple factors such as cell sources, passages, and culture methods. Compared to respective PlexinD1-low LNCaP cells, we demonstrated a significantly higher expression level of PlexinD1 protein in our DU145 cells (New Fig. 1G), while a lower level of PlexinD1 protein was shown in the DU145 cells used by Peacock et al. (Appendix Fig. S3A). The varying PlexinD1 expression levels are likely to account for the contrasting responses of different DU145 cells to PlexinD1 blockade. Also, as mentioned in the response to point #1 for this reviewer above, both 22Rv1 and C4-2B^{ENZR} cell lines express much more abundant levels of PlexinD1 protein compared with LNCaP or its derivative C4-2B cells, as shown by us (New Fig. 1G) and a recent study (refer to Fig. 1H where C4-2B MDVR is the same as our C4-2B^{ENZR} with just different names used, Chen et al., *Oncogene*, 2024, 43:2325-2337, PMID: 38877132), which may render 22Rv1 and C4-2B^{ENZR} cells responsive to D1SP treatment. Nevertheless, coupling our study with others may reveal a correlation between D1SP's effectiveness and PlexinD1 expression levels in prostate cancer cells, which may suggest the potential therapeutic utility of D1SP specifically in PlexinD1-high prostate cancer. We have included a detailed discussion of these points in the revised manuscript (**page 19 line 674 – page 20 line 686**).

"9. Discussion. The authors state that: 'overexpression of PlexinD1 without modulation of Sema ligand levels was sufficient to promote PCa cell proliferation, migration, invasion, ENZ resistance, stemness, and plasticity' - overexpression has been shown to mimic ligand binding, probably due to clustering of receptors, in in vitro ectopic expression experiments (eg Giordano et al.)."

Response: We have cited this reference and added the relevant text in the revised manuscript (**page 18 line 620-622**).

“Minor changes
Figure 3I, the figure key too small”

Response: We have enlarged the figure key of Fig. 3I in the revised manuscript.

“References.

Peacock JW, Takeuchi A, Hayashi N, Liu L et al. *SEMA3C drives cancer growth by transactivating multiple receptor tyrosine kinases via Plexin B1*. *EMBO Mol Med*. 2018

Tam, K.J., Hui, D.H.F., Lee, W.W. et al. *Semaphorin 3 C drives epithelial-to-mesenchymal transition, invasiveness, and stem-like characteristics in prostate cells*. *Sci Rep* 7, 11501 (2017). <https://doi.org/10.1038/s41598-017-11914-6>

Tam K. J, Dalal K., Hsing M., et al *Androgen receptor transcriptionally regulates semaphorin 3C in a GATA2-dependent manner*. *Oncotarget*. 2017; 8: 9617-9633.

Pinskey JM, Hoard TM, Zhao XF, et al. *Plexins promote Hedgehog signaling through their cytoplasmic GAP activity*. *Elife*. 2022

Casazza A, Finisguerra V, Capparuccia L, et al. *Sema3E-Plexin D1 signaling drives human cancer cell invasiveness and metastatic spreading in mice*. *J Clin Invest*. 2010 Aug;120(8):2684-98.

Giordano S, Corso S, Conrotto P, et al. *The semaphorin 4D receptor controls invasive growth by coupling with Met*. *Nat Cell Biol*. 2002 Sep;4(9):720-4.”

Response: We have cited all these reference above in the revised manuscript.

2nd Dec 2024

Dear Dr. Wu,

Thank you for submitting your revised study. We have now received the reports from the three referees who evaluated your revised manuscript. As you will see from the reports below, they are satisfied with the revisions, and I will therefore be able to accept your manuscript once the following editorial issues will be addressed:

1/ Manuscript text:

- Please remove the red text and only keep in track changes mode any new modification.
- We can accommodate a maximum of 5 keywords, please adjust accordingly.
- Methods:
 - o Thank you for providing a reagents and tools table, please remove it from the manuscript, and upload it as a separate file (using our template that you can find in our author guideline <https://www.embopress.org/page/journal/14693178/authorguide#structuredmethods>).
 - o Clinical specimens: please include a statement confirming that informed consent was obtained from all subjects and that the experiments conformed to the principles set out in the WMA Declaration of Helsinki and the Department of Health and Human Services Belmont Report (if applicable), and adjust the checklist accordingly if needed.
 - o Please provide antibodies dilutions/concentrations.
- Data availability: thank you for depositing your sequencing data. Please note that the dataset must be made public before acceptance of the manuscript.
- Please remove "Appendix Supplementary Material - The Appendix Supplementary Material includes 13 appendix figures and 4 appendix tables." from the manuscript.
- Acknowledgements: The information provided in the manuscript and the submission system should match (currently, WSU startup funds; grants W81XWH-18-2-0013, W81XWH-18-2-0015, W81XWH-18-2-0016, W81XWH-18-2-0017, W81XWH-18-2-0018, W81XWH-18-2-0019 are missing from the submission system).
- Author contributions: CRediT has replaced the traditional author contributions section because it offers a systematic machine readable author contributions format that allows for more effective research assessment. Please remove the Authors Contributions from the manuscript and use the free text boxes beneath each contributing author's name in our system to add specific details on the author's contribution. More information is available in our guide to authors.

2/ Figures and Appendix:

- Appendix: please include a table of content and page numbers. Please know that you have the possibility to have up to 5 Expanded View figures that are collapsible/expandable online. EV Figures should be cited as "Figure EV1, Figure EV2" etc... in the text and their respective legends should be included in the main text after the legends of regular figures. For the figures that you do NOT wish to display as Expanded View figures, they should be bundled together with their legends in a single PDF file called *Appendix*.
- Please address the queries from our copy editors in the figure legends:
 1. Please indicate the statistical test used for data analysis in the legends of figures 1A, C, D; 3H, 4D, 5D, 6A, I, J; 7A, 8A, M.
 2. Please note that n=2 in figure 7B. Kindly remove the statistical analyses, error bars, and only keep the individual data points.
 3. Although 'n' is provided, please describe the nature of entity for 'n' in the legends of figures 1F, 2A-G; 6H, 7E, 9D.

3/ Please note that all corresponding authors are required to supply an ORCID ID for their name upon submission of a revised manuscript. An ORCID ID is currently missing for Jing Wang.

4/ Thank you for providing a synopsis text and image. I have cropped the attached portion to serve as thumbnail for the table of content on our webpage. Please let us know in case you disagree and would rather propose a different image (115px x 70 px) as changes during proofing are usually not allowed.

5/ As part of the EMBO Publications transparent editorial process initiative (see our Editorial at <http://embomolmed.embopress.org/content/2/9/329>), EMBO Molecular Medicine will publish online a Review Process File (RPF) to accompany accepted manuscripts.

This file will be published in conjunction with your paper and will include the anonymous referee reports, your point-by-point response and all pertinent correspondence relating to the manuscript. Let us know whether you agree with the publication of the RPF.

I look forward to receiving your revised manuscript.

Yours sincerely,

Lise Roth

To submit your manuscript, please follow this link:

<https://embomolmed.msubmit.net/cgi-bin/main.plex>

***** Reviewer's comments *****

Referee #1 (Remarks for Author):

The Authors have adequately addressed my concerns.

Referee #2 (Comments on Novelty/Model System for Author):

no issues

Referee #2 (Remarks for Author):

The authors responded well to my questions.

Referee #3 (Comments on Novelty/Model System for Author):

Substantial extra work has been done of high technical quality in answer to all my comments. (Although the immunofluorescence images in supplementary figure S1D are improved, they are still not of high quality; however the conclusions from these experiments are backed up by the western blots, which are of high quality, so in my opinion the IF results are adequate).

Novelty: There is some overlap of results with other groups (eg. Peacock et al.2018) but these previous papers have now been cited and discussed.

Medical impact: the manuscript shows a potential therapeutic impact of a decoy PlexinD1 truncated protein, administered by intratumoral injections, on CRPC in mouse models.

Model system: many different prostate cancer cell lines models are used for the many in vitro mechanistic studies, and xenograft models for the in vivo work. Relevance to human cancer is shown in tumour tissue and expression databases.

Referee #3 (Remarks for Author):

I would like to thank the authors for the detailed discussion and substantial new experimental work performed in response to my comments, both of which strengthen the manuscript. I have no further comments to suggest, and I recommend accepting the manuscript in the current state.

Dear Dr. Wu,

During our standard figure check, some anomalies were

found:

- Blot re-use between Figure 4A Actin & Figure 5C Actin. (Blots also look the same in the attached SD.)
- Possible cell reuse within Figure Appendix S13 - G. Vehicle, Top and bottom cells.

Please carefully check you figures/data and clarify.

With kind regards,

Lise Roth

Dear Dr. Roth:

After a careful examination of the problematic images and related source data, we found that we accidentally used the wrong images for the left panel (C4-2B ENZR) Actin in Figure 5C and the 22Rv1 Vehicle top panel in Appendix Figure S13G during preparation of the figures. We are able to provide correct raw images for further evaluation.

We appreciate you giving us the opportunity to make clarification above and apologize for our unintentional oversight regarding the misuse of images. Please let us know if and how we can make corrections and provide updated figures/source data for the manuscript at this point. Thank you.

Best,

-Jason

Boyang (Jason) Wu, Ph.D.
Associate Professor
Dept of Pharmaceutical Sciences
College of Pharmacy and Pharmaceutical Sciences
Washington State University
[E-mail: boyang.wu@wsu.edu](mailto:boyang.wu@wsu.edu)
Phone: 509-368-6691

The authors addressed the remaining formatting issues.

11th Dec 2024

Dear Dr. Wu,

Thank you for submitting the revised files. I am pleased to inform you that your manuscript is accepted for publication and is now being sent to our publisher to be included in the next available issue of EMBO Molecular Medicine.

With kind regards,

Lise Roth
